# Evaluation of global ocean–sea-ice model simulations based on the experimental protocols of the Ocean Model Intercomparison Project phase 2 (OMIP-2)

Hiroyuki Tsujino[1], L. Shogo Urakawa[1], Stephen M. Griffies[2,3], Gokhan Danabasoglu[4], Alistair J. Adcroft[3,2], Arthur E. Amaral[5], Thomas Arsouze[5], Mats Bentsen[6], Raffaele Bernardello[5], Claus W. Böning[7], Alexandra Bozec[8], Eric P. Chassignet[8], Sergey Danilov[9], Raphael Dussin[2], Eleftheria Exarchou[5], Pier Giuseppe Fogli[10], Baylor Fox-Kemper[11], Chuncheng Guo[6], Mehmet Ilicak[12,6], Doroteaciro Iovino[10], Who M. Kim[4], Nikolay Koldunov[13,9], Vladimir Lapin[5], Yiwen Li[14,15], Pengfei Lin[14,15], Keith Lindsay[4], Hailong Liu[14,15], Matthew C. Long[4], Yoshiki Komuro[16], Simon J. Marsland[17], Simona Masina[10], Aleksi Nummelin[6], Jan Klaus Rieck[7], Yohan Ruprich-Robert[5], Markus Scheinert[7], Valentina Sicardi[5], Dmitry Sidorenko[9], Tatsuo Suzuki[16], Hiroaki Tatebe[16], Qiang Wang[9], Stephen G. Yeager[4], Zipeng Yu[14,15]

[1]JMA Meteorological Research Institute (MRI), Tsukuba, Ibaraki, Japan
[2]NOAA Geophysical Fluid Dynamics Laboratory (GFDL), Princeton, NJ 08542, USA
[3]Princeton University Atmospheric and Oceanic Sciences Program, Princeton, NJ 08540, USA
[4]National Center for Atmospheric Research (NCAR), Boulder, CO, USA
[5]Barcelona Supercomputing Center, Barcelona, Spain
[6]NORCE Norwegian Research Centre, Bjerknes Centre for Climate Research, Bergen, Norway
[7]GEOMAR Helmholtz Centre for Ocean Research, Kiel, Germany
[8]Center for Ocean-Atmosphere Prediction Studies (COAPS), Florida State University, Tallahassee, FL, USA
[9]Alfred-Wegener-Institut Helmholtz-Zentrum für Polar- und Meeresforschung (AWI), Bremerhaven, Germany
[10]Ocean Modeling and Data Assimilation Division, Centro Euro-Mediterraneo sui Cambiamenti Climatici, Bologna, Italy
[11]Department of Earth, Environmental, and Planetary Sciences, Brown University, Providence, RI, USA
[12]Eurasia Institute of Earth Sciences, Istanbul Technical University, Istanbul, Turkey
[13]MARUM-Center for Marine Environmental Sciences, Bremen, Germany
[14]LASG, Institute of Atmospheric Physics, Chinese Academy of Sciences, Beijing 100029, China
[15]College of Earth and Planetary Sciences, University of Chinese Academy of Sciences, Beijing 100049, China
[16]Japan Agency for Marine-Earth Science and Technology (JAMSTEC), Yokohama, Japan
[17]CSIRO Oceans and Atmosphere, Aspendale, Australia

*Correspondence to*: Hiroyuki Tsujino (htsujino@mri-jma.go.jp)

**Abstract.** We present a new framework for global ocean–sea-ice model simulations based on phase 2 of the Ocean Model Intercomparison Project (OMIP-2), making use of the JRA55-do atmospheric dataset. We motivate the use of OMIP-2 over the framework for the first phase of OMIP (OMIP-1), previously referred to as the Coordinated Ocean–ice Reference Experiments (CORE), via the evaluation of OMIP-1 and OMIP-2 simulations from eleven (11) state-of-the-science global ocean–sea-ice models. In the present evaluation, multi-model ensemble means and spreads are calculated separately for the OMIP-1 and OMIP-2 simulations and overall performances are assessed considering metrics commonly used by ocean modelers. Both OMIP-1 and OMIP-2 multi-model ensemble ranges capture observations in more than 80% of the time and

region for most metrics, with the multi-model ensemble spread greatly exceeding the difference between the means of the two datasets. Many features, including some climatologically relevant ocean circulation indices, are very similar between OMIP-1 and OMIP-2 simulations, and yet we could also identify key qualitative improvements in transitioning from OMIP-1 to OMIP-2. For example, the sea surface temperature of the OMIP-2 simulations reproduce the observed global warming during the 1980s and 1990s, as well as the warming slowdown in the 2000s and the more recent accelerated warming, which were absent in OMIP-1, noting that the last feature is part of the design of OMIP-2 because OMIP-1 forcing stopped in 2009. A negative bias in the sea-ice concentration in summer of both hemispheres in OMIP-1 is significantly reduced in OMIP-2. The overall reproducibility of both seasonal and interannual variations in sea surface temperature and sea surface height (dynamic sea level) is improved in OMIP-2. These improvements represent a new capability of the OMIP-2 framework for evaluating process-level responses using simulation results. Regarding the sensitivity of individual models to the change in forcing, the models show well-ordered responses for the metrics that are directly forced while they show less-organized responses for those that require complex model adjustments. Many of the remaining common model biases may be attributed either to errors in representing important processes in ocean–sea-ice models, some of which are expected to be reduced by using finer horizontal and/or vertical resolutions, or to shared biases and limitations in the atmospheric forcing. In particular, further efforts are warranted to resolve remaining issues in OMIP-2 such as the warm bias in the upper layer, the mismatch between the observed and simulated variability of heat content and thermosteric sea level before 1990s, and the erroneous representation of deep and bottom water formations and circulations. We suggest that such problems can be resolved through collaboration between those developing models (including parameterizations) and forcing datasets. Overall, the present assessment justifies our recommendation that future model development and analysis studies use the OMIP-2 framework.

## 1 Introduction

The Ocean Model Intercomparison Project (OMIP) was endorsed by the phase 6 of the World Climate Research Programme (WCRP) Coupled Model Intercomparison Project (CMIP6; Eyring et al., 2016). It was proposed by an international group of ocean modelers and analysts involved in the development and analysis of global ocean–sea-ice models that are used as components of the climate and earth system models participating in CMIP6. OMIP consists of physical (Griffies et al., 2016) and biogeochemical (Orr et al., 2017) parts. The physical part of CMIP6-OMIP has been organized by the Ocean Model Development Panel (OMDP) of the WCRP core program Climate and Ocean Variability, Predictability, and Change (CLIVAR). Prior to OMIP, the OMDP developed the Coordinated Ocean–ice Reference Experiments (COREs) framework and comprehensively assessed the performance of global ocean–sea-ice models (Griffies et al., 2009; Danabasoglu et al., 2014, Griffies et al., 2014, Downes et al., 2015, Farneti et al., 2015, Danabasoglu et al., 2016, Wang et al., 2016a; 2016b, Ilicak et al., 2016, Tseng et al., 2016, Rahaman et al., 2020). CORE has successfully evolved into phase 1 of the physical part of OMIP (OMIP-1). The framework of CORE has provided ocean modelers with both a common facility to perform

global ocean–sea-ice model simulations and a useful benchmark for evaluating simulations in comparison with other models
and observations.

The essential element facilitating OMIP is the atmospheric and river runoff forcing datasets for computing boundary
fluxes needed to drive global ocean–sea-ice models. CORE / OMIP-1 make use of the dataset documented by Large and
Yeager (2009). The Large and Yeager (2009) dataset consists of surface atmospheric states based on the National Centers for
Environmental Prediction / National Center for Atmospheric Research (NCEP/NCAR) atmospheric reanalysis (Kalnay et al.,
1996; Kistler et al., 2001), also comprising surface downward radiation based on ISCCP-FD (Zhang et al., 2004), hybrid
precipitation based on several sources, and the river runoff based on Dai and Trenberth (2009). The datasets and protocols
for computing boundary fluxes are designed to study climate mean and variability during the late 20$^{th}$ and early 21$^{st}$ centuries.

The Large and Yeager (2009) forcing dataset has not been updated since 2009 because of the discontinuation of ISCCP-
FD. Hence the CORE forcing only covers the period from 1948 to 2009. Since its release, various state-of-the-science
atmospheric reanalysis products have been produced. Requests for updating the CORE forcing dataset based on these newer
atmospheric reanalyses have naturally emerged. To update the forcing dataset and improve the experimental infrastructure,
Tsujino et al. (2018) developed a surface-atmospheric dataset based on Japanese 55-year atmospheric reanalysis (JRA-55;
Kobayashi et al., 2015), referred to as JRA55-do, under the guidance and support of CLIVAR-OMDP. The JRA55-do
forcing dataset has been endorsed under the protocols for phase 2 of CMIP6-OMIP (OMIP-2). It currently covers the period
from 1958 to 2018 with planned annual updates. Relative to CORE, the JRA55-do forcing has an increased temporal
frequency (from 6 hours to 3 hours) and refined horizontal resolution (from 1.875° to 0.5625°). In developing JRA55-do
forcing, various atmospheric states of JRA-55 have been adjusted to match reference states based on observations or the
ensemble means of atmospheric reanalysis products, as explained in detail by Tsujino et al. (2018). This approach leads to
surface atmospheric forcing fields based on a single reanalysis product (JRA-55) that are more self-consistent than the
previous CORE effort. The continental river discharge is provided by a river-routing model forced by river runoff from the
land-surface component of JRA-55 with adjustments to ensure similar long-term variabilities as seen in the CORE dataset
(Suzuki et al., 2018). Discharge of ice-sheets and glaciers from Greenland (Bamber et al., 2012; Bamber et al., 2018) and
Antarctica (Depoorter et al., 2013) are also incorporated.

As a contribution to CMIP6-OMIP, we present an evaluation of the response of CMIP6-class global ocean–sea-ice models
to the JRA55-do forcing dataset. Our evaluation takes the form of a comparison between OMIP-1 and OMIP-2 simulations
using metrics commonly adopted in the evaluation of global ocean–sea-ice models to assess their biases. As a result, the
present comparison offers an update to the benchmarks for evaluating global ocean–sea-ice simulations. In this first
coordinated evaluation of OMIP-2 simulations we also identify possible directions for revising OMIP-2 by generating
further improvements in the forcing dataset (JRA55-do) and experimental protocols.

In organizing and conducting this model intercomparison project, we use Atmospheric Model Intercomparison Project
(AMIP; Gates et al., 1999) as a guide. In the present assessment, it is beyond our scope to penetrate into any particular aspect
of individual models or specific ocean processes and climatic events. This approach thus offers a glimpse rather than an in-

depth view of the many elements of ocean–sea-ice model performance. Our presentation of the performance of a wide variety of ocean climate models forced by two kinds of atmospheric datasets allows us to establish the state-of-the-science for global ocean–sea-ice modeling in the year 2020.

Note that two companion papers complement aspects of the present assessment of forcing datasets and model performances. Chassignet et al. (2020) compare four pairs of low- and high-resolution ocean and sea-ice simulations forced for one cycle of the JRA55-do dataset to isolate the effects of horizontal resolutions on simulated ocean climate variables. All four low-resolution models (FSU-HYCOM, CESM-POP, AWI-FESOM, and CAS-LICOM3, see Table 1) used by Chassignet et al. (2020) participate in the present study. Stewart et al. (2020) propose repeat year forcing datasets derived from the JRA55-do dataset by identifying 12-month periods (not necessarily a single calendar year) that are most neutral in terms of major climate modes of variability. Each of several candidate periods is used repeatedly to force three CMIP6-class global ocean–sea-ice models for 500 years and simulation results are compared. Two models (CESM-POP and MRI.COM) participate in the present study.

This paper is organized as follows. Section 2 describes the design of the comparison and the experimental protocols for each of the OMIP-1 and OMIP-2 simulations. Section 3 compares spin-up behavior of participating models. Section 4 compares the simulations with contemporary climate. Interannual variability of the last cycle of the simulations are evaluated in section 5. Section 6 discusses aspects of model intercomparison, looking at ordering among models in various metrics and its sensitivity to the change in forcing. Section 7 provides a summary and conclusions.

Appendices offer details relevant to the present assessment. Appendix A presents brief descriptions of the models and experiments of the eleven (11) participating groups. Appendix B presents some sensitivity studies to help understand the present assessment and guide future revisions of forcing datasets and protocols. Appendix C describes observational datasets used in this evaluation. Appendix D presents specific values for metrics realized by individual models. Appendix E applies some typical objective assessments of model performance used by AMIP to the metrics used for evaluating ocean models.

## 2 Design of evaluation of the new framework

One of the main purposes of ocean–sea-ice model simulations forced with a realistic history of surface atmospheric state is to reproduce the contemporary ocean climate. CMIP6-OMIP aims to facilitate such efforts and to provide a benchmark for assessing the simulation quality. Here we conduct a general assessment of global ocean–sea-ice model simulations under a new framework by considering two different atmospheric forcing datasets, OMIP-1 (CORE) and OMIP-2 (JRA55-do), with contributing models using the same configuration for each dataset.

### 2.1 OMIP-1 Protocol

The protocol for the OMIP-1/CORE-forced simulation is detailed in Griffies et al. (2016), and requires five repeated cycles of the 62-year atmospheric forcing. However, in preliminary JRA55-do forced (OMIP-2) runs conducted by many modeling

groups, decline and recovery of the Atlantic meridional overturning circulation (AMOC) occurred during the first few cycles
before it reached a quasi-steady state. We thus found it necessary to perform no less than six cycles of the forcing for
JRA55-do, with the 4th through 6th cycles (that is, the last three cycles) suitable for studying the uptake and spread of
anthropogenic greenhouse gases under the protocols of the biogeochemical part of OMIP (Orr et al., 2017). Hence, to
facilitate a comparison of the behaviors between OMIP-1 and OMIP-2, each model here is run for six cycles under both
forcing, rather than the five cycles originally proposed by Griffies et al. (2016). For OMIP-1, the experiment results in a 372-
year simulation comprised of six cycles of the 62-year (1948–2009) CORE forcing from Large and Yeager (2009). In
addition to atmospheric and river runoff forcing, we restored sea surface salinity to the monthly climatology provided by
CORE, with restoring details, e.g., its strength, determined by the individual modeling groups. Computation of the surface
turbulent fluxes of momentum, heat, and freshwater follows the method detailed by Large and Yeager (2009). In particular,
we note that the flux calculations use the relative winds obtained by subtracting the full ocean surface currents from the
surface winds.

## 2.2 OMIP-2 Protocol

The protocol for the OMIP-2 simulations follows the OMIP-1 protocol yet with a few deviations. The simulation length is
366-years as realized by repeating six cycles of the 61-year (1958–2018) JRA55-do forcing dataset v1.4.0 (Tsujino et al.,
2018). Appendix B1 discusses the results of using common periods (1958–2009) of OMIP-1 and OMIP-2 to force a subset of
models to understand whether the difference in the forcing periods between OMIP-1 and OMIP-2 simulations has any
implications for model performances. Sea surface salinity restoring is based on monthly climatology of the upper 10 m
averaged sea surface salinity from WOA13v2 (Zweng et al., 2013). Though it is recommended to use formulae for the
properties of moist air as presented by Tsujino et al. (2018), we do not impose this condition on all participating groups.
Sensitivity to this setting is reported for the MRI model in Appendix B2.

Regarding the calculation of relative winds in the surface flux computations, we do not set a specified protocol for what
fraction, if any, of the ocean surface currents should be included. The reasons behind this approach are briefly explained
below, with more details presented in Appendix B3. There has been recent process-based research aimed at uncovering the
mechanisms that lead to imprints of ocean surface current on the atmospheric winds via air-sea coupling (Renault et al., 2016,
2017, 2019b). Correspondingly, there is active research in determining how best to force an ocean model with prescribed
atmospheric winds (Renault et al., 2019a, 2020). For example, the wind speed correction approach proposed by Renault et al.
(2016) acknowledges the imprint of the ocean currents on the surface winds in an ocean–sea-ice model (uncoupled from an
atmospheric model). This approach is realized by introducing a dimensionless parameter α that can be set between [0, 1]
when computing the vector velocity difference $\Delta\vec{U} = \vec{U_a} - \alpha\vec{U_o}$, where $\vec{U_a}$ is the surface (atmospheric) wind vector without
the imprint of the ocean current and $\vec{U_o}$ is the surface oceanic current vector (usually the vector at the first model level). The
community has not reached a consensus about the way α should be imposed on ocean–sea-ice models.

There also remains ambiguity as to what is represented by the prescribed winds ($\overrightarrow{U_a}$) depending on the way they are constructed from the satellite-based and reanalysis atmospheric wind products. This ambiguity becomes an issue with the OMIP-2 dataset. First, its wind field is based on the JRA-55 reanalysis, which assimilates scatterometer winds yet not necessarily reproduces winds identical to scatterometer winds depending on the level of assimilation constraints. Since scatterometer winds represent wind relative to the surface current (e.g., Plagge et al., 2012) and contain imprints of surface currents (Renault et al., 2017, 2019b), assimilating scatterometer winds directly, yet not identically, to the absolute surface winds of the atmospheric circulation model would make the feature of surface winds of the JRA-55 reanalysis somewhat ambiguous. Second, only the long-term mean JRA-55 winds are adjusted with respect to the satellite-based winds in constructing the OMIP-2 dataset (JRA55-do). As a result, the long-term mean winds of the OMIP-2 (JRA55-do) dataset could be regarded to be replicating their scatterometer wind counterparts, but ocean current imprints on them have not been clarified yet. On the other hand, in short time scales, ocean current imprints on winds are shown to be small, if not negligible, in the OMIP-2 (JRA55-do) forcing dataset (Abel, 2018), which would make them possible to be treated as absolute winds without imprints of surface currents at least in short time scales. A future version of the OMIP-2 dataset will aim to resolve this ambiguity. Readers are referred to Renault et al. (2020) for more discussion on the issues of using satellite derived winds to force uncoupled ocean models.

Given these ambiguities and lack of a consensus in the community, the OMIP-2 protocol does not specify a value for α. Nevertheless, it is preferable for the groups participating in CMIP6 to use the same value of α as in their CMIP6 climate models. Because many CMIP6 climate models choose α as unity (i.e., full effects of ocean currents are included in the stress calculation), we suggested that participants in the present comparison paper also set α = 1. Even so, it is premature at this time to recommend a specific protocol choice. Sensitivity to various approaches is reported in Appendix B3 by a subset of models in this study.

## 2.3 Model Assessment

Ocean models are known to exhibit a long-term drift after initialization even if they are initialized by modern estimates of temperature and salinity for the World Ocean (e.g., Figure 3 of Griffies et al., 2014). We look at the evolution of selected ocean climate metrics from the start of the integration and determine which metric becomes persistent between forcing cycles by the end (6th cycle) of the integration. Next, we assess the performance of the two forcing frameworks in reproducing contemporary climate by comparing spatial distributions of long-term multi-model ensemble means to those of observations. To represent contemporary climate, we adopt the period 1980–2009. For some metrics, we use different periods depending on availability of reference datasets. Then, interannual variations and trends of important ocean climate indices are assessed. A description about the observationally based datasets used for model evaluation is presented in Appendix C.

We use several statistical approaches to evaluate performances of simulations and forcing datasets. To evaluate the spatial distributions of long-term multi-model ensemble means from OMIP-1 and OMIP-2 simulations, we compare the bias

of the multi-model ensemble mean and the modeled 95% confidence range defined as twice the standard deviation of the multi-model ensemble at the grid point level and then assess whether the bias (the position of the observation relative to the ensemble mean) is within the modeled confidence range whose center is taken as the ensemble mean. Similarly, to evaluate the time series, we compare the bias and the modeled confidence range at each time. To compare the forcing datasets, we test the significance of the difference between OMIP-1 and OMIP-2 simulations using the method proposed by Wakamatsu et al. (2017), where uncertainty is evaluated as the square root of the uncertainty (variance) due to model variability, internal (temporal) variability, and small sample size. An ensemble of time series of the differences between the OMIP-1 and OMIP-2 simulations by models is evaluated to determine uncertainty at each grid point. The uncertainties are then used to test the significance of the ensemble mean of the differences. To evaluate performance of individual models, some globally integrated quantities such as root-mean-square biases and global means of metrics are computed for the OMIP-1 and OMIP-2 simulations by individual models and the robustness of their relative positions against the change in forcing datasets is tested using linear fitting. This assessment is presented in section 6, with results from individual models listed in Appendix D. Some additional statistical assessments on overall performance of models are also presented by following the approach taken by AMIP as detailed in Appendix E.

The diagnostic data needed to perform the above assessments are largely covered by Priority-1 diagnostics of OMIP provided by Griffies et al. (2016). The following additional diagnostics are requested by contributing groups, which can be generated based on the Priority-1 diagnostics.

- Vertically averaged temperature for 0 – 700 m, 0 – 2000 m, and 2000 m – bottom.
- Atlantic meridional overturning circulation (AMOC) maximum at 26.5°N.
- All diagnostics are gridded on a standard 1° latitude – 1° longitude grid with 33 depth levels, used by older versions (until WOA09) of the World Ocean Atlas datasets.

Eleven (11) groups listed on Table 1 participated in this intercomparison paper, with details of model configurations and experiments summarized in Appendix A and Table A1. This is a small number of participating groups relative to more than 60 models that registered for CMIP6-OMIP. The reason for using only a subset of models is that we here compare two simulations, with the OMIP-2 (JRA55-do v1.4) forcing only becoming available in 2018. Nonetheless, the chosen models well represent the diversity in ocean models as of 2020 in terms of modeling group locations (Asia, Europe, America) and model structures (vertical coordinates, horizontal grid structures, parameterizations, grid resolutions). Furthermore, the participating groups are not restricted to those formally participating in CMIP6. Considering that CMIP6 does not cover the entire global ocean modeling in the world, it is appropriate to consider participation from a wider group than those directly contributing to CMIP6-OMIP. However, in the statistical treatment of the multi-model ensemble, we acknowledge that the present multi-model dataset is "ensembles of opportunity" (Tebaldi and Knutti, 2007) by following the approach of Wakamatsu et al. (2017). Specifically, we do not use an unbiased estimate of the variance but divide the sum of squares by the number of models. Thus, the model variances and standard deviations presented in the present assessment tend to be underestimated by not including all of the possible model uncertainties. The contribution from CMIP6-OMIP participating

groups will be eventually available from the ESGF, which is summarized in Table A1. All the data used for this study, including data from those not participating in CMIP6, are available along with the scripts used to process the data.

## 3 Spin-up behavior of model simulations

We compare the spin-up behaviors of OMIP-1 and OMIP-2 simulations with a focus on multi-model ensemble means calculated separately for OMIP-1 and OMIP-2. In computing the ensemble means, we use the eight (8) models which performed the full 6-cycle simulations for both OMIP-1 (372 years) and OMIP-2 (366 years) to make a fair comparison. The three models that are not used in the ensemble means either performed 5-cycle for OMIP-1 or used slightly shorter periods (by one to two years) for forcing cycles before the last cycle in OMIP-1 or OMIP-2 (see also Table A1). See Figs. S1 to S9 for the result of individual models, including those that did not perform the full-length simulations.

We start by looking at spin-up behaviors of temperature and salinity fields. Figure 1 shows drifts of annual mean, global mean sea surface temperature and salinity. First, it should be noticed that large ensemble spreads appear from the first year for both sea surface temperature and salinity and similarly for many metrics shown later in this section. The reason for the apparently instantaneous development of the ensemble spread is that the models have somewhat distinct initial conditions. There are many details about model initialization that can create differences across models, most notably the methods each group uses to interpolate/extrapolate WOA to their grid/topography and how they initialize sea ice. In particular, the choices for how the bottom topography is constructed for a given model can result in significant differences in volume average fields. This issue was encountered by the earlier CORE studies such as Griffies et al (2009) and Griffies et al (2014). We continue to perform model initialization using distinct methods across groups for CMIP6-OMIP. This relaxed protocol for initialization is partly because we are not here focused on prediction (an initial value problem) but instead are most concerned with variations and trends after the initial adjustment phase. To clearly show drifts of the multi-model ensemble means, we will show ensemble means of anomalies relative to the mean of the initial year of each model.

The global mean sea surface temperature closely repeats itself between forcing cycles in both OMIP-1 and OMIP-2 simulations. A notable exception appears for the first 5 years of each forcing cycle for the second cycle and beyond, during which the warmed sea surface temperature from the previous cycle is adjusted to the cooler atmospheric environment at the start of the forcing cycle. The patterns of the interannual variability of sea surface temperature exhibit some notable difference between OMIP-1 and OMIP-2, which is discussed in section 4. In contrast to sea surface temperature, ensemble spreads of the model drifts are larger than the internal variability in sea surface salinity, with some models showing drifts even in the last cycle of OMIP-2. It might seem strange for some models to have such long-term drifts of sea surface salinity despite the restoring toward a reference distribution, this is partly due to the salt conservation conditions applied to the salt fluxes due to surface restoring. For example, although a model with a high bias in the globally averaged sea surface salinity will try to remove salt through salinity restoring, the conservation condition will force the globally integrated salt flux to zero, resulting in insufficient removal of salt from the model.

Drifts of annual mean, global mean vertically averaged (potential) temperatures are depicted in Fig. 2 for four depth ranges (0 – 700 m, 0 – 2000 m, 2000 m – bottom, 0 m – bottom), with Table D1 listing deviations of 1980−2009 mean temperatures of the last cycle relative to the initial year of the integration for all participating models. Note that the depth ranges of 0 – 700 m, 0 – 2000 m are those that many observationally derived estimates use to report long-term variability of vertically averaged temperature. The simulation results are directly compared with those estimates in Section 5. In both
OMIP-1 and OMIP-2, ensemble mean temperatures of the upper layer increase and those of the deep to bottom layer decrease relative to the initial year. Because of the compensation between the upper and the lower layers, the temperature averaged over all depths only slightly decreases. Note that these features do not necessarily explain the behavior of individual models, as indicated by the large model spread. Indeed, there are models with increasing and decreasing temperatures even in the last cycle, with trends largely determined by the deep to bottom layers. The model spread keeps
increasing in the deep to bottom layer (2000 m – bottom). On the other hand, for the upper layer (0 – 700 m), the drifts become small and the model spread even decreases after approximately the third cycle in OMIP-1 and the fourth cycle in OMIP-2, with OMIP-2 giving larger model spreads than OMIP-1. OMIP-2 simulations give higher temperature than OMIP-1 in the upper layer. Appendix B1 discusses the results of using common periods (1958−2009) for forcing OMIP-1 and OMIP-2 to understand whether the difference in the forcing periods between OMIP-1 and OMIP-2 simulations has any
implications for this difference in the heat uptake. As shown there, the difference between the forcing datasets during the common period (1958-2009) can largely determine the difference in the heat uptake by the upper ocean between OMIP-1 and OMIP-2 simulations. In other words, the difference in the heat uptake between OMIP-1 and OMIP-2 simulations does not result from the difference in the forcing periods. This implies that we should focus more on structural differences such as ventilation and subduction in considering the more upper layer warming in OMIP-2. For example, the temperature in the
thermocline depths in the OMIP-2 simulations are higher in the mid to low latitude South Atlantic and Pacific Oceans (Fig. 13e). In the mid-latitude region of the southern hemisphere where these thermocline waters contact the sea surface, the sea surface temperatures are generally higher in OMIP-2 (Fig. 6e).

    Drifts of globally averaged horizontal mean temperature and salinity as a function of depth are useful metrics to assess model spin-up. Figure 3 presents these drifts along with the time evolutions of their model spreads. Temperature drifts are
large for the subsurface and bottom depths in both OMIP-1 and OMIP-2, with OMIP-1 simulations showing relatively smaller drift. The model spread (one standard deviation) in the bottom layer is more than 0.5°C in the last cycle, which is greater than the mean value, implying that the response of the deep to bottom layer of an individual model strongly depends on its own model settings rather than the surface forcing dataset used to force the model. Salinity drifts in OMIP-1 and OMIP-2 show similar behaviors except for the contrasting behavior in the 100 – 500 m depths with very weak drift in
OMIP-1 and persistent salinification in OMIP-2 for many models, which is presumably due to the higher sea surface salinity in the mid-latitude southern hemisphere for OMIP-2 simulations (see also Figs. 7 and 14). Note that the model spreads for both temperature and salinity in the 1000 – 4000 m depths are relatively small, but they keep increasing until the last cycle.

This behavior indicates that these depths are where the long-term thermohaline adjustment takes place and requires much longer integrations to reach a steady state.

Long-term drift of sea-ice is also a useful metric to assess steadiness of the simulated ocean–sea-ice system. Figure 4 shows the drift of ensemble mean sea-ice volume integrated over each hemisphere. Notable drifts are not seen after the second cycle in the ensemble means. Also, the model spread does not show large variation, indicating that individual models do not have major drift or collapse of the sea-ice distribution (e.g., formation of open ocean polynyas) by the end of the spin-up. The ranges of model spreads are very wide, with ratios of the maximum to the minimum reach a factor of two to three,

although these ranges may change slightly when we compare total sea-ice masses, which are obtained by multiplying sea-ice density defined by each model to sea-ice volumes. Note that OMIP-2 simulations have larger sea-ice volume than OMIP-1 simulations in both hemispheres.

In contrast to heat content, the total salt content in the ocean–sea-ice system is essentially constant in nature. In most participating models, the global salt content in the ocean–sea-ice system is explicitly conserved, which is achieved by

removing the globally integrated salt flux arising from salinity restoring at each time step (salinity normalization) as noted earlier. The same adjustment is applied to surface freshwater flux in most participating models, resulting in conservation of total mass of water in the ocean–sea-ice system. Thus, in such models, variation of global mean salinity only occurs due to variation of sea-ice volume and the global mean salinity would not be normally employed as a metric for the purpose of model intercomparison. Figure 4 implies that global mean salinity increases for the first 10 to 15 years of each forcing cycle

and then decreases for the rest of the cycle in both OMIP-1 and OMIP-2 simulations. It also implies that a long-term drift of global mean salinity does not occur in those models that have applied both salinity and freshwater normalization.

Figure 5 shows the time series for key circulation metrics, with Table D2 listing 1980−2009 means of the last cycle for all participating models. The Atlantic meridional overturning circulation (AMOC) at 26.5°N (defined as the vertical maximum of the streamfunction, Fig. 5a-c), which approximately represents the strength of AMOC associated with the North Atlantic

Deep Water formation, shows little drift between cycles in OMIP-1 while it declines in the first cycle and slowly recovers thereafter in OMIP-2. These contrasting behaviors are more clearly recognized by comparing plots for all participating models of OMIP-1 and OMIP-2 (Fig. 5a and 5b, respectively). This initial decline of AMOC in many OMIP-2 simulations is at least partly caused by the larger amount of the mean fresh water discharge from Greenland in the OMIP-2 than the OMIP-1 dataset as described by Tsujino et al. (2018) (See their Fig. 20). This behavior necessitates the 6-cycle protocol for OMIP-2,

which makes the period from 4th to 6th cycles suitable for studying the ocean uptake and spread of anthropogenic greenhouse gasses (1850 to present) in OMIP-2. Drake Passage transport (Fig. 5d-f; positive transport eastward), which measures the strength of the Antarctic Circumpolar Current, shows quite similar behavior between OMIP-1 and OMIP-2 in terms of spin-up and strength, although the model spread is quite large. Drifts become small approximately after the fourth cycle. The same is true for Indonesian Throughflow (Fig. 5g-i; negative transport into the Indian Ocean), which measures

water exchange between the Pacific and Indian Ocean. The long-term drift seen in the first few cycles implies that the Indonesian Throughflow, largely constrained by the topography and wind forcing, is also affected by the long-term

thermohaline adjustment of the Indian and Pacific Oceans (e.g., Sasaki et al., 2018). Global meridional overturning circulation (GMOC) minimum between 2000 m and the bottom at 30°S (Fig. 5j-l), which represents the strength of deep GMOC associated with the Antarctic Bottom Water and Lower Circumpolar Deep Water formation, shows a decreasing trend in the first few cycles, but becomes persistent between forcing cycles after approximately the third cycle. The deep GMOC is slightly stronger in OMIP-2 simulations than OMIP-1 simulations, partly explaining the stronger cooling between 2000 m and the bottom in OMIP-2 simulations (Fig. 2i).

## 3.1 Summary of spin-up behaviors

To summarize the spin-up behaviors, OMIP-1 simulations take about three cycles to spin-up, while OMIP-2 simulations take about four cycles. This behavior motivates the 6-cycle integration for OMIP-2 simulations. Regarding OMIP-1, the 5th and 6th cycles show no major difference in the circulation metrics considered in this section except for the deep to bottom layer temperature and salinity. This fact justifies the inclusion of 5-cycle OMIP-1 simulations to the intercomparison of the "last cycle" as an evaluation of the contemporary climate of individual models as part of the remainder of our assessment.

The overall features of the simulated fields are quite similar between OMIP-1 and OMIP-2, except for some minor differences. Long-term drifts remain in the deep to bottom layer temperature and salinity even in the last cycle of simulations. The deep ocean data from these simulations should be used with care as discussed by Doney et al. (2007). OMIP-2 simulations slightly deteriorate relative to OMIP-1 simulations in some metrics (e.g., warmer upper layer and initial decline of AMOC) and give larger model spreads in temperature and salinity. We expect simulation results to improve as experiences with the OMIP-2 dataset, including refinements to the model configurations, are accumulated and shared among the modeling groups.

## 4 Evaluation of contemporary climate of the last forcing cycle

We compare the contemporary climate of OMIP-1 and OMIP-2 simulations by focusing on the behavior of the multi-model ensemble mean. Here we use the last cycle of all eleven (11) participating models. These include simulations that performed OMIP-1 for 5 cycles and simulations that used slightly shorter periods (by one to two years) for forcing cycles before the last cycle. As shown in the previous section and Appendix B1, for OMIP-1 simulations, the 5th and 6th cycles show no major differences in most metrics except for the deep layer temperature and salinity. Also, a minor difference in the total spin-up period does not result in a major difference in the contemporary climate of the last cycle.

Let us start by looking at sea surface temperature and salinity. Figures 6 and 7 show the ensemble mean bias, ensemble standard deviation, and difference between OMIP-1 and OMIP-2 simulations for the sea surface temperature and salinity, respectively, with Table D3 listing the root-mean-square bias and mean bias of the long-term average (1980−2009) of all participating models. The overall bias patterns of sea surface temperature are similar between OMIP-1 and OMIP-2, with the magnitude of the biases less than 0.4°C in most regions and with root-mean-square error of OMIP-2 reduced from OMIP-1

by about 6%. However, the modeled confidence range given by twice the ensemble standard deviation is greater than the root-mean-square bias, with the observations captured by the modeled confidence range in more than 85% of the region. The

same is true for salinity, with the magnitude of the biases less than 0.4 practical salinity units (psu) in most regions. Note that the bias of OMIP-2 may have been underestimated relative to OMIP-1 because the salinity to which sea surface salinity is restored in OMIP-2 is based on WOA13v2, which is also used as the reference dataset for the evaluation. The ensemble spreads capture the observations in more than 90% of the region. Note that the multi-model ensemble mean gives root-mean-square errors smaller than any individual models in both OMIP-1 and OMIP-2 simulations as shown in Table D3 and Figs.

S10 and S11, a feature already reported from the early stage of the climate model intercomparison activities (e.g., Lambert and Boer, 2001). It is also the case for sea surface salinity (Figs. S13 and S14) and sea surface height (Figs. S24 and S25), except for sea surface height of GFDL-MOM, which performs better than the ensemble mean. Looking regionally, the warm biases and the high salinity biases around the Eastern boundary upwelling region in the Pacific basin, specifically off California and Chile, seen in OMIP-1, are reduced in OMIP-2. It is also the case for the Eastern boundary region in the

Atlantic basin, but the warm bias is somewhat exacerbated offshore in OMIP-2. The biases related to strong oceanic currents such as western boundary currents, Antarctic Circumpolar Current, and Agulhas Current are common between OMIP-1 and OMIP-2. These biases are presumably caused by the relatively coarse horizontal resolution of the models, leading to poor reproducibility of the speed and locations of those currents and the resulting change of material distributions. In a companion paper (Chassignet et al., 2020), we will see how refined horizontal resolution is able to reduce these biases. The ensemble

spread is large in the strong current regions, which are also the region with a large horizontal sea surface temperature gradient (a.k.a. fronts). The spread is also large in the marginal sea-ice zones.

Salinity tends to be higher in the southern hemisphere in OMIP-2, which results in either a reduction or increase of biases depending on locations. Both OMIP-1 and OMIP-2 simulations show high salinity bias in the Arctic Ocean, with some reduction implied for OMIP-2 simulations. The reduction of high salinity bias in the Arctic Ocean in OMIP-2 is partly

explained by the difference in salinity to which sea surface salinity is restored between OMIP-2 (WOA13v2) and OMIP-1 (PHC; Steele et al., 2001) as shown in Fig. 7f. Note that the Arctic Ocean has shown a strong freshening trend over recent decades (Rabe et al., 2014; Wang et al., 2019), thus restoring sea surface salinity to the climatology in the models may result in high salinity biases in recent years. The model spread of salinity is large in the Arctic Ocean, where the diversity among models in the sea ice processes, the surface vertical mixing processes, and the treatment of salinity restoring can lead to large

difference in sea surface salinity. The model spread is also large in the region around the mouths of large rivers such as the Amazon, Yangtze, and Ganges, indicating that the ways the fresh water from rivers is distributed in the models are quite diverse.

How do these bias patterns found after a long-term model integration for sea surface temperature and salinity appear in the initial years of the integration? Figure 8 compares biases for the initial 5-year mean and the long-term mean of the last cycle

from the OMIP-2 simulation of MRI.COM. Some notable biases of sea surface temperature such as the warm bias in the eastern boundary of the South Atlantic and the cold bias in the mid-latitude western North Pacific are already found in the

initial years. When the salinity in the later years is subtracted by its global mean, overall spatial patterns of salinity bias are similar between the initial years and the later years. (Note that the global mean sea surface salinity of MRI.COM is gradually increasing throughout the integration as shown in Fig. 1g). This behavior may not necessarily apply to other metrics, but these results for sea surface temperature and salinity indicate that a short-term integration can be useful for detecting and attributing causes of some biases.

Sea-ice is also an important metric since it comprises the boundary condition for other components of the earth system models, with Fig. 9 presenting an assessment of sea-ice distribution. In northern hemisphere winter (top panels), both OMIP-1 and OMIP-2 reproduce the observed distribution of sea-ice concentration reasonably well. But the sea-ice covers a wider area than the observation in the Greenland-Iceland-Norwegian Seas. In northern hemisphere summer (second row), OMIP-1 clearly underestimates sea-ice concentration, which is improved in OMIP-2, although the sea-ice extent is similar for the two simulations. In the southern hemisphere, again, both OMIP-1 and OMIP-2 reproduce the observed distribution reasonably well in winter (third row), with OMIP-2 generally giving a smaller sea-ice extent than OMIP-1. In summer (bottom row), OMIP-2 reduces the low concentration bias in OMIP-1, thus giving a more realistic sea-ice extent in OMIP-2.

The sea surface height, or ocean dynamic sea level, represents dynamical properties of the ocean, with its horizontal gradient balancing the geostrophic current near the sea surface. Figure 10 presents an assessment of sea surface height, with Table D3 listing the root-mean-square bias of the 1993−2009 mean sea surface height for all participating models. Note that Appendix C details the preprocessing necessary to compare sea surface heights from observation and simulations. The overall bias patterns are quite similar between OMIP-1 and OMIP-2 except for the north equatorial Pacific Ocean. A zonally elongated pattern of positive bias occurs from the western to central basin in OMIP-1 and from the central to eastern basin in OMIP-2. Both OMIP-1 and OMIP-2 ensemble spreads fail to capture the observation there (Figs. 10c and 10d). The issue is related to the wind stress field around the Intertropical Convergence Zone, which will be further discussed when exploring the North Equatorial Counter Current later in this section (see Fig. 18). The positive anomaly in the northern North Pacific of OMIP-2 relative to OMIP-1 is presumably due to the known weaker wind stress in OMIP-2 relative to OMIP-1 (e.g., Taboada et al., 2019), which will be discussed in relation to meridional overturning circulations and northward heat transports later in this section (see Figs. 15 through 17). The zonally elongated pattern of negative and positive biases found along the Kuroshio Extension to the east of Japan is presumably due to the lack of twin recirculation gyres along the Kuroshio Extension in low resolution models (e.g., Qiu et al., 2008; Nakano et al., 2008). The negative bias found along the Gulf Stream extension implies the failure of the models to reproduce the Gulf Stream penetration and associated recirculation gyres. The reason for that failure would not be simple because the western boundary current, the deep water formation, and the bottom topography interact to form the mean state, with very fine (~ 1/50°) horizontal resolution models generally required to reduce the biases (e.g., Chassignet and Xu, 2017). A large difference in sea surface height is found in the eastern Arctic Ocean, with OMIP-2 higher than OMIP-1. This difference is presumably related to the lower upper ocean salinity (and thus less dense water) found in OMIP-2 (Fig. 7e). Note that the inter-model spread is similar between OMIP-1 and OMIP-2, with large spread found in the strong current regions.

Seasonal evolutions of the surface mixed layer depths determine the way the ocean interior is ventilated. The annual maximum and minimum occurring in winter and summer, respectively, are particularly important metrics. Note that the definition for mixed layer depth used in OMIP is explained in Appendix H24 of Griffies et al. (2016). Specifically, mixed layer depth is determined based on the vertical distribution of a buoyancy difference, $\delta B$, computed as

$$\delta B = -g(\rho_{\text{displaced from surface}} - \rho_{\text{local}})/\rho_{\text{local}}, \qquad (1)$$

where

$$\rho_{\text{displaced from surface}} = \rho[S(k=1), \Theta(k=1), p(k)] \quad \text{and} \quad \rho_{\text{local}} = \rho[S(k), \Theta(k), p(k)], \qquad (2)$$

with salinity, temperature, and pressure represented by $S$, $\Theta$, and $p$, respectively. The mixed layer depth is approximated as the first depth from the surface where $\delta B = \Delta B_{\text{crit}} = 0.0003$ m s$^{-2}$ using any kind of interpolation. Note that $\Delta B_{\text{crit}} = 0.0003$ m s$^{-2}$ corresponds to a critical density difference of $\Delta \rho_{\text{crit}} = 0.03$ kg m$^{-3}$, which is adopted by the observational dataset compiled by de Boyer Montégut et al. (2004) used for the present evaluation. Figures 11 and 12 show the biases of the winter and summer mixed layer depth in both hemispheres, respectively, with Table D4 listing the root-mean-square bias and mean bias of the 1980−2009 mean for all participating models. Both OMIP-1 and OMIP-2 biases exhibit similar horizontal distributions with OMIP-2 showing smaller root-mean-square errors. In winter, mixed layer depths of a few hundred meters are formed in the mid-latitude western boundary current extension regions such as the Kuroshio extension and the Gulf Stream extension. Mixed layer depths of more than 1000 meters are formed in the Weddell Sea, the Labrador Sea, and the Greenland-Iceland-Norwegian Seas, where deep and bottom waters are formed in the models. Models tend to show deeper bias in both regions, also exhibiting a large model spread. The mixed layer depth is deeper in the Labrador and Irminger Seas in OMIP-2 than OMIP-1. Around Greenland, the mixed layer is shallower in OMIP-2 than OMIP-1, which is presumably caused by the larger freshwater discharge from Greenland in the OMIP-2 (JRA55-do) dataset. The lower sea surface salinity of OMIP-2 shown in Fig. 7e is also consistent with its shallower mixed layer. The rather deep mixed layer in the modeled Weddell Sea is not found in observations (though observations are rather limited in this region) and may represent an unrealistic formation process of the simulated Antarctic Bottom Water.

In summer, both OMIP-1 and OMIP-2 exhibit biases less than 10 m in most regions implying that the observational estimates are well reproduced. One notable exception is that the summer mixed layer depth in OMIP-2 is deeper by about 10 m around the Antarctic Circumpolar Current region, with the OMIP-2 behavior closer to observational estimates. Model spreads of OMIP-1 and OMIP-2 are also similar.

We will proceed with the evaluation toward the ocean interior. Figures 13 and 14 show the basin-wide zonal mean temperature and salinity, respectively, with Tables D5 and D6 listing the root-mean-square bias of the 1980−2009 mean of temperature and salinity for all participating models. First, it is notable that the bias patterns of OMIP-1 and OMIP-2 are similar. Also note that the biases of temperature and salinity show very similar patterns, thus indicating that they are compensating each other in their effects on density biases (small density biases can be expected). The cold and fresh biases in the 1000 – 2000 m depth range of the northern Indian Ocean and the subsurface South Pacific seen in OMIP-1 are reduced in OMIP-2, while the warm and salty bias in the 2000 – 3000 m depth range and the cold and fresh bias in the bottom of the

Atlantic Ocean in OMIP-1 are slightly exacerbated in OMIP-2. Note that large model spreads are found for the cold and fresh biases in the 1000 – 2000 m depth range of the northern Indian Ocean and the warm and salty bias in the 1000 – 3000 m depth range in the high-latitude North Atlantic Ocean. These are the regions where an exchange of water masses occurs between an oceanic basin and marginal seas through oceanic sills (between the Indian Ocean and Red Sea/Persian Gulf and between the Atlantic Ocean and Greenland-Iceland-Norwegian Seas). Models show diverse behaviors according to the

representation of topography and the parameterization of unresolved mixing and transport. Bottom water temperature shows a model spread (~ 0.5 – 1°C) larger than the difference between OMIP-1 and OMIP-2 in all basins (~ 0.1 °C). The model spread for bottom water salinity shows different patterns than those of temperature, but the model spread for bottom water salinity is larger than the difference of salinity between OMIP-1 and OMIP-2 in all basins.

The basin-wide averaged material distributions and thus important climate metrics such as the meridional heat transports

are largely determined by the meridional overturning circulations, with Fig. 15 showing the stream functions of basin-wide meridional overturning circulations. The difference between OMIP-1 and OMIP-2 is less than 1 Sv (1 Sv = $10^6$ m$^3$ s$^{-1}$) in most regions. The subtropical cells in the upper layer of the Indo-Pacific sector and the clockwise cell in the Southern Ocean sector are weaker in OMIP-2, which is presumably due to the known weaker wind stress in OMIP-2 relative to OMIP-1 (e.g., Taboada et al., 2019). The upper anticlockwise cell in the mid- to high-latitude North Pacific sector is also weaker in OMIP-

2. Figure 16 shows the multi-model mean, basin-wide averaged zonal wind stress for OMIP-1 and OMIP-2. The zonal wind stress of OMIP-2 is weaker than OMIP-1, but OMIP-2 is closer to observational estimates. This difference is due to the difference in the treatment of equivalent neutral wind between the OMIP-1 and OMIP-2 datasets as explained by Tsujino et al. (2018). The model spreads of meridional overturning circulations (Figs. 15c and 15d) are large in the maximum and minimum of major meridional overturning circulation cells that represent the thermohaline circulations, whereas the model

spreads are relatively small in the upper few hundred meters presumably because the upper ocean meridional overturning circulation cells are dynamically constrained by the surface wind stress. Note that the large model spreads near the surface in the Southern Ocean (north of ~60°S) and over the tropical cells in the Indo-Pacific Ocean are likely due to differences in the implementation and the parameters for the eddy induced transport parameterizations in models, with details given in Appendix A and references therein.

The northward heat transports are assessed by Fig. 17. Although both OMIP-1 and OMIP-2 are largely within the uncertainty range of observational estimates, northward heat transport in the Atlantic Ocean is significantly smaller than the observational estimates at 26.5°N in both cases and OMIP-2 is smaller than OMIP-1 almost everywhere. Note that a recent estimate by Trenberth and Fasullo (2017) gives around $1.0 \pm 0.1$ PW for the peak value of the North Atlantic, which overlaps better with the OMIP-1 and OMIP-2 envelope. The difference between OMIP-1 and OMIP-2 simulations is qualitatively

consistent with the implied northward heat transport of OMIP-1 and OMIP-2 forcing datasets (Tsujino et al., 2018). The difference is presumably attributed to the known weaker wind speed of OMIP-2 (e.g., Taboada et al., 2019) as explained earlier in this section. The cooling near the surface in the tropical North Pacific Ocean and warming below in OMIP-2

relative to OMIP-1 for the zonally averaged temperatures as shown in Fig. 13e further weakens the northward heat transport in the North Pacific in OMIP-2, though it is notable that these changes reduce the temperature biases in OMIP-2.

In the tropical Pacific Ocean, mean surface and subsurface zonal currents can reach more than several tens of cm s$^{-1}$ (Johnson et al. 2002) and thus they can have non-trivial impact on material circulations and distributions in this region. In particular, the collective effect of the climatological currents on the advection of anomalous temperature is to damp growth of El Ninõ and Southern Oscillation (ENSO) (Jin et al., 2006; Kim and Jin, 2011) and the mean currents are thought be important to characterize the representation of ENSO in coupled models (Bellenger et al., 2014). Figure 18 shows the zonal
velocity across a latitude-depth section along 140°W of the eastern tropical Pacific Ocean. The eastward Equatorial Undercurrent around 100 m depth and the westward South Equatorial Current at the surface are reproduced well in both simulations. However, as reported by Tseng et al. (2016), the surface eastward current of the North Equatorial Counter Current at 6–8°N is weak in OMIP-1 simulations. This bias has been improved only slightly in OMIP-2 simulations. The reason for this bias is presumably related to the method used to adjust the wind vector in both OMIP-1 (CORE2) and OMIP-
2 (JRA55-do) forcing fields as noted by Sun et al. (2019). The weak wind variabilities in the Intertropical Convergence Zone (ITCZ) in the original reanalysis products have been adjusted by increasing the wind speed in both forcing datasets (See Figure 10 of Tsujino et al. 2018). This wind speed increase results in the erroneous strengthening of the weaker mean easterly wind along the ITCZ relative to its surroundings, which was reproduced rather realistically in the original JRA-55 reanalysis. The result after the adjustment is a shallowing of the minimum of the mean easterly winds along the ITCZ and a
weakening of the wind stress curl both north and south of the ITCZ, leading to a weakening of the eastward North Equatorial Counter Current and bias in the sea surface height shown in Fig. 10. Note also that the strengthening of the easterly wind over the surface eastward current of the North Equatorial Counter Current results in the weakening of the eastward current in the simulations because the wind stress further weakens the current as shown by Yu et al. (2000). As a final note, the majority of participating models with horizontal resolution around 1° fail to reproduce the subsurface eastward currents in
the 200 – 300 m depth range both north and south of the Equator (a.k.a. Tsuchiya-jets; Tsuchiya, 1972, 1975). Ishida et al. (2005) demonstrated that a model with 1/4° horizontal resolution can reproduce Tsuchiya-jets. Indeed, the models with higher horizontal resolutions (GFDL-MOM with 1/4° and Kiel-NEMO with 1/2°) reproduce these subsurface jets (Figs. S44 and S45).

## 4.1 Summary of contemporary ocean climate

The overall features of the mean state are quite similar between OMIP-1 and OMIP-2 except for some minor differences. Root-mean-square errors are reduced in sea surface temperature and sea surface salinity in moving to OMIP-2. The positive bias of sea surface temperature and salinity off the western coast of North and South America and South Africa in OMIP-1 is reduced in OMIP-2, while Sea surface temperature further offshore of South Africa is slightly deteriorated in OMIP-2. Summer sea-ice distributions in both hemispheres are improved in OMIP-2. Northward heat transport in OMIP-2 is weaker
than OMIP-1, presumably caused by the weaker meridional overturning circulations (AMOC and the North Pacific

Subtropical Cell) in OMIP-2. The weaker North Pacific Subtropical Cell in OMIP-2 is directly related to the weaker zonal wind stress in OMIP-2, although the zonal wind stress of OMIP-2 is closer to observations than that of OMIP-1. The eastward current of the North Equatorial Counter Current is slightly improved in OMIP-2, but it still has a weak bias.

**5 Interannual variability of the last forcing cycle**

We assess interannual variability of key ocean-climate indices in the last forcing cycle. All participating models are included in the ensemble mean. The horizontal distributions of the reproducibility of seasonal and interannual variability for sea surface temperature and sea surface height and seasonal variability for mixed layer depth are presented in Appendix E.

The annual mean Atlantic meridional overturning circulation (AMOC) maximum at 26.5°N is shown in Fig. 19. The ensemble means of OMIP-1 and OMIP-2 show very similar behavior in the common period (1958–2009); an increasing

tendency toward the mid-1990s and a decreasing tendency thereafter as was demonstrated for CORE (predecessor to OMIP-1) simulations by Danabasoglu et al. (2016), with this behavior also inferred from observations (e.g., Robson et al., 2014). However, the AMOC strength under both OMIP-1 and OMIP-2 is smaller than the estimate based on RAPID observations (e.g., Smeed et al., 2019). In OMIP-2, the AMOC keeps declining in recent years contrary to observations. The observed increasing trend after 2010 has not been reported in the literature and the reason has not yet been clarified. An internal

assessment conducted by the development group of the forcing dataset and protocols suggested that the recent increase in the runoff from Greenland as reported by Bamber et al. (2018) does not have a major impact on the simulated decline in AMOC in OMIP-2. This is a subject warranting further research.

The annual mean Drake Passage transport (positive transport eastward), which measures the strength of Antarctic Circumpolar Current, is shown in Fig. 20. An increasing trend is found for OMIP-1 after the 1970s while this trend is far less

in OMIP-2, which is presumably due to difference in the trends of the imposed westerly winds (not shown). In OMIP-2, the models with small Drake Passage transport (AWI-FESOM, Kiel-NEMO, MIROC-COCO4.9, and CAS-LICOM3) are presumably related to the low density of the simulated Antarctic Bottom Water around Antarctica. This feature is reflected in the fact that these four models have the weaker deep to bottom layer cell of the global meridional overturning circulation stream-function (< 10 Sv in the last cycle) as shown in Fig. 5k and Table D2. The multi-model ensemble means of both

OMIP-1 and OMIP-2 are in the range of observational estimates.

The annual mean, globally averaged sea surface temperature (SST) is shown in Fig. 21. Consistent with the findings of Griffies et al. (2014), OMIP-1 simulations do not show the warming trend in the 1980s and 1990s due to the rapid warming during the latter half of the 1970s. This is consistent with the excessive warming seen from the mid-1970s to the mid-1980s in the surface heat flux diagnosed using the OMIP-1 (CORE) dataset and observationally derived SST datasets as shown in

Figure 22e of Tsujino et al. (2018). As a result, a slowdown of global surface warming persists from the 1980s to 2000s, while the observed global surface warming slowdown occurs only during the 2000s. In contrast, OMIP-2 simulations closely follow the interannual variability and the trend of observed SST. OMIP-2 simulations also reproduce the rapid SST rise

observed after 2015. This behaviour is a clear improvement that further motivates analyses of OMIP-2 simulations in terms of ocean climate variability and trends.

The sea-ice extent in the northern and southern hemispheres is shown in Fig. 22, with Table D7 listing the 1980-2009 mean sea-ice extent for all participating models. OMIP-1 simulations, in general, show small sea-ice extent in the summer of both hemispheres, compared to a satellite-derived sea-ice extent. This bias is reduced in OMIP-2 simulations, although the summer sea-ice extent is still smaller than observations in the southern hemisphere. The overall reduction of the mean bias in the southern hemisphere in OMIP-2 in both seasons is due to the improvement of outliers. It is also notable that the year-to-

year variability of the multi-model ensemble mean is much improved in the southern hemisphere in OMIP-2. This finding is reflected in the performance of individual models as shown in the Taylor diagrams (Fig. 23). The improvement in OMIP-2, represented by the increased correlation coefficients and reduced distance from observations, found in the southern hemisphere winter (Fig. 23d) and the northern hemisphere summer (Fig. 23b) is particularly striking. Note that the models showing large standard deviations in the northern hemisphere summer in their OMIP-1 simulations (CAS-LICOM3, CESM-

POP, CMCC-NEMO, FSU-HYCOM, NorESM-BLOM) is using either CICE4 (Hunke and Lipscomb, 2010) or CICE5.1.2 (Hunke et al., 2015) as their sea-ice model.

    Globally integrated ocean heat content anomaly in four depth ranges and the thermosteric sea level anomaly are shown in Figs. 24 and 25, respectively, relative to the 2005 – 2009 means. These two diagnostics are almost equivalent, so either one is sufficient for evaluating model performance. Nonetheless, we evaluate both because decomposing the heat content into

several depth ranges renders extra insight into thermosteric sea level changes. For 0 – 700 m and 0 – 2000 m, the ocean heat content anomalies start to follow the observation-based estimates only after the mid-1990s. We suggest that the mismatch between the observed and simulated heat content trajectory is linked to the long ocean memory (Zanna et al., 2019, Gebbie and Huybers, 2019) in comparison to the relatively short duration (or length) of the OMIP forcing datasets. Recent studies have demonstrated that the deep ocean has only recently started warming after a long period of cooling since the medieval

warm period (Gebbie and Huybers, 2019). However, the OMIP-1 and OMIP-2 forcing datasets only extend back to the mid-20[th] century, eventually spinning up the ocean towards a relatively warm state to start the last cycle of the simulation. Therefore, it is only during the 1990's that the simulated ocean heat content matches the observations, after which the models follow the observed trajectory as expected.

    The multi-model mean thermosteric sea level rise after 1992 in OMIP-2 is slower than OMIP-1 and fails to reproduce the

observed rapid rise after 2010. The more rapid decline of ocean heat content anomaly and thermosteric sea level in the year around 1991 in OMIP-1 is presumably due to the representation of the volcanic eruption of Pinatubo, leading to lower downward shortwave radiation. This eruption is absent in the OMIP-2 (JRA55-do) dataset, resulting in stronger cooling by 5 W m$^{-2}$ only in OMIP-1 for the year 1991 according to Tsujino et al. (2018) (see their Figure 22). The decline found in OMIP-2 is due to the low air temperature assimilated in the original JRA-55 analysis product, which turned out to be insufficient to

reproduce the observed cooling in 1991. In a future version of the OMIP-2 dataset, this specific volcanic effect should be included in the downward shortwave radiation.

Large drifts remain below 2000 m in many OMIP-1 and OMIP-2 simulations, which eventually dominate the heat content drift of all layers (and the thermosteric sea level rise). If a linear trend (determined separately for each model) in the last cycle is subtracted from each model, the models show very similar behaviors. This similarity implies that there could be a better method to separate model drifts from internal variabilities, with this question left for future studies.

Overall, the OMIP simulations under the protocol of repeating many cycles of the entire period of the atmospheric forcing dataset do not capture variability of heat content and thermosteric sea level in the entire atmospheric dataset period. Only recent (after 1990s) upper layer heat content variability is reproduced. This limitation should be taken into account in analysing the results of the OMIP simulations. However, we note that the results still represent the redistribution of upper layer water masses due to wind forcing variability. Figure 26 shows the horizontal distribution in the trend of vertically averaged temperature in the upper 700 m depth. This diagnostic is determined by both surface heating and mass redistribution due to wind forcing variability. As reported by Griffies et al. (2014), OMIP-1 simulations fail to reproduce the warming trend off the Philippines. OMIP-2 simulations are successful at reproducing this feature, although the magnitude is smaller than the observational estimates. Other horizontal distributions are largely reproduced well, and notably spurious cooling in the equatorial Pacific and Atlantic Oceans are much reduced in OMIP-2.

**5.1 Summary of interannual variability**

Improvements in moving to OMIP-2 are identified for interannual variability of SST and sea-ice extent. The spatial distribution of the trend of vertically averaged temperature in the upper 700 m depth is also improved. In each forcing cycle, it is only during the most recent 20 years that the warming signal is large enough to emerge from the model's mean state and any inherent model drift/trends. Contrary to observations, AMOC keeps declining in recent years in OMIP-2. The reason for this decline should be investigated in a future study, including the role of increasing runoff from Greenland in the JRA55-do forcing dataset. Overall, except for some minor differences, OMIP-1 and OMIP-2 simulations show similar interannual variability.

**6 Statistical evaluations**

Results of the statistical tests for the difference between OMIP-1 and OMIP-2 simulations are shown in the previous sections for the metrics with two-dimensional distribution (e.g., Fig. 6e). Table 2 lists results of the same test applied to the metrics consisting of time series of index values. The differences due to the change in the forcing datasets are not statistically significant in most regions and time series. This insignificance of the differences is caused by the basic similarity between the two forcing datasets. The large model spread is also contributing to this statistical insignificance.

We also compare model performances in this section. First, we consider ordering among the models in the metrics and how the change in experimental framework (i.e., the forcing dataset) affects the ordering. Specifically, for each metric, a scatter diagram comparing values of the metric computed for the OMIP-1 and OMIP-2 simulations of all models is drawn and the robustness of the relative positions among the models against the change in forcing datasets is tested using linear

fitting. The metrics assessed in the proceeding sections and listed on Tables D1–D8 are considered. Figures 27 and 28 show some examples and Table 3 lists $r^2$-scores of linear fits for all the metrics considered here. Note that $r^2$-score is the square of the correlation coefficient. In the present intercomparison with 11 independent participating models, the correlation coefficient with 1% level of significance is 0.735 ($r^2 \sim 0.54$) for 9 degrees of freedom. Figure 27 shows scatter diagrams with linear fitting and its $r^2$-score for root-mean-square bias and mean bias of SST and SSS (see Table D3 for the specific values). It would be notable that these metrics correlate well between OMIP-1 and OMIP-2. Hence, change in the relative performance among the models is small against the change in forcing datasets for these metrics. Figure 28 shows the similar diagrams for metrics related to large scale circulations (see Table D2 for the specific values). Correlation coefficients are generally low except for the Indonesian Through Flow, which is thought to be determined by the model topography by the first order approximation.

When all metrics listed on Table 3 are taken into consideration, it is found that, among the many metrics whose $r^2$-score exceeds 1% level of significance, particularly high scores ($r^2 > 0.8$) are found for sea surface temperature, sea surface salinity, sea surface height, sea ice extent, mixed layer depth in both winter and summer, zonal mean salinity in the Atlantic Ocean, zonal mean temperature and salinity in the Indian Ocean, and Indonesian Through Flow. These metrics are generally determined by one-to-one relationship between model settings and forcing and do not involve complex adjustment processes (except perhaps for zonal mean salinity in the Atlantic Ocean). On the other hand, $r^2$-scores are low ($r^2 < 0.54$) for some circulation metrics such as AMOC and GMOC (bottom water circulation), ACC, and zonal mean temperature in the Southern Ocean. This result indicates that those metrics that involve complex adjustment processes in models are sensitive to differences in the forcing dataset. Therefore, when a modeling group is not satisfied with the performance of its model in a certain metric in comparison with other models, it might be possible to improve the performance by reviewing its choice of model settings if $r^2$-score of the metric is high. On the other hand, if $r^2$-score of the metric is low, the situation would not be that simple. One will need to look into the subtle difference in the forcing if the model shows different performances between its OMIP-1 and OMIP-2 simulations. However, it would be still useful to review the model setting if both OMIP-1 and OMIP-2 simulations are outliers among the bulk of models.

Appendix E presents a statistical assessment of model performances in reproducing observed seasonal and interannual variability. Both OMIP-1 and OMIP-2 simulations exhibit high performances for seasonal and interannual variability of sea surface temperature, sea surface height, and seasonal variability of mixed layer depth, with the OMIP-2 simulations showing a slight improvement. We find that the assessment of temporal variability should be applied with care for models populated with mesoscale eddies since, for example, reproducibility of temporal variability of sea surface height could be particularly low for such models, thus necessitating a novel method to assess these eddying simulations.

## 7 Summary and conclusion

In this paper, we presented an evaluation of a new framework prepared for the second phase of Ocean Model Intercomparison Project (OMIP-2). The OMIP-2 framework involves an update of the atmospheric forcing dataset for computing boundary fluxes and the protocols for running global ocean–sea-ice models. This new framework aims to replace that of the first phase (OMIP-1) for further advancing ocean modeling activities.

  We compared the two sets of simulations (OMIP-1 and OMIP-2), which differ in datasets and protocols for computing

surface fluxes, conducted by eleven (11) groups, with each group using the identical global ocean–sea-ice model for their respective OMIP-1 and OMIP-2 simulations. Multi-model ensemble means and spreads were calculated separately for the OMIP-1 and OMIP-2 simulations and overall performances were compared in terms of metrics commonly used by ocean modelers. We did not focus on individual model performances in detail nor did we look deeply into specific oceanic processes. We expect that many research activities will follow this benchmark paper to study the specific questions raised by

our results.

  The general performance comparison using the two forcing datasets and protocols for OMIP-1 and OMIP-2, respectively, provides a record of the-state-of-science of global ocean–sea-ice models in the late 2010s and early 2020s. Furthermore, by presenting the general performance of these CMIP-6 class ocean–sea-ice models, we hope to have widened the window for ocean modelers to communicate with the broader Earth System Modelling community, even those not necessarily familiar

with ocean sciences.

  Many simulated features are very similar between OMIP-1 and OMIP-2 simulations. This commonality is not surprising because the OMIP-1 forcing dataset has been produced after very careful considerations among experts under the support of an international group of ocean modelers, and the organization of the OMIP-2 framework basically follows the approach taken by OMIP-1. Many of the model biases that remain common to both sets of simulations may be attributed to errors in

representing and reproducing important processes in ocean–sea-ice models, some of which are expected to be reduced by adopting finer horizontal resolutions. Further common biases can point to limitations in the forcing datasets. One example includes the weak eastward North Equatorial Counter Current arising from the method used to adjust the wind field. Another is the mismatch between the observed and simulated variability of heat content and thermosteric sea level before the 1990s, presumably linked to the long ocean memory in comparison to the relatively short length of the OMIP forcing datasets.

These and other limitations will be addressed in a future version of the JRA55-do dataset.

  Remarkable improvements were identified in the transition from the OMIP-1 to OMIP-2 framework. For example, the sea surface temperature of the OMIP-2 simulations can reproduce the observed global warming of sea surface temperature during the 1980s and 1990s, the slowdown in the 2000s, and the accelerated warming thereafter, particularly through 2018 (Fig. 21). In contrast, these recent events of sea surface temperature variability are not well reproduced in the OMIP-1

simulations partly because OMIP-1 forcing stopped in 2009. In comparison to available observations and to OMIP-1 simulations, additional improvements with OMIP-2 include reduction of the negative bias in the summer sea-ice

concentration of both hemispheres; better interannual variability of sea-ice extent; and better overall reproducibility of both seasonal and interannual variation in sea surface temperature and sea surface height. These represent a new capability of the OMIP-2 framework for evaluating process-level responses using simulation results. Several minor deteriorations were also identified in the transition to OMIP-2. For example, the weaker northward heat transport and AMOC, warmer upper layer, and colder deep/bottom layer. We expect simulation results to improve as experiences with the OMIP-2 dataset, including model development based on JRA55-do simulations, are accumulated and shared among the modeling groups.

The OMIP-2 simulations in 2010s, the period not covered by OMIP-1, show that AMOC keeps declining, which is contrary to observations. Furthermore, the global thermosteric sea level rise is weaker than the observational estimates. The reason for these biases warrants future targeted investigations aiming to understand the mechanisms governing these important ocean climate signals.

Regarding the ordering of performances among models and its sensitivity to the change in the forcing datasets, the models show well-ordered responses for the metrics that are directly forced while they show less-organized responses for those that require complex model adjustments. It is also noted that there is no obvious grouping of models in model skill metrics in terms of model formulation (e.g., the hybrid vertical coordinate models) and model code (e.g., the NEMO models).

To support further studies with OMIP-2, the OMIP framework (forcing dataset and protocol) will be continually reviewed and updated by taking into consideration the present assessment study and feedback from other future studies. Sensitivity experiments using a subset of models presented in Appendix B indicate that the difference in forcing periods by roughly ten years (Appendix B1), the use of an accurate formulae for the properties of moist air (Appendix B2), and the changing contribution of ocean surface currents to the computation of relative winds (Appendix B3), each produces only minor differences to the simulations by an individual model. They are indeed much smaller than the difference between distinct models. We therefore suggest that it is unlikely these details significantly impact any observational comparisons in other CMIP6-class models. In contrast, the changing contribution of ocean surface currents on the computation of relative winds has been reported to impact ocean mesoscale currents in fine resolution models (e.g., Renault et al., 2019a). This issue will therefore become more important as the community refines the grid used in global simulations. The present assessment also indicates that the forcing dataset should be extended back to around 1900 to reproduce longer term trends of heat content and thermosteric sea level in the simulations. Modifications of the OMIP-2 forcing dataset and protocol will be reported when they become available.

Overall, the present assessment justifies our recommendation that future model development and analysis studies use the OMIP-2 framework, particularly considering that the OMIP-2 forcing dataset has higher temporal and spatial resolutions compared to the OMIP-1 dataset and will be updated frequently to keep it current. However, further efforts are warranted to reduce the biases remaining in ocean–sea-ice simulations under the new framework. Some outstanding problems, especially the erroneous representation of deep and bottom water formations and circulations, leading to the large model spread of deep to bottom layer temperature and salinity, can be resolved through strong collaborations between model and forcing dataset developers, process-based researchers, and analysts.

**Appendix A: Contributing models in alphabetical order**

In this appendix, a brief description is given to the model used, and the OMIP-1 and OMIP-2 simulations conducted, by each participating group. The explanations about the simulations will include any deviations from the protocols, the salinity restoring methods, and the treatment of the surface current in computing turbulent surface fluxes in OMIP-2, specifically the value of α in $\Delta\vec{U} = \vec{U_a} - \alpha\vec{U_o}$, where $\vec{U_a}$ is the surface wind vector and $\vec{U_o}$ is the surface oceanic current vector (usually the vector at the first model level). Table A1 summarizes model configurations and experiments of participating groups.

**A1 AWI-FSOM**

Finite Element/volumE Sea-ice Ocean Model (FESOM) is the ocean–sea-ice component of the coupled Alfred Wegener Institute Climate Model (AWI-CM, Sidorenko et al., 2015). It works on unstructured triangular meshes for both the ocean and sea-ice modules (Danilov et al., 2004; Wang et al., 2008; Timmermann et al., 2009). FESOM version 1.4 (Wang et al., 2014; Danilov et al., 2015) is employed in this study and all the CMIP6 simulations as well. A flux-corrected-transport advection scheme is used in tracer equations. The KPP scheme (Large et al., 1994) is used for vertical mixing. The background vertical diffusivity is latitude and depth dependent (Wang et al., 2014). Mesoscale eddies are parameterized by using along-isopycnal mixing (Redi,1982) and Gent-McWilliams advection (Gent and McWilliams, 1990) with vertically varying diffusivity as implemented in Danabasoglu et al. (2008). The eddy parameterization is switched on where the first baroclinic Rossby radius is not resolved by local grid size. In the momentum equation the Smagorinsky (1963) viscosity in a biharmonic form is applied. The sea-ice module employs the Parkinson and Washington (1979) thermodynamics. It includes a prognostic snow layer with the effect of snow to ice conversion accounted. The Semtner (1976) zero-layer approach, assuming linear temperature profiles in both snow and sea-ice, is used in this model version. The elastic-viscous-plastic (EVP, Hunke and Dukowicz, 1997) rheology is used with modifications that improved convergence (Danilov et al., 2015, Wang et al., 2016c).

The horizontal model resolution used in this study is nominal 1° in the bulk of the global domain, with the North Atlantic sub-polar gyre region and Arctic Ocean set to 25 km. Along the equatorial band the resolution is 1/3°. In the vertical 46 z-levels are used, with 10 m layer thicknesses within the upper 100 m depth. The North Pole is displaced over Greenland to avoid singularity. Sea surface salinity is restored to monthly climatology with a piston velocity of 50 m over 900 days inside the Arctic Ocean and three times stronger elsewhere. The two simulations (6 cycles each) are driven with the CORE2 and JRA55-do forcing following the OMIP protocol. The air-sea turbulence fluxes are calculated using the Large and Yeager (2009) bulk formulae. The full ocean surface velocity is used in the calculations (alpha equals one).

**A2 CAS-LICOM**

LICOM (LASG/IAP Climate system Ocean Model) is a global ocean general circulation model developed by LASG, Institute of Atmospheric Physics (IAP), Chinese Academy of Sciences (CAS, Zhang and Liang, 1989; Liu et al., 2004; Liu et

al., 2012; Yu et al., 2018). LICOM is also the ocean component of both Flexible Global Ocean–Atmosphere–Land System model (FGOALS, e.g., Li et al., 2013, Bao et al., 2013) and CAS Earth System Model (CAS-ESM, private communication with Prof. Minghua Zhang). LICOM version 3 (LICOM3) coupled with Community Ice Code version 4 (CICE4) through the NCAR flux coupler 7 (Craig et al., 2012; Lin et al., 2016) are employed for the OMIP-1 and OMIP-2 experiments following the protocols. A restoring term with the piston velocity of 20 m per year has been applied to the virtual salinity flux. The feedback of surface currents to compute the turbulence fluxes is fully applied ($\alpha = 1$). An experiment with $\alpha = 0.7$ is also conducted.

LICOM3 is an ocean model with free sea surface. The primitive equations with Boussinesq and hydrostatic approximations are adopted and solved on the Murray's (1996) tripolar grid with two North "poles" at (65°N, 65°E) and (65°N, 115°W). The horizontal and vertical grid systems are Arakawa B-grid with about 1° grid distant in both longitude and latitude directions, and eta-coordinate (Mesinger and Janjic, 1985) with 30 or 80 levels, respectively. Only the 30-level version, which has 10 m resolution in the upper 150 m, was employed for both OMIP-1 and OMIP-2 runs. The low-resolution LICOM3 had total 360 and 218 number of grids in horizontal. The central difference advection scheme was used in the momentum equations. The Leapfrog with Robert filter is used for the time integration of the momentum equation. The two-step preserved shape advection scheme (Yu, 1994; Xiao, 2006) and the implicit vertical viscosity/diffusivity (Yu et al., 2018) were adopted for the tracer equations.

The vertical viscosity and diffusion coefficients in the mixed layer have been computed by the scheme of Canuto et al. (2001, 2002) with the background values of $2 \times 10^{-6} m^2/s$ and the upper limit of $2 \times 10^{-2} m^2/s$. Recently, a tidal mixing scheme of St. Laurent et al. (2002) has been adopted in LICOM3 by Yu et al. (2017). The Laplacian form with the coefficient of 5400 $m^2 s^{-1}$ are adopted for the horizontal viscosity. The isopycnal tracer diffusion scheme of Redi (1982) and the eddy-induced tracer transport scheme of Gent and McWilliams (1990) with the same coefficients are used to parameterize the effects of mesoscale eddies on the large-scale circulation. Two tapering factors of Large et al. (1997) and a buoyancy frequency ($N^2$) related thickness diffusivity of Ferreira et al. (2005) are also employed. Besides, the chlorophyll-a dependent solar penetration of Ohlmann (2003) was introduced (Lin et al., 2007) for the simulations. The details of the experiments and the preliminary validation can be found in Lin et al. (2020).

**A3 CESM-POP**

The NCAR contribution uses the Parallel Ocean Program version 2 (POP2), a level-coordinate model (Smith et al., 2010) and the sea-ice model version 5.1.2 (CICE5.1.2; Hunke et al., 2015). These models are the ocean and sea-ice components of the Community Earth System Model version 2 (CESM2), and the simulations are performed with this framework. The basic configuration of the model is also used for Stewart et al. (2020) and is described there. The description is briefly summarized here for completeness. POP2 and CICE5.1.2 use the same displaced North Pole grid with a horizontal resolution of nominal 1° with increased meridional resolution of 0.27° near the equator. There are 60 vertical levels in the ocean model,

monotonically increasing from 10 m in the upper ocean to 250 m in the deep ocean. Although the POP2 version used here is similar to the one used in previous CORE studies (Danabasoglu et al. 2014, 2016; see also Danabasoglu et al. 2012 for further details), the present version includes several new features that are briefly summarized in Danabasoglu et al. (2020). Noteworthy updates include a new parameterization for mixing effects in estuaries (Sun et al., 2019); use of salinity dependent freezing-point together with the sea-ice model (Assur, 1958); a new Langmuir mixing parameterization (Li et al.,

2016); and a new time filtering scheme based on an adaption of the Robert filter to enable sub-diurnal coupling of the ocean model (Danabasoglu et al., 2020). Sea surface salinity is restored to monthly WOA13 data with a piston velocity of 50 m over one year. As also summarized in Danabasoglu et al. (2020), CICE5.1.2 incorporates several new features that include a mushy-layer thermodynamics approach (Turner and Hunke, 2015) where the vertical profile of salinity within the ice is prognostic; increased vertical resolution to better resolve the salinity and temperature profiles; and an updated melt pond

parameterization (Hunke et al., 2013) so that ponds preferentially form on undeformed sea-ice.

       The OMIP-1 and OMIP-2 simulations of NCAR are integrated for 372 and 366 years, respectively, which correspond to the 6 cycles of 62- (1948–2009) and 61-year (1958–2018) forcing periods, respectively. We use 1 for $\alpha$ for the momentum flux calculation for both OMIP-1 and OMIP-2.

## A4 CMCC-NEMO

The CMCC contribution uses the ocean and sea-ice components of the coupled CMCC-climate model version 2, CMCC-CM2 (Cherchi et al., 2019). This model system is based on the Community Earth System Model (CESM) version 1.2.2, in which the ocean component is replaced by NEMO-OPA version 3.6 (Madec and the NEMO team, 2016).

       The ocean horizontal mesh is tripolar, based on a 1° Mercator grid, but with additional refinement of the meridional grid to 1/3° near the Equator; the model resolution is about 50 km over the Arctic Ocean. The vertical grid has 50 geopotential

levels, ranging from 1 to 400m.

       A linear free-surface formulation is employed (Roullet and Madec, 2000), where lateral fluxes of volume, tracers and momentum are calculated using fixed reference ocean surface height. Temperature and salinity are advected with the total variance dissipation scheme (Cravatte et al., 2007). An energy and enstrophy conserving scheme (Le Sommer et al., 2009) is used for momentum.

Momentum and tracers are mixed vertically using a turbulent kinetic energy (TKE) scheme (Blanke and Delecluse, 1993) plus parameterizations of Langmuir cell, and surface wave breaking. Lateral diffusivity is parameterized by an iso-neutral Laplacian operator. An additional eddy-induced velocity is also computed with a spatially and temporally varying coefficient. Lateral viscosity uses a space-varying coefficient and is parameterized by a horizontal Laplacian operator with free slip boundary condition. A bottom intensified tidally driven mixing, a diffusive bottom boundary layer scheme, and a nonlinear

bottom friction are applied at the ocean floor.

CMCC-NEMO makes use of the Large and Yeager (2009) bulk formula where the full ocean surface velocity is used ($\alpha=1$) to compute surface wind stresses, in both simulations. Sea surface salinity is restored to monthly climatology provided with the forcing data sets, with a piston velocity of 50m over one year (6 months for OMIP-2), except below sea-ice.

The sea-ice component is based on version 4.1 of Community Ice CodE (CICE) sea-ice model (Hunke and Lipscomb, 2010), which shares the same horizontal grid as NEMO. The CICE model uses a prognostic ice thickness distribution (ITD) with five thickness categories, multi-layer vertical thermodynamics with 4 layers of ice and 1 of snow, elastic–viscous–plastic (EVP) rheology for ice dynamics. Radiative transfer is calculated using the Delta-Eddington multiple scattering radiative transfer model. The OMIP-1 and OMIP-2 simulations of CMCC are spun up for the 6 cycles of 62- (1948–2009) and 61-year (1958–2018) forcing periods, respectively

## A5 EC-Earth3-NEMO

The ocean component of the EC-Earth-NEMO model is the Nucleus for European Modelling of the Ocean (NEMO; Madec et al., 2016). We use the EC-Earth NEMO3.6, revision r9466. NEMO3.6 includes the ocean model OPA (Ocean PArallelise) and the Louvain la Neuve sea-ice model LIM3 (Rousset et al., 2015). OPA is a primitive equation model of ocean circulation, allowing for various choices for the physical subgrid scalar parametrization as well as the numerical algorithms. EC-Earth-NEMO uses the Turbulent Kinetic Energy (TKE) scheme for vertical mixing. The main difference of the OPA-version used in EC-Earth compared to the reference OPA-version of NEMO3.6 is that the parameterization of the penetration of TKE below the mixed layer due to internal and inertial waves is switched off. Other modifications compared to the standard NEMO setup from the ORCA1-shared configuration for NEMO (ShacoNemo), are a slightly enhanced conductivity of snow (rn_cdsn=0.4) on ice and strengthened Langmuir Cell circulation (rn_lc=0.2). EC-Earth-NEMO uses mixed layer eddy parameterization following Fox-Kemper (Fox-Kemper et al., 2008) and a tidal mixing parameterization (Koch-Larrouy et al., 2007). EC-Earth/-NEMO configuration uses ORCA1, a tripolar grid based on the semi-analytical method of Madec and Imbard (1996), with 75 vertical levels and nominal 1° horizontal resolution with reduced resolution around the equator. Salinity restoring is applied evenly throughout the ocean surface (including below sea-ice) with a piston velocity of 50 m over 6 months for both OMIP-1 and OMIP-2. EC-Earth3-NEMO does not take into account ocean surface velocity ($\alpha=0$) to compute surface wind stress.

## A6 FSU-HYCOM

The FSU-HYCOM is a global configuration of the HYbrid Coordinate Ocean Model (HYCOM) (Bleck, 2002; Chassignet et al., 2003; Halliwell, 2004). The grid is a tripolar Arakawa C-grid of 0.72° horizontal resolution with refinement to 0.36° at the equator (500 cells on the zonal direction and 382 in the meridional direction). The bottom topography is derived from the 2-minute NAVO/Naval Research Laboratory DBDB2 global dataset. Forty-one hybrid coordinate layers are used with $\sigma_2$ target densities ranging from 17.00 to 37.42 kg/m³. The vertical discretization combines fixed pressure coordinates in the mixed layer and unstratified regions, isopycnic coordinates in the stratified open ocean, and terrain-following coordinates

over shallow coastal regions. The initial conditions in temperature and salinity are given by the Levitus-PHC2. The ocean model is coupled with the sea-ice model CICE (Hunke and Lipscomb, 2010) that provides the ocean-ice fluxes.

Turbulent air-sea fluxes are computed using the Large and Yeager (2004) bulk formulation except for the surface wind-stress that is calculated *with* the surface currents when forced with CORE2 and *without* surface currents when forced with JRA55-do. No restoration is applied on the sea surface temperature. A surface salinity restoration is applied over the entire domain with a salinity piston velocity of 50m/4 years everywhere, except for the Arctic and Antarctic region where the surface salinity relaxation is set up at 50m/1 year and 50m/6 months, respectively. In addition, a global normalization is
applied to the salinity flux at each time step.

Vertical mixing is provided by the KPP scheme (Large et al., 1994) with a background diffusivity of $10^{-5}$ $m^2$/s and tracers are advected using a second-order flux corrected transport scheme. A Laplacian diffusion of (0.03 m/s) *$\Delta x$ is applied on temperature and salinity and a combination of Laplacian [(0.03 m/s) *$\Delta x$] and biharmonic [(0.05 m/s) *$\Delta x^3$] dissipation is applied on the velocities. The model baroclinic and barotropic time steps are 1800s (leap-frog) and 56.25s (explicit)
respectively. Interface height smoothing (corresponding to Gent and McWilliams (1990)) is applied through a biharmonic operator, with a mixing coefficient determined by the grid spacing $\Delta x$ (in m) times a velocity scale of 0.02 m $s^{-1}$ everywhere except in the North Pacific and North Atlantic where a Laplacian operator with a velocity scale of 0.01 m $s^{-1}$ is used. For regions where the FSU-HYCOM has coordinate surfaces aligned with constant pressure (mostly in the upper ocean mixed layer), Gent and McWilliams (1990) is not implemented, and lateral diffusion is oriented along pressure surfaces rather than
rotated to neutral directions. No parameterization has been implemented for the overflows.

## A7 GFDL-MOM

The GFDL contribution uses the OM4 configuration (Adcroft et al., 2019) of the MOM6 ocean code coupled to the SIS2 sea-ice code. OM4 uses a C-grid stencil configured at nominally 1/4° resolution with a tripolar (Murray, 1996) grid. MOM6 makes use of a vertical Lagrangian-remapping algorithm for the vertical (Bleck, 2002) with OM4 configured with a hybrid
depth-isopycnal (potential density referenced to 2000dbar) coordinate. OM4 is the ocean–sea-ice component of GFDL's coupled climate model CM4 (Held et al., 2019).

For use in OMIP-1 and OMIP-2, OM4 makes use of the Large and Yeager (2009) bulk formula with α=1 to compute surface wind stresses (as in the coupled climate model CM4). Sea surface salinity is restored using a piston velocity of 50m/300days, which is the same value as used by GFDL-MOM5 in the CORE simulations (e.g., Griffies et al., 2009,
Danabasoglu et al., 2014).

## A8 Kiel-NEMO

The Kiel-NEMO configurations have been developed within the DRAKKAR collaboration based on the NEMO (Nucleus for European Modelling of the Ocean) code version 3.6 (Madec et al., 2016). The ocean component of NEMO is based on

Océan Parallélisé (OPA; Madec et al., 1998). The Louvain-la-Neuve Ice Model (LIM2; Fichefet and Morales Maqueda,
1997; Vancoppenolle et al., 2009) sea-ice model is used. A configuration with a global, orthogonal, curvilinear, tripolar,
Arakawa-C type grid with 0.5° horizontal resolution (ORCA05) is used. The vertical grid consists of 46 levels with 6 m
thickness of the surface grid cell, increasing to a maximum of 250 m at depth and a partial-cell formulation at the bottom
(Barnier et al., 2006).

The Total Variance Dissipation (TVD) scheme (Zalesak, 1979) is used for the advection of tracers, whereas an energy and
enstrophy conserving second-order centered scheme adapted from Arakawa and Hsu (1990), modified to suppress
Symmetric Instability of the Computational Kind (Ducousso et al., 2017), is used for the advection of momentum.
Horizontal diffusion is bi-laplacian for momentum (with a background viscosity parameter of $-6.0 \times 10^{11}$ m$^4$/s$^2$ at the equator,
decreasing dependent on latitude) and laplacian for tracers (background diffusivity parameter of 600 m$^2$/s). The scheme of
Gent and McWilliams (1990) is used to parameterize tracer transport by mesoscale eddies and a TKE turbulent closure
scheme is used for the vertical diffusion (Gaspar et al., 1990; Blanke and Delecluse, 1993).

The initialization and atmospheric forcing follow the protocols for OMIP-1 and OMIP-2, and the bulk formulations
proposed by Large and Yeager (2004) are used to calculate the atmosphere-ocean fluxes for both OMIP-1 and OMIP-2. The
surface velocity of the ocean is fully taken into account when computing the momentum fluxes ($\alpha = 1$, relative winds). Sea
Surface Salinity (SSS) is restored toward monthly WOA13 data with -137 mm/day, corresponding to a relaxation time scale
of one year over a 50 m surface layer. No SSS restoring is applied under sea-ice and in grid cells in which runoff enters the
ocean.

A set of scripts used to prepare the input and output together with the model configurations (model reference, code
modifications, and namelists) is available from https://git.geomar.de/cmip6-omip.

**A9 MIROC-COCO4.9**

The MIROC group contribution uses COCO4.9, which is the sea-ice–ocean component of MIROC6 (Model for
Interdisciplinary Research on Climate version 6; Tatebe et al., 2019). The oceanic part is based on the primitive equations
under the hydrostatic and Boussinesq approximations with the explicit free surface. The tripolar coordinate system of
Murray (1996) is used as the horizontal coordinate system, with two singular points in the bipolar region placed at around
63°N. The longitudinal and latitudinal grid spacing in the geographical coordinate region is 1 and 0.5–1 degree, respectively.
There are 62 vertical levels, 31 of which are within the upper 500 m, in a hybrid σ-z vertical coordinate system. The sea-ice
part shares the horizontal coordinate system of the oceanic part and uses a subgrid-scale sea-ice thickness distribution
following Bitz et al. (2001) with five thickness categories.

The oceanic component employs a second-order moment tracer advection (Prather, 1986), a surface mixed layer
parameterization (Noh and Kim, 1999), an oceanic thickness diffusion (Gent et al., 1995), and a bottom boundary layer
parameterization (Nakano and Suginohara, 2002). As the background vertical diffusivity, type III profile of Tsujino et al.
(2000) is used, but the smaller coefficient is given in the uppermost 50 m in order to improve the surface stratification in the

Arctic Ocean (Komuro, 2014). The sea-ice component uses elastic-viscous-plastic rheology (Hunke and Dukowicz, 1997) for solving sea-ice dynamics. In calculating stress between sea-ice and ocean, the surface (1st level) oceanic current is referred to with ice-ocean turning angle of 0°; this treatment is different from that in MIROC6 (11th level and 25°). Albedo on sea-ice varies from 0.8 to 0.68 depending on sea-ice surface condition. Sea surface salinity is restored to the climatology provided with the forcing datasets. The restoring time scale is one year for 50 m except to the south of 60°S, where the time scale is 60 days, with a buffer zone between 50°S and 60°S. The wind speed correction coefficient α is set to 1 in both the OMIP-1 and OMIP-2 simulations.

Simulation lengths for OMIP-1 and OMIP-2 are 310 years (5 cycles) and 366 years (6 cycles), respectively. During the spin-up, the full length of the forcing is used in both the simulations. Note that a 1-2-1 horizontal filter has been applied to the raw SSH output, which includes 2-grid noise arising from the specification of COCO.

### A10 MRI.COM

The JMA-MRI contribution uses the MRI Community Ocean Model version 4 (MRI.COMv4; Tsujino et al., 2017). MRI.COMv4 is a free-surface, depth-coordinate ocean–sea-ice model that solves the primitive equations using Boussinesq and hydrostatic approximations on a structured mesh. The basic configuration of the global ocean–sea-ice model used for CMIP6-OMIP is identical to that of MRI-ESM2 and fully described by Yukimoto et al. (2019) and Urakawa et al. (2020), which is briefly summarized here. The horizontal grid system adopts the Murray's (1996) tripolar grid and the nominal horizontal resolution is 1-degree in longitude and 0.5° in latitude with an enhancement to 0.3° between 10°S and 10°N. The vertical grid system adopts a vertically rescaled height coordinate (z* coordinate) proposed by Adcroft and Campin (2004). The number of vertical layers is 60, with the layer thicknesses not exceeding 10 m in the upper 200 m and a bottom boundary layer (BBL) of Nakano and Suginohara (2002) of 50 m thickness attached to the bottom.

The model adopts the generalized Arakawa scheme as described by Ishizaki and Motoi (1999) for the momentum advection terms and the second order moment scheme of Prather (1986) with the flux limiter with the method B proposed by Morales Maqueda and Holloway (2006) for the tracer advection terms. The flow-dependent anisotropic horizontal viscosity scheme of Smith and McWilliams (2003) is used. As a turbulence closure scheme for boundary layer mixing, we adopted the generic length scale scheme of Umlauf and Burchard (2003), where a prognostic equation of the (generic) length scale is solved along with that of the turbulence kinetic energy. The background vertical diffusion coefficients have the 3-dimensional empirical distribution based on Decloedt and Luther (2010). The vertical diffusivity was locally set to a large value of 1 $m^2 s^{-1}$ whenever unstable stratification is detected in the model. The isopycnal tracer diffusion scheme of Redi (1982) and the eddy induced tracer transport scheme of Gent and McWilliams (1990) are used to parameterize stirring by mesoscale eddies. The constant isopycnal tracer diffusivity is 1500 $m^2 s^{-1}$ with two tapering factors applied for the oceanic interior and a layer near the sea surface, proposed by Danabasoglu and McWilliams (1995; their Eq. A.7a with Sc = 0.08 and Sd = 0.01) and Large et al. (1997; their Eq. B.4), respectively. Since these tapering factors were applied to all elements of the isopycnal tracer diffusion tensor except for the horizontal diagonal ones, the isopycnal diffusion is gradually modified to the

horizontal diffusion around steeply tilted isopycnal surfaces and within surface diabatic layer. The coefficient for Gent and McWilliams parameterization is calculated with schemes of Danabasoglu and Marshall (2007) and Danabasoglu et al. (2008). It depends on a local buoyancy frequency and ranges from 300 $m^2$ $s^{-1}$ in weakly stratified regions to 1500 $m^2$ $s^{-1}$ in strongly stratified regions. Within the surface diabatic layer, the parameterized eddy induced transport is modified, so that a corresponding meridional overturning streamfunction linearly tapers to zero at the sea surface from the bottom of the

diabatic layer. We put a ceiling at 0.005 of the isopycnal slope evaluated in this parameterization.

In the sea-ice component, the thermodynamics is based on Mellor and Kantha (1989). For categorization by thickness, ridging, and rheology, those of Los Alamos National Laboratory sea-ice model (CICE; Hunke and Libscomb, 2006) are adopted. Fractional area, snow volume, ice volume, ice energy, and ice surface temperature of each thickness category are transported using the multidimensional positive definite advection transport algorithm (MPDATA) of Smolarkiwicz (1984).

Both OMIP-1 and OMIP-2 simulation conducted by MRI.COM follow the protocols without notable deviations. For salinity restoring, a piston velocity of 50 m per 365 days is applied to all ocean grid points except for coastal grid points with sea-ice. For computing surface turbulent fluxes, the velocity vector at the first vertical layer of the model is fully subtracted from the surface wind vector (i.e., $\alpha = 1$).

**A11 NorESM-BLOM**

The NorESM-BLOM contribution uses the ocean and sea-ice components of the Norwegian Earth System Model version 2 (NorESM2; Seland et al., 2020) and the configuration and parameters of these active OMIP model components are identical in all CMIP6 contributions of NorESM2. The model framework is based on CESM2 and the application of OMIP-1 and OMIP-2 forcing is identical to that of CESM-POP (appendix A3) and specifically $\alpha = 1$ is used in the estimation of the near-surface wind correction.

The ocean component Bergen Layered Ocean Model (BLOM) shares many features of the ocean component in the BERGEN contribution to in previous CORE studies (Danabasoglu et al., 2014) and uses a C-grid discretization with 51 isopycnic layers referenced at 2000 dbar and a surface mixed layer divided into two non-isopycnic layers. A second-order turbulence closure ($k$-$\varepsilon$ model) is now used for vertical shear-induced mixing. The parameterization of mesoscale eddy-induced transport is modified to more faithfully comply with the Gent and McWilliams (1990) formulation. Mixed layer

physics have been improved, in part to enable sub-diurnal coupling of the ocean. The hourly coupling now used has made it possible to add additional energy sources for upper ocean vertical mixing such as wind work on near-inertial motions and surface turbulent kinetic energy source due to wind stirring to the $k$-$\varepsilon$ model. To achieve more realistic mixing in gravity currents, the layer thickness at velocity points has been redefined and realistic channel widths are used (e.g., Strait of Gibraltar). The sea-ice model is CICE5.1.2 which is identical to the sea-ice model of CESM-POP except for some notable

differences: it is configured on a different horizontal grid; a parameterization of wind drift of snow similar to Lecomte et al. (2013) is implemented and enabled; accurate time averaging of zenith angle used in albedo calculations is applied. More details on the model formulations can be found in Bentsen et al. (2019).

The same tripolar grid locations with 1° resolution along the equator as in the previous BERGEN CORE contribution are used, but with the following grid differences: the ocean mask is modified to allow the B-grid staggered sea-ice model to transport sea-ice in narrow passages; the Black Sea is connected to the Mediterranean and the Caspian Sea is closed resulting in no disconnected basins in NorESM-BLOM; sill depths in the region of the Indonesian Throughflow and passages through mid-ocean ridges are revised and edited to observed depths.

BLOM was initialized with temperature and salinity fields from the Polar science center Hydrographic Climatology (PHC) 3.0 (updated from Steele et al., 2001). Sea surface salinity is restored to monthly climatology with a piston velocity of 50 m per 300 days applied globally for both OMIP-1 and OMIP-2 simulations. The restoring salt flux is normalized so that the global area weighted sum of the restoring flux is zero. The OMIP-1 and OMIP-2 simulations of NorESM-BLOM have completed 6 forcing cycles of the forcing periods 1948–2009 and 1958–2018, respectively.

## Appendix B: Discussion of OMIP-2 forcing datasets and experimental protocols

In Appendix B, results from additional sensitivity studies are presented to further understand the present assessment and to discuss future revision of protocols and datasets. Information about the additional experiments is summarized in Table B1.

### B1 Sensitivity to the period of forcing datasets used for repeating the simulation cycles

To make a fair comparison between OMIP-1 and OMIP-2 forcing datasets, the common period (1958–2009) of OMIP-1 (CORE) and OMIP-2 (JRA55-do) datasets is used in additional experiments to force models for six cycles of the forcing dataset by two groups (MIROC-COCO4.9 and MRI.COM). These simulations, by comparing with the simulation results using the full length of the forcing datasets, can also be used to isolate the effect of the first (1948–1957) and final (2010–2018) decade of the forcing dataset on the full-length OMIP-1 and OMIP-2 simulations, respectively.

Figures B1 and B2 show the long-term drift of heat content and ocean circulation metrics, respectively. Overall, cutting the first ten years and the final nine years from OMIP-1 and OMIP-2 simulations respectively does not result in major differences in the metrics. This means that the features of long-term mean and drift in OMIP-1 and OMIP-2 simulations by individual models are largely determined by the common 52 years (1958–2009) of the forcing datasets. Specifically, the memory of the rapid warming in the final nine years (2010–2018) in OMIP-2 simulations is lost in the first ten to fifteen years of the following cycle (e.g., Fig. B1b). Note that the increasing difference in some metrics (e.g., the difference of heat content between 2000 m and the bottom in OMIP-2 simulations of MIROC-COCO4.9) is presumably caused by the difference in the total simulation lengths (shorter by nine years in each cycle of the 1958–2009 simulations of OMIP-2 relative to the full-length simulations).

**B2 Sensitivity to formulae computing property of moist air**

It has been recommended to use a set of formulae for computing properties of moist air provided by Gill (1982) instead of Large and Yeager (2004; 2009) by Tsujino et al. (2018). However, for this study we did not impose this on all participating groups. Sensitivity to the change of formulae is reported in this appendix by using OMIP-2 simulations conducted by MRI.COM.

Figure B3 shows the long-term drift of heat content of the two experiments that change the set of formulae for properties of moist air used to compute surface turbulent fluxes. The use of Large and Yeager (2004; 2009) formulae results in the slightly colder temperature in the deep to bottom layer, which results in the slightly stronger (by less than 1 Sverdrups) global meridional overturning circulation associated with the Antarctic Bottom Water/Circumpolar Deep Water formation. However, differences are generally very small.

Figure B4 compares the biases of sea surface temperature (SST). The use of Gill (1982) formulae results in lower SST in the tropics. This is presumably caused by the higher saturation specific humidity for a temperature range higher than about 25°C in the Gill (1982) formula than Large and Yeager (2004; 2009) formula, resulting in the larger latent heat flux out of the ocean with the Gill (1982) formula. For MRI.COM, the use of Gill (1982) formulae results in a smaller root-mean-square error of SST in the OMIP-2 simulation. Overall, one can make a safe transition in the use of formulae for properties of moist air from Large and Yeager (2004; 2009) to Gill (1982).

**B3 Sensitivity to the contribution of oceanic surface currents to relative winds**

There has been progress in understanding the air-sea coupling processes in producing air-sea stresses and their impacts on ocean circulation and energetics. Particularly notable is the finding of the imprint of ocean currents on the atmospheric winds, which is found in atmosphere–ocean coupled models (Renault et al., 2016, 2019b) and confirmed in the winds measured by satellites (Renault et al., 2017). The air-sea stresses are known to dampen mesoscale eddy fields (e.g., Zhai and Greatbatch, 2007), but the imprints of such mesoscale ocean currents on the atmospheric winds are shown to partly reenergize mesoscale ocean currents. Correspondingly, there is active research in determining how best to force an ocean model with prescribed atmospheric winds (Renault et al., 2019a, 2020). Renault et al. (2019a) suggested two approaches: One is to correct the computation of the relative wind (Renault et al., 2016), and the other is to correct the wind stress (Renaults et al., 2017), with the latter recommended by Renault et al. (2020) based on an atmosphere–ocean coupled model. We note that the wind stress correction approach only corrects wind stress and that it may come at the expense of a known relationship among the turbulent fluxes (momentum, specific heat, and water vapor fluxes). If we follow the wind correction approach, possibly at the expense of a less realistic representation of the mesoscale activity, the relative winds can be obtained from $\Delta\vec{U} = \vec{U_a} - \alpha\vec{U_o}$, where $\vec{U_a}$ is the (atmospheric) surface wind vector, $\vec{U_o}$ is the ocean surface current vector (usually the vector at the first ocean model level), and $\alpha$ is a parameter between 0 and 1 controlling the fraction of the ocean surface currents to be included in the relative wind calculation. Renault et al. (2019b) suggested $\alpha \sim 0.70$ based on an average between 45°S–45°N in their

atmosphere–ocean coupled model. The community has not yet reached a consensus on the way $\alpha$ should be imposed in ocean–sea-ice simulations.

To study the sensitivity to changing the contribution from ocean surface currents to relative winds for computing surface turbulent fluxes in the OMIP-2 framework, we compare simulations conducted by CAS-LICOM3 ($\alpha = 0.7$, 1.0) and MRI.COM ($\alpha = 0.0$, 0.7, 1.0) that used different $\alpha$'s. It turned out that differences in the spin-up behaviors and mean values of metrics caused by the change of $\alpha$ are generally much smaller than the model – model differences, nor do they significantly impact any observational comparisons. A notable exception is the surface zonal current in the eastern tropical Pacific, with the eastward flowing North Equatorial Counter Current reaching 0.1 m s$^{-1}$ for the case of $\alpha = 0.0$ of MRI.COM, which compares more favourably with the observational estimates (see Fig. 18 and Fig. S45) than about 0.05 m s$^{-1}$ obtained with $\alpha = 0.7$ and 1.0. Note that simulations with $\alpha = 0.0$ (OMIP-1 and OMIP-2 by EC-Earth3-NEMO and OMIP-2 by FSU-HYCOM) also produce relatively strong North Equatorial Counter Current (Figs. S45 and S46). However, we note that the present low sensitivity of metrics generally found in these simulations may only apply to low resolution models. In fine horizontal resolution models where active mesoscale eddy field and boundary currents are well resolved, the impact of changing $\alpha$ is expected to be enhanced. Thus, a careful consideration is required for the treatment of surface currents in generating surface wind products as well as in defining the method for computing surface turbulent fluxes to further advance the ocean modelling activity with the OMIP-2 framework.

**Appendix C: Observational data used for validation**

This appendix gives a summary of observational datasets used to evaluate OMIP-1 and OMIP-2 simulations and additional processing on the observational datasets done before they are directly compared with simulations. Table C1 summarizes the variables and their sources and locations where they are available for downloading.

Most datasets were able to be used for evaluation as they were, but the sea surface height (dynamic sea level) provided by CMEMS needed some preprocessing. First, the data was averaged temporally to generate a monthly mean time series and then regridded spatially from the original 0.25° latitude – 0.25° longitude grid to the 1° latitude – 1° longitude grid using a gaussian filter with a half width of 1.5°. This treatment is to reduce the imprints of individual mesoscale eddies in the dataset before the dataset is compared with the results of low-resolution models that do not resolve mesoscale eddies. Then, in each month, the data was offset by subtracting the quasi-global mean value computed by averaging the data over the points where valid values are available during the whole period (Jan 1993–Dec 2009) for comparison. The same operation is applied to the simulated sea surface height, specifically, the monthly simulation data is offset by subtracting its quasi-global mean value computed by averaging the data over the points where valid values from CMEMS are available during the period for comparison. Note that the Mediterranean and Black Sea are excluded from this averaging operation.

**Appendix D: Metrics of individual models**

In this appendix, we list specific values for the following metrics from individual models.

- Drift of vertically averaged temperatures evaluated as the deviation of the long-term (1980–2009) mean of the last cycle relative to the annual mean of the initial year of integration (Table D1);
- Circulation metrics determined by the long-term (1980–2009) means from the last cycle (Table D2);
- Biases of sea surface temperature, salinity, and height (Table D3);
- Biases of mixed layer depth in winter and summer as well as the winter mixed layer depth in the subpolar North Atlantic and the marginal seas around Antarctica (Table D4);
- Biases of basin-wide averaged temperature (Table D5) and salinity (Table D6);
- Mean sea-ice extent in summer and winter of both hemispheres (Table D7).

Note that for the MMM rows included in the tables, multi-model mean fields are constructed first and then metrics are computed. In contrast, for the ensemble mean and ensemble std rows, metrics of individual models are computed first and then their ensemble mean and standard deviation are computed.

The multi-model mean outperforms the majority of models in its root-mean square bias for many metrics, though we note that the GFDL-MOM configuration performs best among the models for many metrics. We found no obvious grouping of model skill metrics in terms of model formulation (e.g., the hybrid vertical coordinate models) and model code (e.g., the NEMO models). Discussions using these tables are given in the corresponding main part of the paper, and Section 6 discusses ordering among the models as listed in Table 3.

**Appendix E: Statistical assessments of seasonal and interannual variability**

In this appendix, we present some objective assessments of model performances. The reproducibility of seasonal and interannual variations is assessed using statistics employed by AMIP (Gates et al., 1999).

**E1. Mathematical formulations**

We first present mathematical formulations for the statistical properties used to assess model performances in this appendix. These statistical properties were used for the AMIP paper (Gates et al., 1999). The notations used in Table 1 of Wigley and Santer (1990) is followed.

First, we define the fields to be tested. For seasonal variability, the anomaly of monthly climatology from the climatology of annual mean is used as the test field,

$$d'_{xt} = \text{(monthly climatology)} - \text{(annual mean climatology)}. \quad \text{(E1)}$$

For interannual variability, the deviation of monthly times series from the monthly climatology is used,

$$d'_{xt} = \text{(monthly time series)} - \text{(monthly climatology)}.$$

Temporal correlation coefficients ($r_x$) with the reference field $m'_{xt}$ are calculated locally to depict two-dimensional distribution of correlation coefficients,

$$r_x = \sum_t (d'_{xt} - \bar{d}'_{x,}) \cdot (m'_{xt} - \overline{m}'_{x,})/N_t s_{d',x} s_{m',x}, \quad \text{(E2)}$$

where

$$s_{d',x}^2 = \sum_t (d'_{xt} - \bar{d}'_{x,})^2 / N_t \text{ and } s_{m',x}^2 = \sum_t (m'_{xt} - \overline{m}'_{x,})^2 / N_t.$$

As in the AMIP paper (Gates et al. 1999), overall space-time correlation coefficient ($r$) is computed to draw Taylor diagrams for the test field,

$$r = \sum_{x,t} [(d'_{x,t} - \langle d' \rangle)] \cdot [(m'_{x,t} - \langle m' \rangle)]/N_x N_t \bar{s}_{d'} \bar{s}_{m'}, \quad \text{(E3)}$$

where

$$\langle d \rangle = \sum_{x,t} d_{x,t} / N_x N_t, \; \bar{s}_{d'}^2 = \sum_{x,t} \frac{(d'_{x,t} - \langle d' \rangle)^2}{N_x N_t}, \text{ and } \bar{s}_{m'}^2 = \sum_{x,t} \frac{(m'_{x,t} - \langle m' \rangle)^2}{N_x N_t}.$$

Taylor diagram for $d'_{x,t}$ relative to $m'_{x,t}$ can be drawn by using $r$, $\bar{s}_{d'}^2$, $\bar{s}_{m'}^2$.

Taylor diagram and correlation coefficients do not give information about the model error of the long-term mean. Another assessment diagram is also proposed by the AMIP paper.

SITES (depicted on abscissa in Figs. E6 and E7) as introduced by Preisendorfer and Barnett (1983) measures bias of a long-term mean,

$$\text{SITES} = N_t \sum_x (\bar{d}_{x,} - \bar{m}_{x,})^2 / \sigma_D \sigma_M, \quad \text{(D4)}$$

where

$$\sigma_D^2 = \sum_{x,t} (d_{x,t} - \bar{d}_{x,})^2 \text{ and } \sigma_M^2 = \sum_{x,t} (m_{x,t} - \bar{m}_{x,})^2.$$

RBAR ($\bar{r}$, depicted on ordinate in Figs. E6 and E7) as introduced by Wigley and Santer (1990) measures temporal evolution of spatial pattern correlation to assess spatio-temporal variability,

$$\text{RBAR} = \bar{r} = \sum_t r_t / N_t, \quad \text{(D5)}$$

where

$$r_t = \sum_x [(d_{xt} - \bar{d}_{x,}) - (\bar{d}_{,t} - \langle d \rangle)] \cdot [(m_{xt} - \bar{m}_{x,}) - (\bar{m}_{,t} - \langle m \rangle)]/N_x \bar{s}_{d,t} \bar{s}_{m,t}, \quad \text{(D6)}$$

$$\bar{s}_{d,t}^2 = \sum_x [(d_{xt} - \bar{d}_{x,}) - (\bar{d}_{,t} - \langle d \rangle)]^2 / N_x, \text{ and } \bar{s}_{m,t}^2 = \sum_x [(m_{xt} - \bar{m}_{x,}) - (\bar{m}_{,t} - \langle m \rangle)]^2 / N_x.$$

**E2. Assessment**

Figure E1 presents the assessment of monthly climatology of sea surface temperature (SST) for the period 1980–2009. The correlation coefficient between each simulation and PCMDI-SST is calculated locally and the multi-model mean is shown in Figs. E1a and E1b for OMIP-1 and OMIP-2, respectively. The low correlation coefficients around the equator in OMIP-1 are improved in OMIP-2. Figure E1d shows the Taylor diagram for the total space-time pattern variability. This figure is used to

compare overall performance of simulations in an objective manner and it shows that all simulations well reproduce the SST seasonal variability.

Figure E2 presents the SST interannual variability for the period 1980–2009. The correlation coefficient of the time series of monthly anomalies relative to monthly climatology between each simulation and PCMDI-SST is calculated locally and the multi-model mean is shown in Figs. E2a and E2b respectively. The correlation coefficients become slightly higher by about 0.05 in most regions in OMIP-2. The Taylor diagram for the total space-time pattern variability (Fig. E2d) shows that the correlation coefficients become high in OMIP-2, while the amplitudes of variability become slightly smaller.

Figures E3 and E4 present the corresponding analysis of SSH. There is no notable difference in the performance of monthly climatology, while correlation coefficients for interannual variabilities become higher in the low latitude regions in OMIP-2 relative to OMIP-1. The amplitudes of variability of OMIP-2 are slightly smaller than OMIP-1.

Figure E5 presents the monthly climatology of mixed layer depths. Note that the regions where mixed layer depths reach more than 1000 meters in winter, specifically the Weddell Sea and the high latitude north Atlantic Ocean (Fig. 11), are excluded from this assessment because amplitudes of seasonal variation there dominate the global assessment. Both OMIP-1 and OMIP-2 simulations give lower correlation coefficients in low latitudes than in high latitudes. The Taylor diagram suggests that the overall performance is similar between OMIP-1 and OMIP-2.

Figures E6 and E7 present another perspective for the assessment of model performance. The normalized error of the long-term annual mean (SITES; abscissa), which measures the bias of a long-term mean, and the temporal mean of the spatial pattern correlation coefficients (RBAR; ordinate), which measures spatio-temporal variability, relative to reference datasets, are plotted for all simulations. These diagrams clearly show that the space-time variabilities are reproduced better in OMIP-2 than in OMIP-1. On the other hand, errors of the long-term mean are modestly improved for SST while slightly degraded for SSH. It is noted that no decisive relation between bias and correlation is found in these diagrams, as also noted in the AMIP paper (Gates et al., 1999). It is noted that the rather low score of the space-time variability (RBAR) of SSH for GFDL-MOM despite its best performance of the long-term mean (SITES), is presumably because mesoscale eddies appear in this model by employing the 1/4° grid spacing. Correlation coefficients of GFDL-MOM are lower than other models in the Antarctic Circumpolar Current region and the western boundary current regions, which are populated with mesoscale eddies (Figs. S58 and S59). This would call for more improved methods to statistically assess the performance of models that resolve mesoscale eddies.

**Code availability**

Python scripts used to process data and generate figures (Tsujino et al., 2020a) are available at https://doi.org/10.26300/02tn-gf72.

**Data availability**

The forcing dataset for OMIP-1 is available at https://data1.gfdl.noaa.gov/nomads/forms/core.html and that for OMIP-2 is available through input4MIPs (https://esgf-node.llnl.gov/search/input4mips/). An archive for all of the model outputs
(Tsujino et al., 2020b) is available at https://doi.org/10.26300/g2a0-5x34 and that for the analysis and observational data
(Tsujino et al., 2020c) used for evaluation is available at https://doi.org/10.26300/60wh-ak09.

**Author contribution**

HT, SMG, and GD proposed and led this evaluation study. BFK and SJM organized and supervised the overall activities as co-chairs of the CLIVAR-OMDP. HT and LSU processed the model outputs and produced figures. The following authors
are responsible for individual models, simulations, and diagnostics: QW, SD, NK, and DS for AWI-FESOM; PL, HL, YL, and ZY for CAS-LICOM3; WMK, SGY, GD, LK, and MCL for CESM-POP; DI, PGF, and SM for CMCC-NEMO; RB, AEA, TA, EE, VL, YRR, VS for EC-Earth3-NEMO; AB and EPC for FSU-HYCOM; SMG, RD, and AJA for GFDL-MOM; MS, JKR, and CWB for Kiel-NEMO; YK, TS, and HT for MIROC-COCO4.9; LSU and HT for MRI.COM; and MB, CG, AN, and MI for NorESM-BLOM. All authors contributed to the writing and editing processes.

**Competing interests**

The authors declare that they have no conflict of interest.

**Acknowledgements**

This work is benefitted from the continuous support and feedback from the members of CLIVAR-OMDP as well as the many ocean modelers and modelling groups who used the CORE and JRA55-do datasets. Comments from Frank O. Bryan,
an anonymous referee, and the editor greatly help improve the earlier version of the manuscript. We acknowledge the World Climate Research Programme, which, through its Working Group on Coupled Modelling, coordinated and promoted CMIP6. We thank the Earth System Grid Federation (ESGF) for archiving the data and providing access and the multiple funding agencies who support CMIP6 and ESGF.

Sea surface height (dynamic sea level) data from satellite altimetry is provided by E.U. Copernicus Marine Service
Information. Other observational datasets used for the present evaluations are provided by authors of the dataset or institutions that maintain them as summarized in Appendix C.

The JMA-MRI contribution to this study was supported by Meteorological Research Institute. The MIROC-COCO4.9 and JMA-MRI contributions were partially supported by the Integrated Research Program for Advancing Climate Models (TOUGOU) Grant Number JPMXD0717935457 and JPMXD0717935561, respectively, from the Ministry of Education,

Culture, Sports, Science and Technology (MEXT), Japan. The AWI contributors (Qiang Wang, Dmitry Sidorenko, Sergey Danilov and Nikolay Koldunov) acknowledge funding from the projects S1 (Diagnosis and Metrics in Climate Models) and S2 (Improved parameterizations and numerics in climate models) of the Collaborative Research Centre TRR 181 "Energy Transfer in Atmosphere and Ocean" funded by the Deutsche Forschungsgemeinschaft (DFG, German Research Foundation) – project no. 274762653, Helmholtz Climate Initiative REKLIM (Regional Climate Change) and European Union's Horizon

2020 Research & Innovation programme through grant agreement No. 727862 APPLICATE. The NorESM-BLOM contribution was supported by the Research Council of Norway (projects EVA (229771) and INES (270061)) and Centre for Climate Dynamics at the Bjerknes Centre for Climate Research and simulations were performed on resources provided by UNINETT Sigma2 - the National Infrastructure for High Performance Computing and Data Storage in Norway. NCAR contribution was supported by the National Oceanic and Atmospheric Administration (NOAA) Climate Program Office

Climate Variability and Predictability Program. NCAR is a major facility sponsored by the US National Science Foundation (NSF) under Cooperative Agreement No. 1852977. Pengfei Lin, Hailong Liu, Zipeng Yu and Yiwen Li are supported by the National Natural Science Foundation of China (Grants No. 41931183 and 41976026). Raphael Dussin's research at the Geophysical Fluid Dynamics Laboratory is supported by NOAA's Science Collaboration Program and administered by UCAR's Cooperative Programs for the Advancement of Earth System Science (CPAESS) under awards NA16NWS4620043

and NA18NWS4620043B. Alistair Adcroft acknowledges support for his work at the Geophysical Fluid Dynamics Laboratory from Award NA18OAR4320123 of the National Oceanic and Atmospheric Administration.

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

**Tables**

**Table 1. Configurations of participating models. See appendix A for detailed descriptions.**

| Model name | Configuration | Ocean model and version | Sea ice model and version | Horizontal grid (arrangement) | Orientation | Nominal horizontal resolution | Vertical grid (the number of levels) |
|---|---|---|---|---|---|---|---|
| AWI-FESOM | | FESOM v1.4 | FESIM v2 | Unstructured | displaced | $1^{\circ\#}$ | z (46) |
| CAS-LICOM3 | | LICOM3 | CICE4 | Structured (B) | tripolar | $1^{\circ\#}$ | $\eta$ (30) |
| CESM-POP | | POP2 | CICE 5.1.2 | Structured (B) | displaced | $1^{\circ\#}$ | z (60) |
| CMCC-NEMO | | NEMO v3.6 | CICE 4.1 | Structured (C) | tripolar | $1^{\circ\#}$ | z (50) |
| EC-Earth3-NEMO | ORCA1 | NEMO v3.6 | LIM 3 | Structured (C) | tripolar | $1^{\circ\#}$ | z (75) |
| FSU-HYCOM | | HYCOM | CICE 4.1 | Structured (C) | tripolar | $0.72^{\circ\#}$ | hybrid z-$\rho(\sigma_2)$-$\sigma$ (41)$^{\#}$ |
| GFDL-MOM | OM4 | MOM6 | SIS2 | Structured (C) | tripolar | $1/4^{\circ}$ | hybrid z-$\rho(\sigma_2)$ (75)$^{\#}$ |
| Kiel-NEMO | ORCA05 | NEMO v3.6 | LIM 2 | Structured (C) | tripolar | $0.5^{\circ}$ | z (46) |
| MIROC-COCO4.9 | | COCO4.9 | COCO4.9 | Structured (B) | tripolar | $1^{\circ\#}$ | $\sigma$-z (62+BBL) |
| MRI.COM | GONDOLA100 | MRI.COMv4 | CICE3, Mellor and Kantha (1989) | Structured (B) | tripolar | 100 km$^{\#}$ | z* (60+BBL) |
| NorESM-BLOM | | BLOM | CICE 5.1.2 | Structured (C) | tripolar | $1^{\circ\#}$ | $\rho(\sigma_2)$ (51) |

# See appendix A for additional details.

**Table 2.** *z*-scores of the difference between OMIP-2 and OMIP-1 simulations for metrics consisting of time series of index values. The differences are evaluated for 1980–2009 of the last cycle. Note that if a *z*-score is beyond ±1.64, the difference is statistically significant at 90% confidence level. The uncertainty of multi-model mean difference is computed based on the method proposed by Wakamatsu et al. (2017). Abbreviations used for metrics are *VAT* vertically averaged temperature, *SST* sea surface temperature, *SIE* sea ice extent, *SIV* sea ice volume, *NH* northern hemisphere, *SH* southern hemisphere, *AMOC* Atlantic meridional overturning circulation maximum at 26.5°N, *GMOC* Global meridional overturning circulation minimum in 2000 m – bottom depth at 30°S, *ACC* Antarctic circumpolar current passing through the Drake Passage, and *ITF* Indonesian through flow. Note that VAT drift is evaluated as the deviation of the 1980–2009 mean of the last cycle relative to the annual mean of the initial year of the simulation by each model.

| metric | *z*-score of omip2 – omip1 | metric | *z*-score of omip2 – omip1 | metric | *z*-score of omip2 – omip1 |
|---|---|---|---|---|---|
| VAT (0–700 m) drift | 0.77 | SIE NH Mar | −0.53 | AMOC | 0.04 |
| VAT (0–2000 m) drift | 0.61 | SIE NH Sep | 0.32 | GMOC | −0.08 |
| VAT (2000 m–bottom) drift | −0.16 | SIE SH Mar | −1.49 | ACC | −0.19 |
| VAT (top–bottom) drift | 0.14 | SIE SH Sep | 1.21 | ITF | −0.13 |
| SST | −0.46 | SIV NH | 0.88 | SIV SH | 0.77 |

**Table 3.** *r²*-scores of linear fits for model scatters between OMIP-1 and OMIP-2 simulations in some globally integrated/averaged quantities and circulation metrics. High *r²*-scores (> 0.8) are emphasized with bold letters. The symbol in the parentheses after each metric indicates the table number in Appendix D (Tables D1–D7) listing specific values from individual models. See the caption of that table for the explanation about the metric.

| metric | *r²*-score | metric | *r²*-score | metric | *r²*-score |
|---|---|---|---|---|---|
| VAT (0–700 m) drift (D1) | 0.644 | SST bias rmse (D3) | **0.961** | ZMT Southern Ocean bias rmse (D5) | 0.308 |
| VAT (0–2000 m) drift (D1) | 0.615 | SST bias mean (D3) | **0.951** | ZMT Atlantic bias rmse (D5) | 0.753 |
| VAT (2000 m–bottom) drift (D1) | 0.673 | SSS bias rmse (D3) | **0.934** | ZMT Indian bias rmse (D5) | **0.938** |
| VAT (top–bottom) drift (D1) | 0.665 | SSS bias mean (D3) | **0.819** | ZMT Pacific bias rmse (D5) | 0.725 |
| AMOC (D2) | 0.510 | MLD Win bias rmse (D4) | **0.965** | ZMS Southern Ocean bias rmse (D6) | 0.674 |
| GMOC (D2) | 0.431 | MLD Win bias mean (D4) | **0.830** | ZMS Atlantic bias rmse (D6) | **0.867** |
| ACC (D2) | 0.415 | MLD Sum bias rmse (D4) | **0.812** | ZMS Indian bias rmse (D6) | **0.848** |
| ITF (D2) | **0.910** | MLD Sum bias mean (D4) | **0.861** | ZMS Pacific bias rmse (D6) | 0.592 |
| MLD N Atlantic (D4) | 0.436 | SIE NH Mar (D7) | **0.982** | SIE SH Mar (D7) | 0.631 |
| MLD Antarctica (D4) | 0.613 | SIE NH Sep (D7) | **0.951** | SIE SH Sep (D7) | **0.955** |
| SSH bias rmse (D3) | **0.910** | | | | |

**Table A1. Experimental settings and the minimum information to identify the experiments in ESGF for participating models. See appendix A for detailed descriptions.**

| Model name | Salinity restoring | | Surface current contribution to relative wind (α) | | Source ID in CMIP6/ESGF | Variant Label in CMIP6/ESGF | | Additional notes Notable deviations from the protocols |
|---|---|---|---|---|---|---|---|---|
| | OMIP-1 | OMIP-2 | OMIP-1 | OMIP-2 | | OMIP-1 | OMIP-2 | |
| AWI-FESOM | 50m/300days[#] | 50m/300days[#] | 1 | 1 | tbd | tbd | tbd | |
| CAS-LICOM3 | 20m/1yr | 20m/1yr | 1 | 1 | FGOALS-f3-L | r1i1f1p1 | r1i1f1p1 | |
| CESM-POP | 50m/1yr | 50m/1yr | 1 | 1 | CESM2 | r2i1f1p1 | r1i1f1p1 | The r1i1f1p1 of OMIP-1 at ESGF is run for 5 cycles. |
| CMCC-NEMO | 50m/1yr[#] | 50m/6month[#] | 1 | 1 | CMCC-CM2-SR5 | r1i1p1f1 | r1i1p1f1 | |
| EC-Earth3-NEMO | 50m/180days | 50m/180days | 0 | 0 | EC-Earth3 | tbd | tbd | Atmosphere–ocean coupled model also uses α = 0. |
| FSU-HYCOM | 50m/4yr[#] | 50m/4yr[#] | 1 | 0 | n/a | n/a | n/a | 1958-2015 for 1-5 cycles of OMIP-2[%] |
| GFDL-MOM | 50m/300days | 50m/300days | 1 | 1 | GFDL-CM4 | r1i1f1p1 | r1i1f1p1 | 1948-2007 for 1-5 cycles of OMIP-1[%] 1958-2017 for 1-3 cycles of OMIP-2[%] |
| Kiel-NEMO | 50m/1yr | 50m/1yr | 1 | 1 | n/a | n/a | n/a | |
| MIROC-COCO4.9 | 50m/1yr[#] | 50m/1yr[#] | 1 | 1 | MIROC6 | r1i1p1f1 | r1i1p1f1 | 5 cycles for OMIP-1[%] |
| MRI.COM | 50m/365days[#] | 50m/365days[#] | 1 | 1 | MRI-ESM2-0 | r2i1p1f1 | r1i1p1f1 | Gill (1982) is used to compute properties of moist air. |
| NorESM-BLOM | 50m/300days | 50m/300days | 1 | 1 | NorESM2-LM | r1i1p1f1 | r1i1p1f1 | The r1i1p1f1 of OMIP-1 at ESGF contains only the first 5 cycles. |

# See appendix A for additional details.

% Since this is not a full length (372 years for OMIP-1 and 366 years for OMIP-2) simulation, both OMIP-1 and OMIP-2 simulations by this model are not included in the multi-model ensemble means to compare spin-up behaviors of OMIP-1 and OMIP-2 simulations in Section 3.

**Table B1. Description of the additional experiments conducted for Appendix B with the minimum information to identify the experiments (if available) in ESGF.**

| model name | description (variant_info) | Source ID | Experiment ID | variant label |
|---|---|---|---|---|
| MIROC.COCO4.9 | CMIP6 omip1 experiment run for 6 cycles of 1958–2009 OMIP-1 (CORE-II) forcing. | MIROC6 | omip1 | r2i1p1f1 |
| MIROC.COCO4.9 | CMIP6 omip2 experiment run for 6 cycles of 1958–2009 OMIP-2 (JRA55-do-v1.4) forcing. | MIROC6 | omip2 | r2i1p1f1 |
| MRI.COM | CMIP6 omip1 experiment run for 6 cycles of 1958–2009 OMIP-1 (CORE-II) forcing. | MRI-ESM2-0 | omip1 | r3i1p1f1 |
| MRI.COM | CMIP6 omip2 experiment run for 6 cycles of 1958–2009 OMIP-2 (JRA55-do-v1.4) forcing. | MRI-ESM2-0 | omip2 | r2i1p1f1 |
| MRI.COM | CMIP6 omip2 experiment using empirical formulae for computing properties of moist air given by Large and Yeager (2004) instead of those given by Gill (1982). | MRI-ESM2-0 | omip2 | r1i1p4f1 |
| CAS-LICOM | CMIP6 omip2 experiment where 70% of ocean surface currents are subtracted from surface winds in the calculation of relative winds for the surface flux computations. ($\alpha=0.7$) | FGOALS-f3-L | omip2 | n/a |
| MRI.COM | CMIP6 omip2 experiment where 70% of ocean surface currents are subtracted from surface winds in the calculation of relative winds for the surface flux computations. ($\alpha=0.7$) | MRI-ESM2-0 | omip2 | r1i1p3f1 |
| MRI.COM | CMIP6 omip2 experiment where ocean surface currents are not subtracted from surface winds in the calculation of relative winds for the surface flux computations. ($\alpha=0.0$) | MRI-ESM2-0 | omip2 | r1i1p2f1 |

**Table C1. Observational data used to evaluate simulations.**

| Variable | Data name and source | Available online at |
|---|---|---|
| Sea surface temperature and Sea ice concentration | PCMDI-SST: SST/sea ice consistency criteria by Hurrell et al. (2008) are applied to merged SST based on UK MetOffice HadISST and NCEP OI2. | https://esgf-node.llnl.gov/search/input4mips/ (PCMDI-AMIP-1-1-4) |
| | COBE-SST: Ishii et al. (2005) | https://ds.data.jma.go.jp/tcc/tcc/products/elnino/cobesst/cobe-sst.html |
| Sea ice extent in each hemisphere | National snow and ice data center Sea Ice Index, Fetterer et al. (2017) | https://nsidc.org/data/seaice_index/archives |
| Temperature and salinity climatology | World ocean atlas 2013 version 2, Locarnini et al. (2013), Zweng et al. (2013) | https://www.nodc.noaa.gov/OC5/woa13/woa13data.html |
| Ocean heat content | Zanna et al. (2019) | https://laurezanna.github.io/post/ohc_updated_data/ |
| | Chen et al. (2017) | http://159.226.119.60/cheng/ |
| | Ishii et al. (2017) | https://climate.mri-jma.go.jp/pub/ocean/ts/v7.2/doc/00README |
| | Levitus et al. (2012) | https://www.nodc.noaa.gov/OC5/3M_HEAT_CONTENT/basin_data.html |
| Thermosteric sea level | Global sea level budget, WCRP global sea level budget group (2018) | https://www.seanoe.org/data/00437/54854/ |
| Mixed layer depth | de Boyer Montégut et al. (2004) | http://www.ifremer.fr/cerweb/deboyer/mld |
| Sea surface height | CMEMS | http://marine.copernicus.eu/services-portfolio/access-to-products/ |
| Surface wind stress | Scatterometer Climatology of Ocean Winds (SCOW), Risien and Chelton (2008) | http://cioss.coas.oregonstate.edu/scow/ |
| Northward heat transport | McDonald and Baringer (2013) | Tables 29.3-29.5 |
| Zonal current at 140°W | Johnson et al. (2002) | https://floats.pmel.noaa.gov/gregory-c-johnson-home-page |
| AMOC at 26.5°N | Smeed et al. (2019) | https://www.rapid.ac.uk/rapidmoc/rapid_data/datadl.php |


**Table D1. Drift of vertically averaged temperature (°C) evaluated as the deviation of the 1980–2009 mean of the last cycle relative to the annual mean of the initial year of the simulation by each model for four depth ranges. Model(s) with the smallest drift in each simulation is emphasized with bold letters.**

| model name | 0–700 m drift (°C) | | 0–2000 m drift (°C) | | 2000 m–bottom drift (°C) | | top–bottom drift (°C) | |
|---|---|---|---|---|---|---|---|---|
| | omip1 | omip2 | omip1 | omip2 | omip1 | omip2 | omip1 | omip2 |
| AWI-FESOM | 0.20 | 0.30 | 0.17 | 0.28 | **−0.01** | 0.11 | 0.08 | 0.19 |
| CAS-LICOM3 | 0.23 | 0.33 | 0.28 | 0.33 | 0.34 | 0.29 | 0.32 | 0.31 |
| CESM-POP | 0.33 | 0.35 | −0.04 | −0.14 | −0.09 | −0.65 | −0.06 | −0.39 |
| CMCC-NEMO | 0.23 | 0.14 | −0.05 | −0.09 | −0.10 | −0.12 | −0.10 | −0.13 |
| EC-Earth3-NEMO | −0.13 | **−0.11** | −0.30 | −0.25 | −0.54 | −0.50 | −0.43 | −0.38 |
| FSU-HYCOM | 0.05 | 0.28 | −0.24 | 0.05 | −0.20 | −0.11 | −0.22 | −0.03 |
| GFDL-MOM | **0.00** | 0.15 | **0.02** | 0.12 | **0.01** | −0.08 | **−0.01** | **0.00** |
| Kiel-NEMO | 0.39 | 0.28 | 0.10 | **−0.02** | 0.11 | **0.00** | 0.11 | −0.01 |
| MIROC-COCO4.9[#] | 0.21 | 0.32 | 0.05 | 0.18 | 0.33 | 0.56 | 0.19 | 0.37 |
| MRI.COM | 0.52 | 0.68 | 0.21 | 0.35 | −0.08 | −0.13 | 0.06 | 0.10 |
| NorESM-BLOM | 0.15 | 0.41 | −0.04 | 0.16 | −0.55 | −0.50 | −0.37 | −0.24 |
| ensemble mean | 0.20 | 0.28 | 0.01 | 0.09 | −0.07 | −0.10 | −0.04 | −0.02 |
| ensemble std | 0.17 | 0.18 | 0.17 | 0.19 | 0.28 | 0.34 | 0.22 | 0.24 |

# For MIROC-COCO4.9, the fifth cycle is used for both omip-1 and omip-2.

**Table D2. Circulation metrics as 1980–2009 means of the last cycle. All metrics are in units of Sverdrups (Sv; 1 Sv = $10^9$ kg s$^{-1}$). Observational estimates at the bottom row are due to the RAPID observation (e.g., Smeed et al. 2019) for the Atlantic meridional overturning circulation (AMOC) at 26.5°N, Talley (2013) for the bottom water circulation cell of the Global meridional overturning circulation (GMOC) at 30°S, Cunningham et al. (2003) 134 ± 27 Sv and Donohue et al. (2016) 173.3 ± 10.7 Sv for the Antarctic Circumpolar Current (ACC), and Sprintall et al. (2009) for the Indonesian Through Flow (ITF).**


| model name | AMOC (Sv) omip1 | AMOC (Sv) omip2 | GMOC (Sv) omip1 | GMOC (Sv) omip2 | ACC (Sv) omip1 | ACC (Sv) omip2 | ITF (Sv) omip1 | ITF (Sv) omip2 |
|---|---|---|---|---|---|---|---|---|
| AWI-FESOM | 12.0 | 12.0 | −7.5 | −3.4 | 139.8 | 111.6 | −11.2 | −11.3 |
| CAS-LICOM3 | 17.7 | 16.3 | −1.1 | −1.5 | 127.5 | 127.4 | −7.8 | −7.0 |
| CESM-POP | 19.7 | 15.9 | −5.4 | −14.9 | 134.7 | 178.9 | −11.9 | −12.2 |
| CMCC-NEMO | 10.5 | 14.3 | −12.5 | −14.9 | 147.9 | 142.3 | −13.0 | −13.9 |
| EC-Earth3-NEMO | 15.2 | 15.9 | −14.5 | −15.0 | 197.5 | 189.1 | −13.2 | −13.9 |
| FSU-HYCOM | 10.9 | 15.8 | −8.4 | −3.9 | 162.2 | 158.5 | −14.4 | −16.4 |
| GFDL-MOM | 16.3 | 14.6 | −10.8 | −12.4 | 154.9 | 160.6 | −14.6 | −14.0 |
| Kiel-NEMO | 11.5 | 11.8 | −5.2 | −7.6 | 110.8 | 113.9 | −12.4 | −11.5 |
| MIROC-COCO4.9[#] | 16.0 | 16.2 | −7.9 | −2.7 | 161.2 | 114.7 | −13.3 | −11.7 |
| MRI.COM | 15.5 | 13.9 | −11.4 | −12.4 | 172.0 | 173.2 | −11.3 | −11.7 |
| NorESM-BLOM | 20.7 | 20.6 | −12.4 | −12.2 | 171.0 | 162.9 | −18.1 | −19.3 |
| ensemble mean | 15.1 | 15.2 | −8.8 | −9.2 | 152.7 | 148.5 | −12.8 | −13.0 |
| ensemble std | 3.3 | 2.3 | 3.8 | 5.2 | 22.9 | 26.7 | 2.4 | 3.0 |
| OBS | 18 | | −29 | | 134 − 173 | | −15 | |

# For MIROC-COCO4.9, the fifth cycle is used for both omip-1 and omip-2.

**Table D3. Root-mean-square bias and mean bias of the 30-year mean (1980–2009) sea surface temperature (°C) and salinity (psu) relative to observations (PCMDI-SST and WOA13v2, respectively) and root-mean-square bias of the 17-year mean (1993–2009) SSH (cm) relative to observations (CMEMS) for individual models. Model(s) with the smallest root-mean-square bias in each simulation is emphasized with bold letters.**

| model name | SST bias rmse (°C) | | SST bias mean (°C) | | SSS bias rmse (psu) | | SSS bias mean (psu) | | SSH bias rmse (cm) | |
|---|---|---|---|---|---|---|---|---|---|---|
| | omip1 | omip2 | omip1 | omip2 | omip1 | omip2 | omip1 | omip2 | omip1 | omip2 |
| AWI-FESOM | 0.671 | 0.675 | −0.171 | −0.205 | **0.355** | 0.314 | −0.091 | −0.099 | 10.66 | 10.75 |
| CAS-LICOM3 | 0.597 | 0.581 | 0.042 | 0.033 | 0.458 | 0.471 | 0.078 | 0.083 | 12.61 | 12.03 |
| CESM-POP | 0.577 | 0.581 | 0.073 | 0.029 | 0.494 | **0.386** | 0.054 | 0.221 | 11.74 | 11.53 |
| CMCC-NEMO | 0.578 | 0.523 | 0.053 | 0.024 | 0.597 | 0.593 | 0.106 | 0.081 | 9.20 | 10.02 |
| EC-Earth3-NEMO | 0.617 | 0.568 | 0.170 | 0.141 | 0.560 | 0.564 | −0.036 | −0.035 | 9.16 | 8.74 |
| FSU-HYCOM | 0.717 | 0.690 | 0.192 | 0.125 | 0.555 | 0.602 | 0.306 | 0.306 | 11.67 | 12.74 |
| GFDL-MOM | **0.493** | **0.467** | 0.042 | 0.027 | 0.481 | 0.408 | 0.215 | 0.205 | **8.04** | **8.42** |
| Kiel-NEMO | 0.955 | 0.874 | 0.105 | 0.042 | 1.333 | 1.117 | 0.033 | −0.008 | 10.09 | 9.83 |
| MIROC-COCO4.9 | 0.593 | 0.578 | −0.065 | −0.084 | 0.558 | 0.516 | 0.149 | 0.127 | 15.49 | 18.48 |
| MRI.COM | 0.585 | 0.568 | 0.096 | 0.102 | 0.457 | 0.428 | 0.241 | 0.276 | 11.25 | 11.82 |
| NorESM-BLOM | 0.579 | 0.572 | 0.082 | 0.034 | 0.519 | 0.568 | 0.167 | 0.188 | 10.72 | 11.38 |
| MMM | 0.491 | 0.462 | 0.062 | 0.030 | 0.348 | 0.314 | 0.106 | 0.119 | 8.52 | 8.67 |


**Table D4. Root-mean-square bias and mean bias of the 30-year mean (1980–2009) mixed layer depth (m) relative to observationally derived mixed layer depth data from de Boyer Montégut et al. (2004) in summer and winter and the maximum depth of the 30-year mean (1980–2009) winter mixed layer depth in the North Atlantic (50°−80°N; 80°W–30°E) and in the marginal seas around Antarctica (south of 60°S) for individual models. Root-mean-square bias and mean bias in winter are computed by excluding the above regions of the North Atlantic and the marginal seas around Antarctica. Model(s) with the smallest root-mean-square bias in each simulation is emphasized with bold letters.**


| model name | Winter MLD bias rmse (m) | | Winter MLD bias mean (m) | | Summer MLD bias rmse (m) | | Summer MLD bias mean (m) | | North Atlantic (m) | | Antarctica (m) | |
|---|---|---|---|---|---|---|---|---|---|---|---|---|
| | omip1 | omip2 | omip1 | omip2 | omip1 | omip2 | omip1 | omip2 | omip1 | omip2 | omip1 | omip2 |
| AWI-FESOM | 44.88 | 45.55 | 10.33 | 10.22 | 13.79 | 14.48 | −5.62 | −5.66 | 2001.7 | 2089.0 | 1539.8 | 994.4 |
| CAS-LICOM3 | 55.94 | 55.16 | −10.59 | −15.17 | 19.28 | 18.62 | −8.08 | −8.46 | 1802.0 | 1674.8 | 523.0 | 392.4 |
| CESM-POP | 35.12 | 31.56 | 11.57 | 10.59 | 11.01 | 10.17 | 1.76 | 2.19 | 1654.2 | 1527.5 | 294.7 | 1200.5 |
| CMCC-NEMO | 37.02 | **30.90** | 12.60 | 1.69 | 10.99 | 12.94 | −5.04 | −9.13 | 1011.7 | 1713.6 | 1183.4 | 1209.2 |
| EC-Earth3-NEMO | 36.71 | 32.88 | 4.98 | 3.80 | 10.94 | **9.93** | −3.32 | −1.92 | 1216.9 | 1305.0 | 1918.0 | 1465.9 |
| FSU-HYCOM | 67.69 | 80.25 | 22.62 | 34.95 | 12.92 | 12.28 | 3.09 | 4.11 | 2269.8 | 2575.6 | 4136.7 | 3368.4 |
| GFDL-MOM | **33.62** | 32.59 | −7.70 | −9.47 | **10.46** | 10.02 | −4.07 | −3.86 | 2641.7 | 2501.3 | 1749.4 | 2094.8 |
| Kiel-NEMO | 39.43 | 35.78 | 8.59 | −0.73 | 11.96 | 14.12 | −7.25 | −10.77 | 1288.3 | 1656.0 | 357.9 | 524.5 |
| MIROC-COCO4.9 | 40.73 | 38.59 | 12.75 | 6.51 | 11.46 | 9.99 | 4.64 | 2.33 | 1678.0 | 1509.1 | 3680.0 | 876.9 |
| MRI.COM | 49.95 | 48.35 | 17.86 | 15.69 | 12.06 | 11.47 | −5.68 | −5.08 | 2270.0 | 1414.8 | 4764.8 | 4846.1 |
| NorESM-BLOM | 45.46 | 46.86 | 19.53 | 20.96 | 14.64 | 14.97 | −0.07 | 1.85 | 2150.9 | 2141.8 | 2500.3 | 1459.8 |
| MMM | 33.08 | 30.93 | 8.92 | 6.74 | 10.43 | 9.55 | −2.82 | −3.24 | - | - | - | - |
| ensemble mean | - | - | - | - | - | - | - | - | 1816.8 | 1828.0 | 2058.9 | 1675.7 |

**Table D5. Root-mean-square bias of the 30-year mean (1980–2009) basin-wide averaged zonal mean temperature (°C) relative to observations (WOA13v2) for individual models. Model(s) with the smallest root-mean-square bias in each simulation is emphasized with bold letters.**

| | ZMT rmse Southern (°C) | | ZMT rmse Atlantic (°C) | | ZMT rmse Indian (°C) | | ZMT rmse Pacific (°C) | |
|---|---|---|---|---|---|---|---|---|
| model name | omip1 | omip2 | omip1 | omip2 | omip1 | omip2 | omip1 | omip2 |
| AWI-FESOM | 0.43 | 0.56 | 0.69 | 0.74 | 0.46 | **0.41** | 0.46 | 0.45 |
| CAS-LICOM3 | 0.73 | 0.65 | 1.51 | 1.44 | 0.71 | 0.74 | 0.55 | 0.54 |
| CESM-POP | 0.31 | 0.98 | 0.86 | 0.67 | 0.91 | 0.91 | 0.49 | 0.81 |
| CMCC-NEMO | 0.23 | **0.20** | 0.68 | 0.60 | 0.80 | 0.82 | 0.41 | 0.40 |
| EC-Earth3-NEMO | 0.63 | 0.63 | 1.00 | 1.00 | 1.02 | 1.01 | 0.75 | 0.71 |
| FSU-HYCOM | 0.40 | 0.41 | 0.86 | 1.10 | 1.31 | 1.21 | 0.54 | 0.48 |
| GFDL-MOM | **0.22** | 0.24 | 0.65 | 0.70 | **0.44** | 0.57 | **0.28** | **0.26** |
| Kiel-NEMO | 0.53 | 0.40 | **0.53** | **0.56** | 0.93 | 0.99 | 0.50 | 0.46 |
| MIROC-COCO4.9 | 0.29 | 0.85 | 0.72 | 0.99 | 1.00 | 1.08 | 0.47 | 0.56 |
| MRI.COM | 0.37 | 0.48 | 0.85 | 0.94 | 0.91 | 0.89 | 0.46 | 0.52 |
| NorESM-BLOM | 1.07 | 1.09 | 0.78 | 0.88 | 0.79 | 0.87 | 1.04 | 0.98 |
| MMM | 0.17 | 0.21 | 0.52 | 0.60 | 0.69 | 0.66 | 0.36 | 0.34 |


**Table D6. Same as Table D5, but for zonal mean salinity (psu).**

| model name | ZMS rmse Southern (psu) | | ZMS rmse Atlantic (psu) | | ZMS rmse Indian (psu) | | ZMS rmse Pacific (psu) | |
|---|---|---|---|---|---|---|---|---|
| | omip1 | omip2 | omip1 | omip2 | omip1 | omip2 | omip1 | omip2 |
| AWI-FESOM | 0.050 | 0.051 | 0.144 | 0.153 | **0.113** | **0.100** | 0.073 | 0.065 |
| CAS-LICOM3 | 0.064 | 0.057 | 0.231 | 0.213 | 0.131 | 0.141 | 0.087 | 0.082 |
| CESM-POP | 0.053 | 0.090 | 0.133 | 0.139 | 0.160 | 0.140 | **0.051** | 0.060 |
| CMCC-NEMO | 0.050 | 0.053 | 0.165 | 0.156 | 0.153 | 0.137 | 0.060 | 0.057 |
| EC-Earth3-NEMO | 0.057 | 0.057 | 0.130 | 0.128 | 0.150 | 0.157 | 0.087 | 0.081 |
| FSU-HYCOM | 0.083 | 0.078 | 0.159 | 0.178 | 0.241 | 0.244 | 0.077 | 0.065 |
| GFDL-MOM | 0.040 | 0.048 | **0.085** | **0.108** | 0.116 | 0.146 | **0.051** | **0.054** |
| Kiel-NEMO | **0.037** | **0.045** | 0.128 | 0.131 | 0.164 | 0.169 | 0.073 | 0.066 |
| MIROC-COCO4.9 | 0.057 | 0.049 | 0.089 | 0.114 | 0.177 | 0.180 | 0.080 | 0.074 |
| MRI.COM | 0.080 | 0.111 | 0.144 | 0.176 | 0.178 | 0.179 | 0.059 | 0.074 |
| NorESM-BLOM | 0.097 | 0.116 | 0.129 | 0.147 | 0.118 | 0.134 | 0.078 | 0.083 |
| MMM | 0.036 | 0.049 | 0.090 | 0.106 | 0.123 | 0.117 | 0.047 | 0.042 |

**Table D7. The 30-year mean (1980–2009) sea-ice extent ($10^6$ km$^2$) in both hemispheres in winter and summer for individual models. Observational estimates are due to National snow and ice data center Sea Ice Index (NSIDC-SII; Fetterer et al. 2017).**

| model name | SIE NH Mar ($10^6$ km$^2$) | | SIE NH Sep ($10^6$ km$^2$) | | SIE SH Sep ($10^6$ km$^2$) | | SIE SH Mar ($10^6$ km$^2$) | |
|---|---|---|---|---|---|---|---|---|
|  | omip1 | omip2 | omip1 | omip2 | omip1 | omip2 | omip1 | omip2 |
| AWI-FESOM | 15.53 | 15.71 | 7.37 | 7.47 | 19.10 | 18.34 | 2.36 | 4.11 |
| CAS-LICOM3 | 17.09 | 16.84 | 4.79 | 5.47 | 24.92 | 23.01 | 2.12 | 3.22 |
| CESM-POP | 14.93 | 14.88 | 3.84 | 5.30 | 18.15 | 16.05 | 1.41 | 2.19 |
| CMCC-NEMO | 15.65 | 15.46 | 3.88 | 4.84 | 18.63 | 17.44 | 2.29 | 2.74 |
| EC-Earth3-NEMO | 18.12 | 17.57 | 8.43 | 7.85 | 22.99 | 20.63 | 4.53 | 4.32 |
| FSU-HYCOM | 13.11 | 12.82 | 4.75 | 5.58 | 14.88 | 14.90 | 0.97 | 1.53 |
| GFDL-MOM | 14.30 | 14.20 | 6.24 | 6.10 | 17.75 | 16.48 | 1.95 | 3.31 |
| Kiel-NEMO | 14.48 | 14.67 | 8.04 | 7.44 | 18.66 | 17.92 | 2.35 | 4.14 |
| MIROC-COCO4.9 | 13.62 | 13.49 | 6.37 | 6.33 | 18.89 | 17.98 | 0.60 | 1.94 |
| MRI.COM | 15.62 | 15.51 | 7.40 | 7.36 | 18.11 | 16.95 | 1.71 | 2.04 |
| NorESM-BLOM | 15.04 | 14.87 | 5.60 | 6.32 | 19.54 | 18.06 | 1.86 | 3.26 |
| ensemble mean | 15.22 | 15.09 | 6.07 | 6.37 | 19.24 | 17.98 | 2.01 | 2.98 |
| ensemble std | 1.32 | 1.25 | 1.48 | 0.94 | 2.43 | 2.03 | 0.92 | 0.88 |
| OBS | 15.46 | | 6.51 | | 18.49 | | 4.01 | |

**Figures**

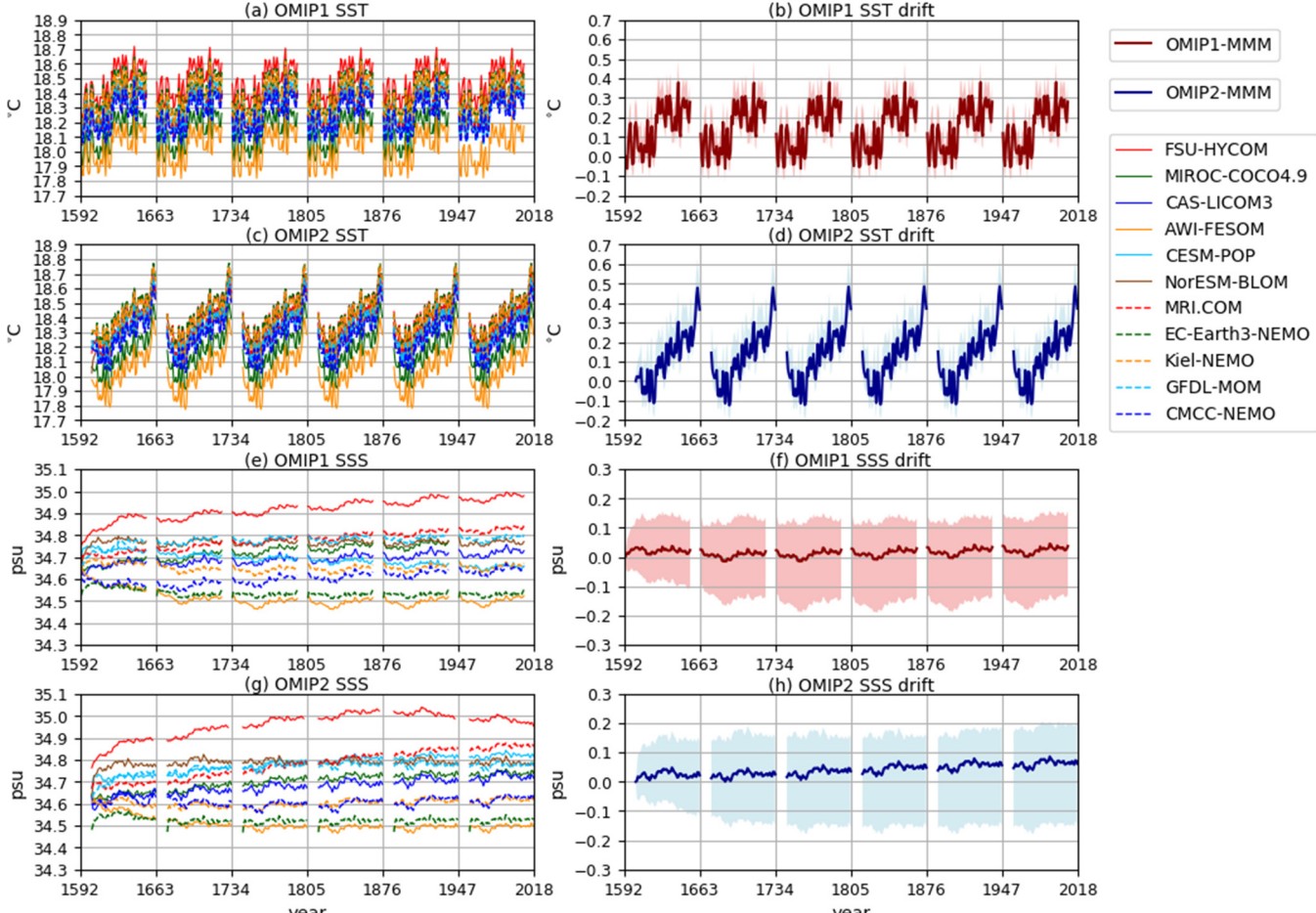

**Figure 1: Drift of annual mean, global mean sea surface temperature (units in °C) and salinity (units in practical salinity units (psu)). Sea surface temperature for (a) OMIP-1 and (c) OMIP-2. Sea surface salinity for (e) OMIP-1 and (g) OMIP-2. (b, d, f, h) Multi-model ensemble mean (lines) of deviations from the annual mean of the initial year of the simulation by each model and spread defined as the range between maximum and minimum (shades) for (b) OMIP-1 and (d) OMIP-2 sea surface temperature and (f) OMIP-1 and (h) OMIP-2 sea surface salinity. The spin-up behavior of the multi-model ensemble mean in Figs. 1 to 5 is based on the following eight (8) models which performed the full 6-cycle simulations for both OMIP-1 (6 x 62 years) and OMIP-2 (6 x 61 years): AWI-FESOM, CAS-LICOM3, CESM-POP, CMCC-NEMO, EC-Earth3-NEMO, Kiel-NEMO, MRI.COM, NorESM-BLOM. See Fig. 21 for a closer look at sea surface temperature of the last cycle from individual models.**

## Vertically averaged temperature

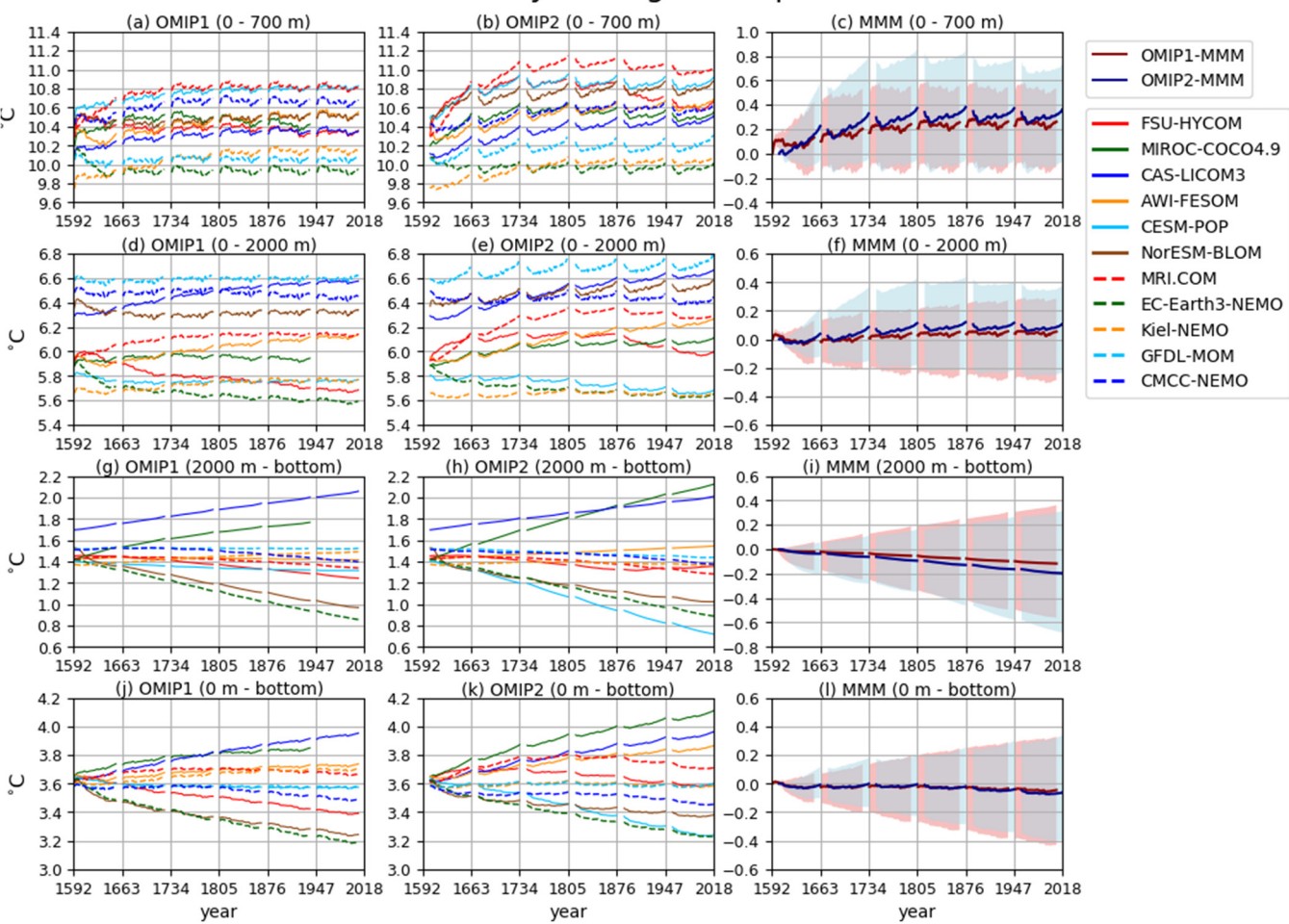

**Figure 2: Drift of annual mean, global mean vertically averaged temperatures (units in °C) for four depth ranges (a-c) 0 – 700m, (d-f) 0 – 2000m, (g-i) 2000m – bottom, (j-l) 0 m – bottom. (a, d, g, j) OMIP1 and (b, e, h, k) OMIP2. (c, f, i, l) Multi-model ensemble mean (lines) of deviations from the annual mean of the initial year of the simulation by each model and spread defined as the range between maximum and minimum (shades) of OMIP-1 (red) and OMIP-2 (blue). See Figs. S1 and S2 for a closer look at individual models.**


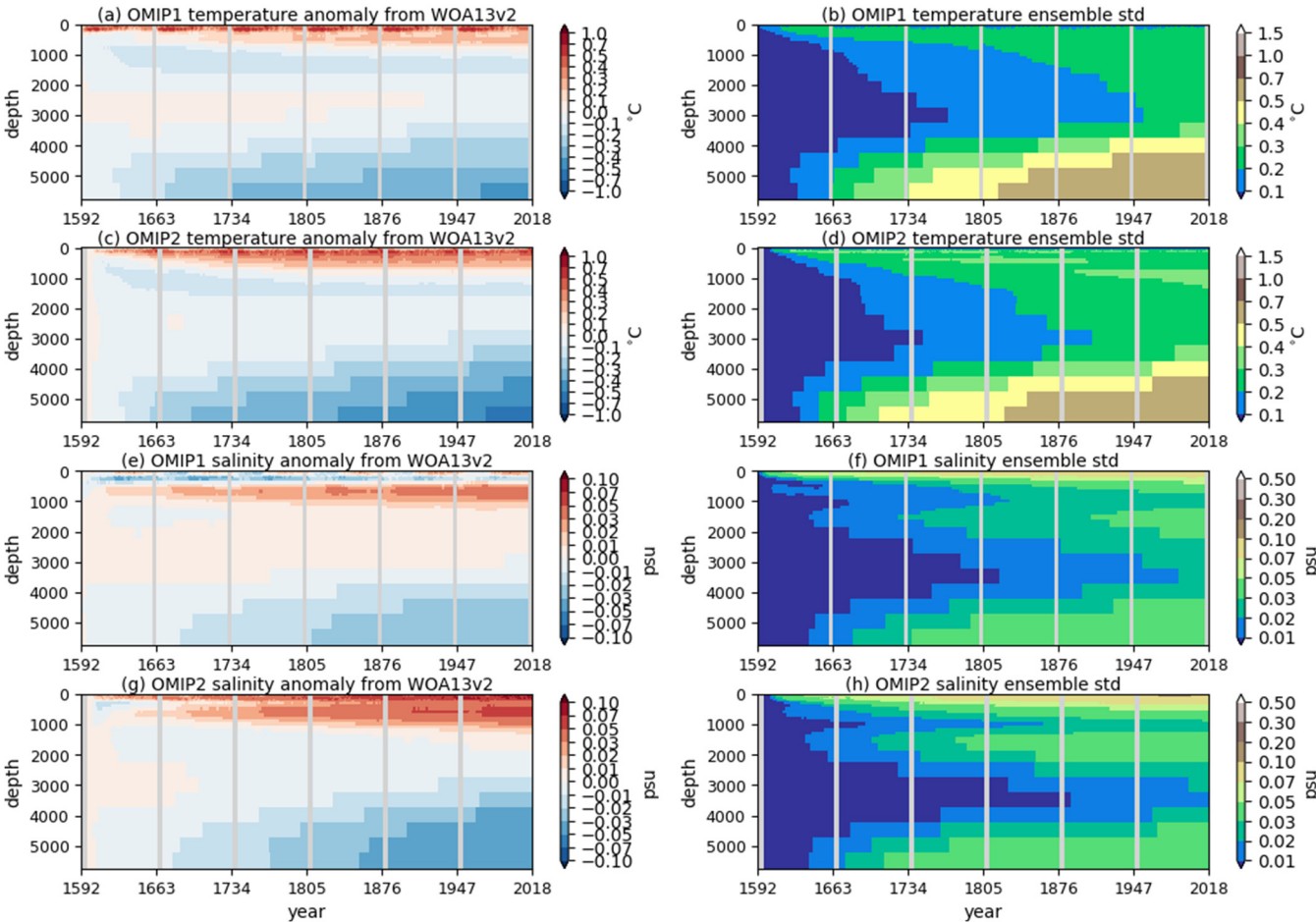

**Figure 3:** Globally averaged drift of multi-model mean horizontal mean (a, c) temperature (°C) and (e, g) salinity (psu) as a function of depth and time. The drift is defined as the deviation from the annual mean of the initial year of the simulation by each model. For each, time evolution of the standard deviation of the model ensemble is depicted to the right. (a, b) OMIP-1 temperature, (c, d) OMIP-2 temperature, (e, f) OMIP-1 salinity, and (g, h) OMIP-2 salinity. See Figs. S3 through S6 for results of individual models.


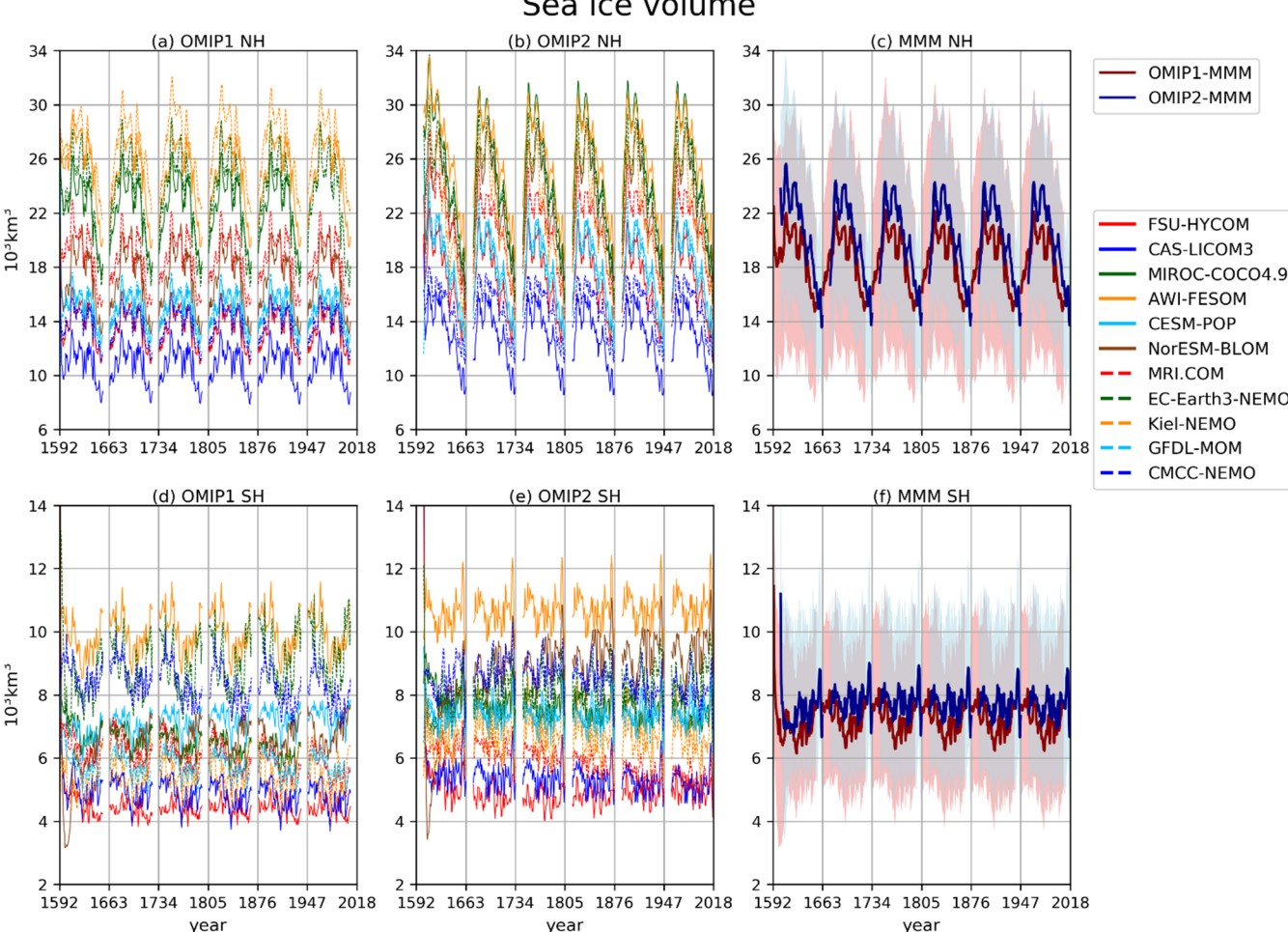

**Figure 4: Time series of annual mean sea ice volume integrated over the northern hemisphere (upper panels) and the southern hemisphere (lower panels). (a, d) OMIP-1 and (b, e) OMIP-2. (c, f) Multi-model mean (lines) and spread defined as the range between maximum and minimum (shades) of OMIP-1 (red) and OMIP-2 (blue). Units are $10^3$ km$^3$. See Fig. S7 for a closer look at individual models.**



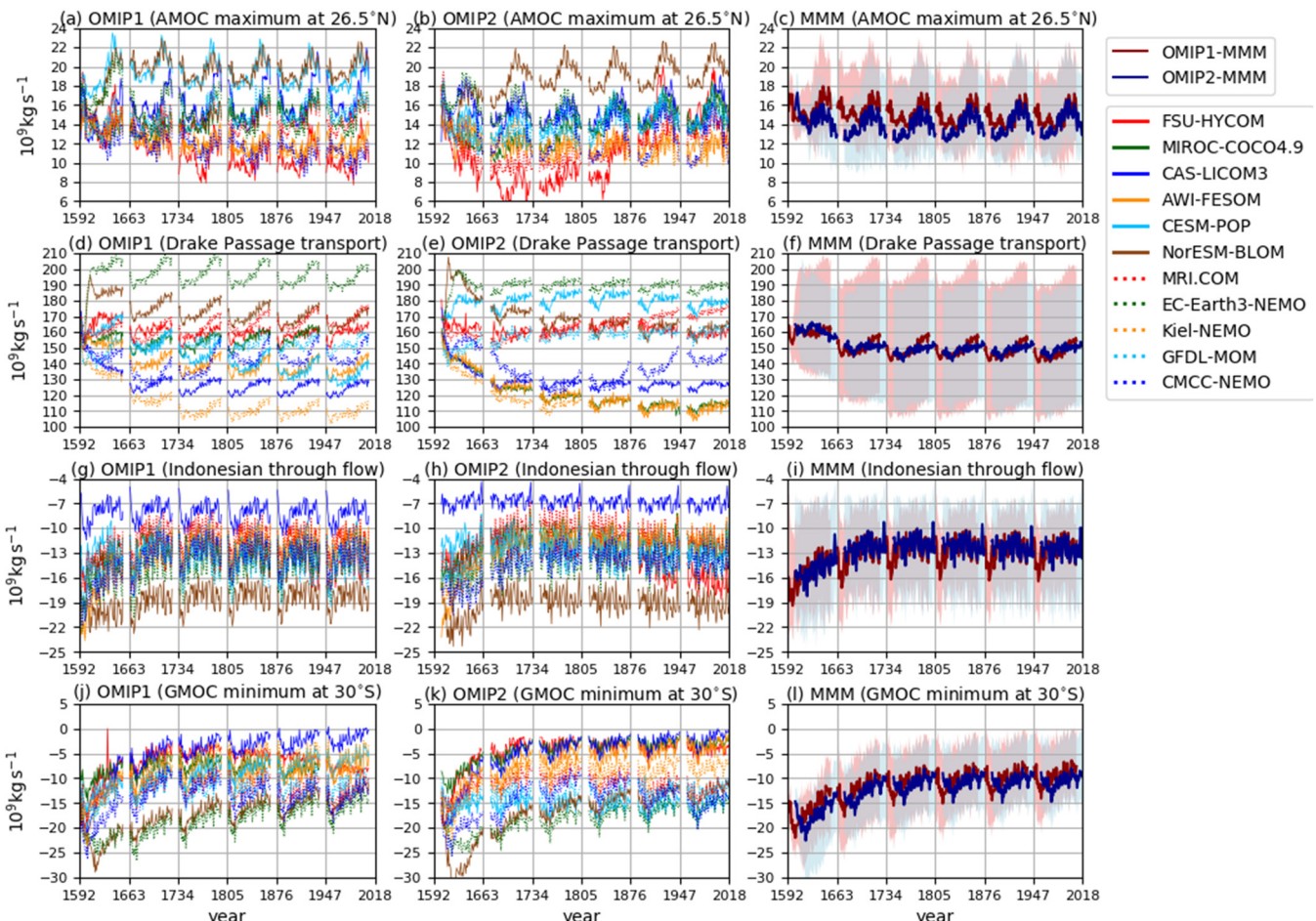

**Figure 5: Time series of annual mean ocean circulation metrics. (a-c) Atlantic meridional overturning circulation (AMOC) maximum at 26.5°N, which approximately represents the strength of AMOC associated with the North Atlantic Deep Water formation. (d-f) Drake passage transport (positive eastward), which represents the strength of Antarctic Circumpolar Current. (g-i) Indonesian Throughflow (negative into the Indian Ocean), which represents water exchange between the Pacific and Indian Ocean. (j-l) Global meridional overturning circulation (GMOC) minimum in 2000 m – bottom depths at 30°S, which represents the strength of deep to bottom layer GMOC associated with the Antarctic Bottom Water and Lower Circumpolar Deep Water formation. (a,d,g,j) OMIP-1 and (b,e,h,k) OMIP-2. (c,f,i,l) Multi-model mean (lines) and spread defined as the range between maximum and minimum (shades) of OMIP-1 (red) and OMIP-2 (blue). Units are $10^9$ kg s$^{-1}$. See Figs. S8 and S9 for a closer look at individual models.**

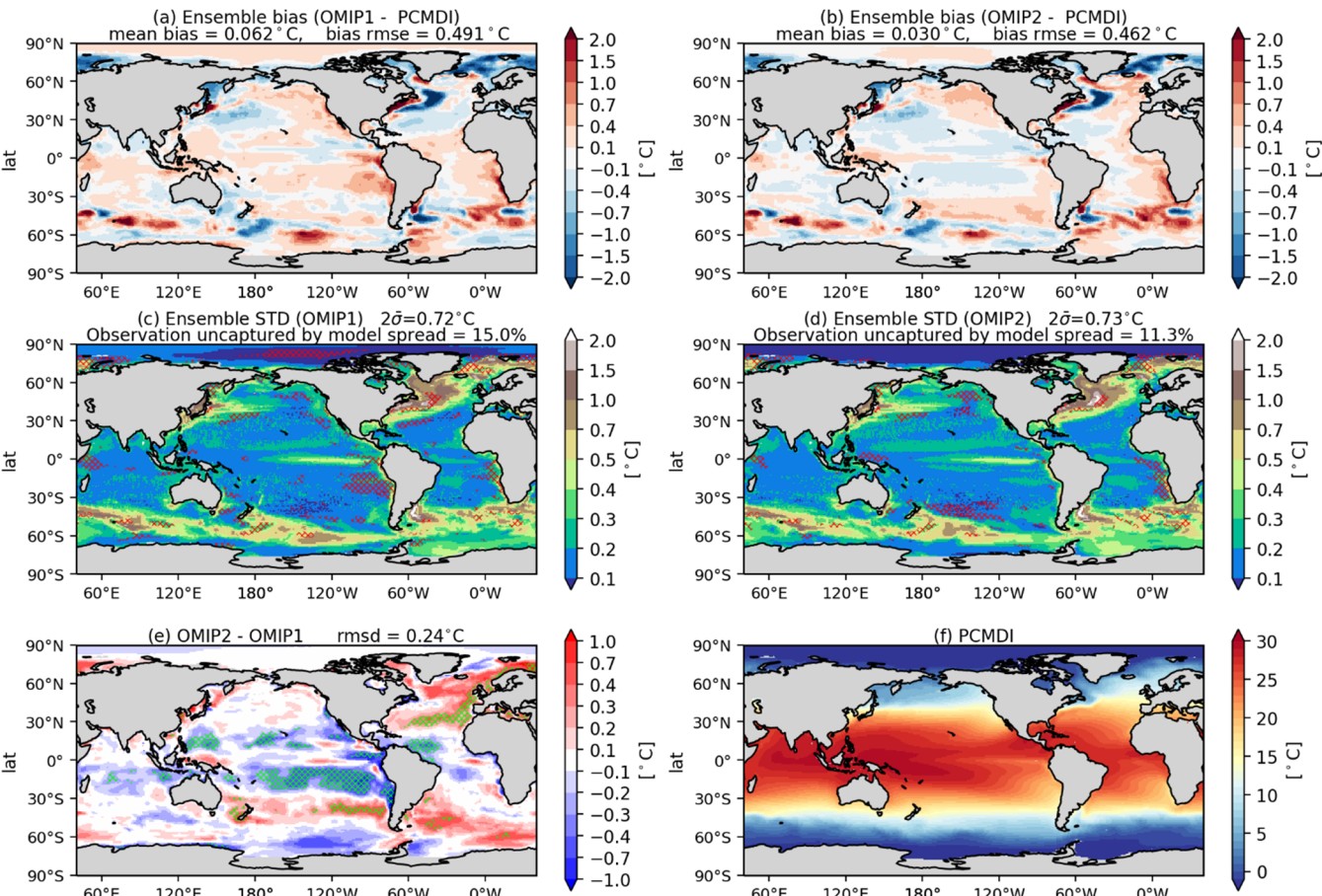

**Figure 6: Evaluation of the simulated mean sea surface temperature (SST; units in °C). Upper two panels show the bias of the multi-model mean, 30-year (1980–2009) mean SST relative to an observational estimate provided and updated by Program for Climate Model Diagnosis and Intercomprison (PCMDI) following a procedure described by Hurrell et al. (2008) (hereafter referred to as PCMDI-SST). (a) OMIP-1 and (b) OMIP-2, with global mean bias and global root-mean-square bias depicted on the top. The middle two panels show the standard deviation of the ensemble, with the regions where the observation is outside the 95% confidence range of the model spread (±2σ) hatched with red. (c) OMIP-1 and (d) OMIP-2, with the global mean confidence range (twice the standard deviation) and the fraction of the region where observation is uncaptured by the model confidence range depicted on the top. (e) Difference between OMIP-1 and OMIP-2 (OMIP-2 minus OMIP-1), with the global root-mean-square difference depicted on the top. The regions where the difference is significant at 95% confidence level are hatched with green, with the uncertainty of multi-model mean difference computed based on the method proposed by Wakamatsu et al. (2017). (f) 30-year (1980–2009) mean SST of PCMDI-SST. In the following figures, all models are used for multi-model mean. See Figs. S10 through S12 for results of individual models.**

## Multi Model Mean SSS (ave. from 1980 to 2009)

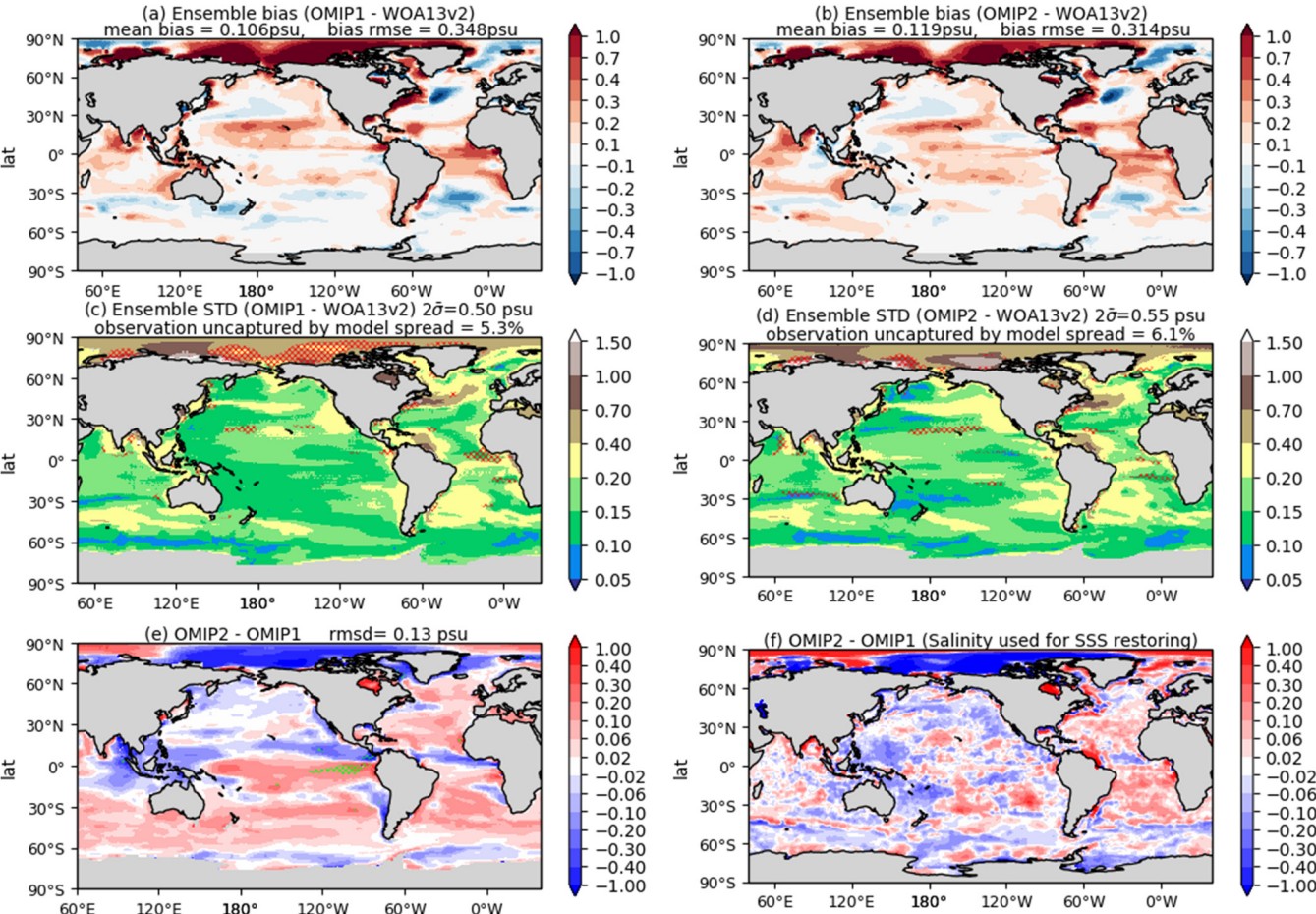

**Figure 7: Evaluation of simulated sea surface salinity (SSS; units in psu). Upper two panels show the bias of the multi-model mean 30-year (1980–2009) mean SSS relative to WOA13v2 (Zweng et al. 2013). (a) OMIP-1 and (b) OMIP-2. The middle two panels show the standard deviation of the ensemble, with the regions where the observation is outside the 95% confidence range of the model spread (±2σ) hatched with red. (c) OMIP-1 and (d) OMIP-2. (e) Difference between OMIP-1 and OMIP-2 (OMIP-2 minus OMIP-1), with the regions where the difference is significant at 95% confidence level hatched with green as in Fig. 6. (f) Difference of salinity to which sea surface salinity is restored in OMIP-1 and OMIP-2 (OMIP-2 minus OMIP-1). On the top of each panel, global mean values are depicted as in Fig. 6. See Figs. S13 through S15 for results of individual models.**

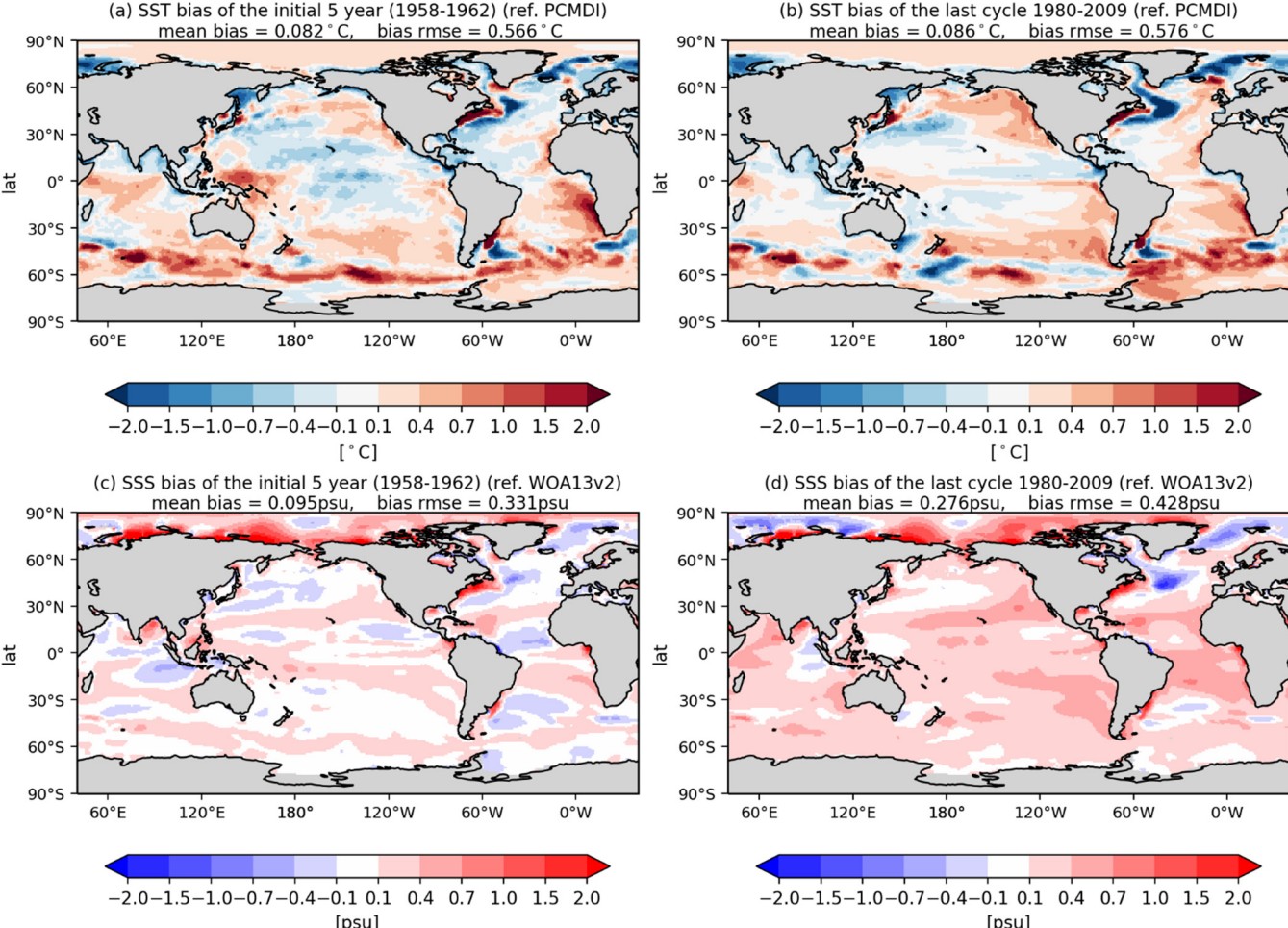

**Figure 8:** Comparison of SST (a,b) and SSS (c,d) biases relative to observations (PCMDI-SST and WOA13v2, respectively) for the initial 5-year mean (left panels) and the long-term mean (1980–2009) in the last cycle (right panels) from the OMIP-2 simulation of MRI.COM. Pattern correlation of biases between the initial 5 year mean and the long-term mean in the last cycle is 0.75 for SST and 0.85 for SSS.

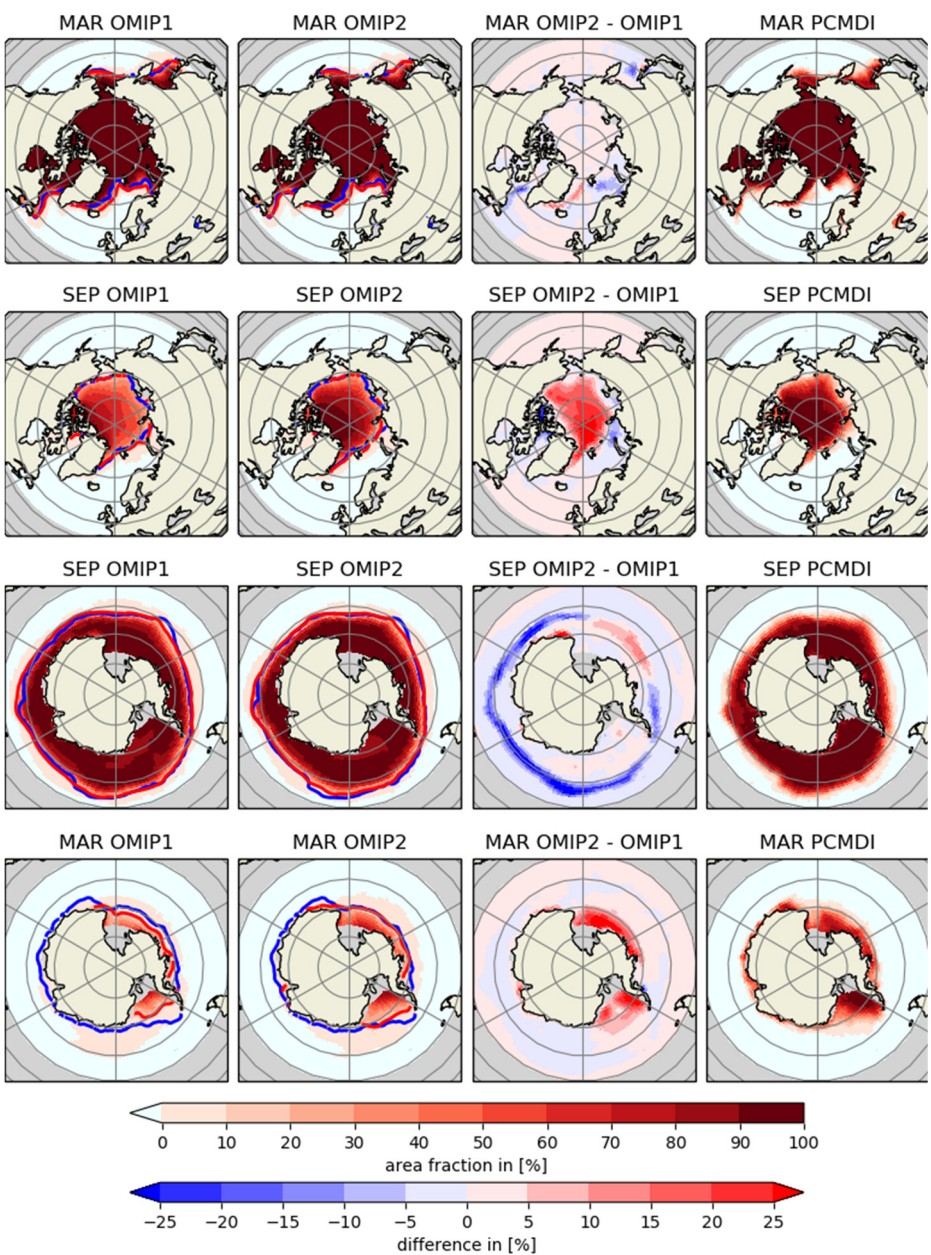

**Figure 9: Multi-model mean 30-year (1980–2009) mean sea ice concentration (%). Columns are (from the left) OMIP-1, OMIP-2, OMIP-2 − OMIP-1, and an observational dataset provided by PCMDI-SST. Rows are (from the top) March and September in the Northern hemisphere, and September and March in the Southern hemisphere. Blue lines are contours of 15% concentration of the PCMDI-SST dataset and red lines are those of multi-model mean. See Figs. S16 through S23 for results of individual models.**

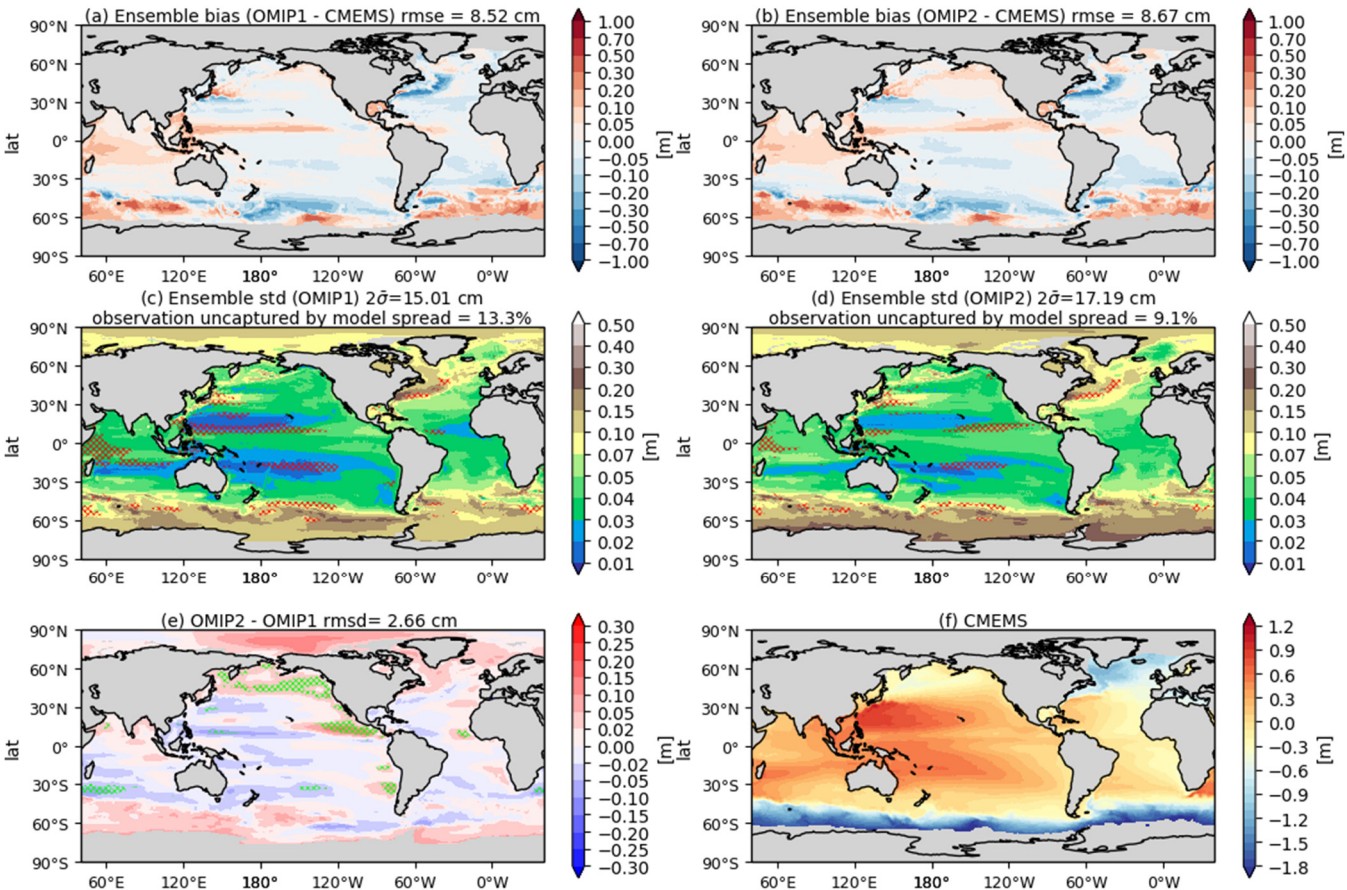

Figure 10: Evaluation of simulated sea surface height (m). Upper two panels show the bias of the multi-model mean, 17-year (1993–2009) mean SSH relative to CMEMS. (a) OMIP-1 and (b) OMIP-2. The middle two panels show the standard deviation of the ensemble, with the regions where the observation is outside the 95% confidence range of the model spread (±2σ) hatched with red. (c) OMIP-1 and (d) OMIP-2. (e) Difference between OMIP-1 and OMIP-2 (OMIP-2 minus OMIP-1), with the regions where the difference is significant at 95% confidence level hatched with green as in Fig. 6. (f) Annual mean SSH of CMEMS. Note that all SSH fields are offset by subtracting their respective quasi-global mean values before evaluation as described in Appendix C. On the top of each panel, global mean values are depicted as in Fig. 6. See Figs. S24 through S26 for results of individual models.

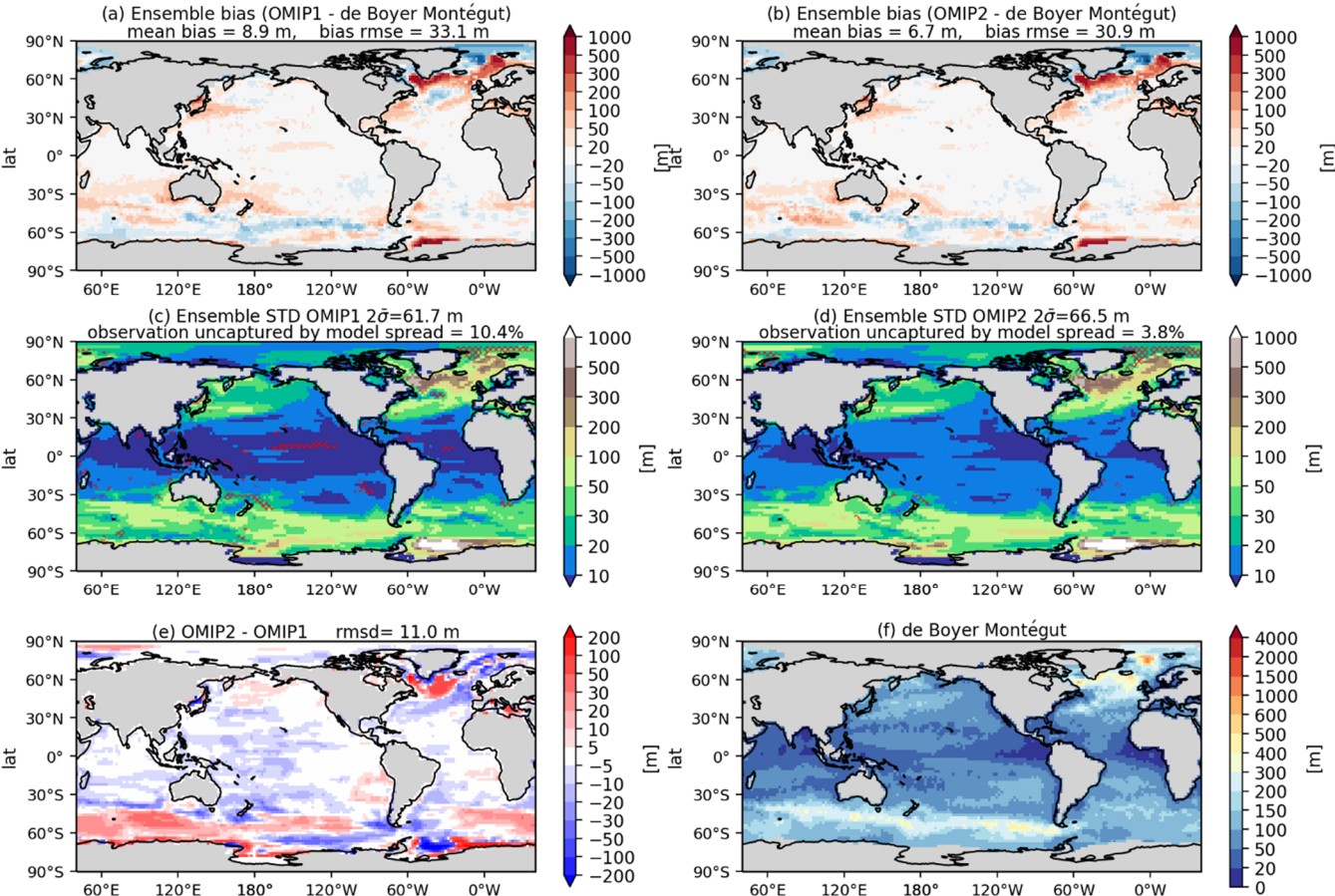

**Figure 11: Evaluation of simulated mixed layer depth (m). Upper two panels show the bias of the multi-model mean, 30-year (1980–2009) mean winter mixed layer depth in both hemispheres relative to observationally derived mixed layer depth data from de Boyer Montégut et al. (2004). January-February-March mean for the northern hemisphere and July-August-September mean for the southern hemisphere. (a) OMIP-1 and (b) OMIP-2. The middle two panels show the standard deviation of the ensemble, with the regions where the observation is outside the 95% confidence range of the model spread (±2σ) hatched with red. (c) OMIP-1 and (d) OMIP-2. (e) Difference between OMIP-1 and OMIP-2 (OMIP-2 minus OMIP-1), which is not statistically significant at 95% confidence level everywhere. (f) Observationally derived mixed layer depth data from de Boyer Montégut et al. (2004). On the top of each panel, global mean values are depicted as in Fig. 6. Note that the regions where mixed layer depths could reach more than 1000 meters in winter, specifically the marginal seas around Antarctica (south of 60°S) and the high latitude North Atlantic (50°−80°N; 80°W−30°E) are excluded from the computation of global means. See Figs. S27 through S29 for results of individual models.**

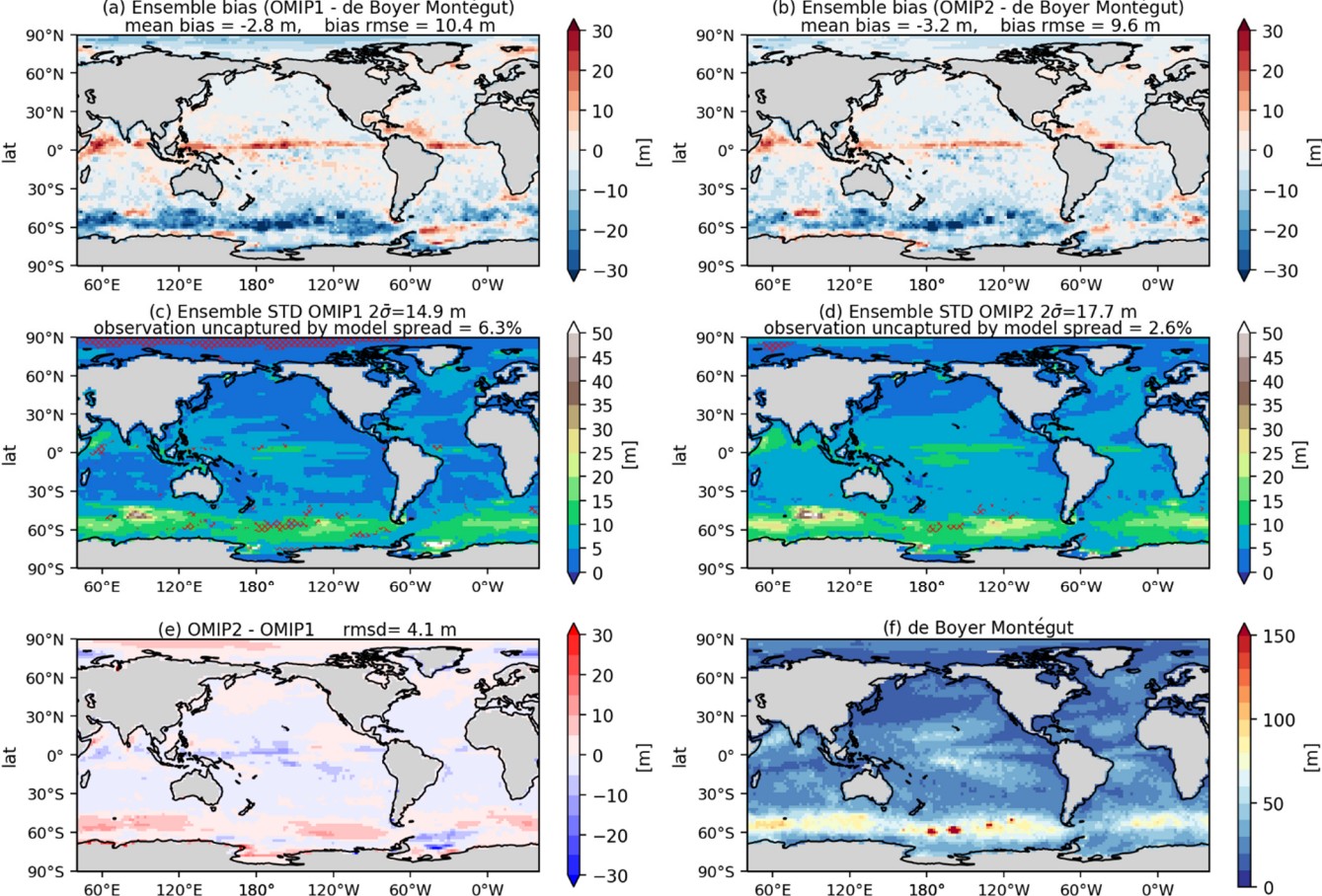

**Figure 12: Same as Fig. 11 except for summer: July-August-September mean for the northern hemisphere and January-February-March mean for the southern hemisphere. The difference between OMIP-1 and OMIP-2 is not statistically significant at 95% confidence level everywhere. On the top of each panel, global mean values are depicted as in Fig. 6. For summer, the entire oceanic region is used to evaluate global means. See Figs. S30 through S32 for results of individual models.**

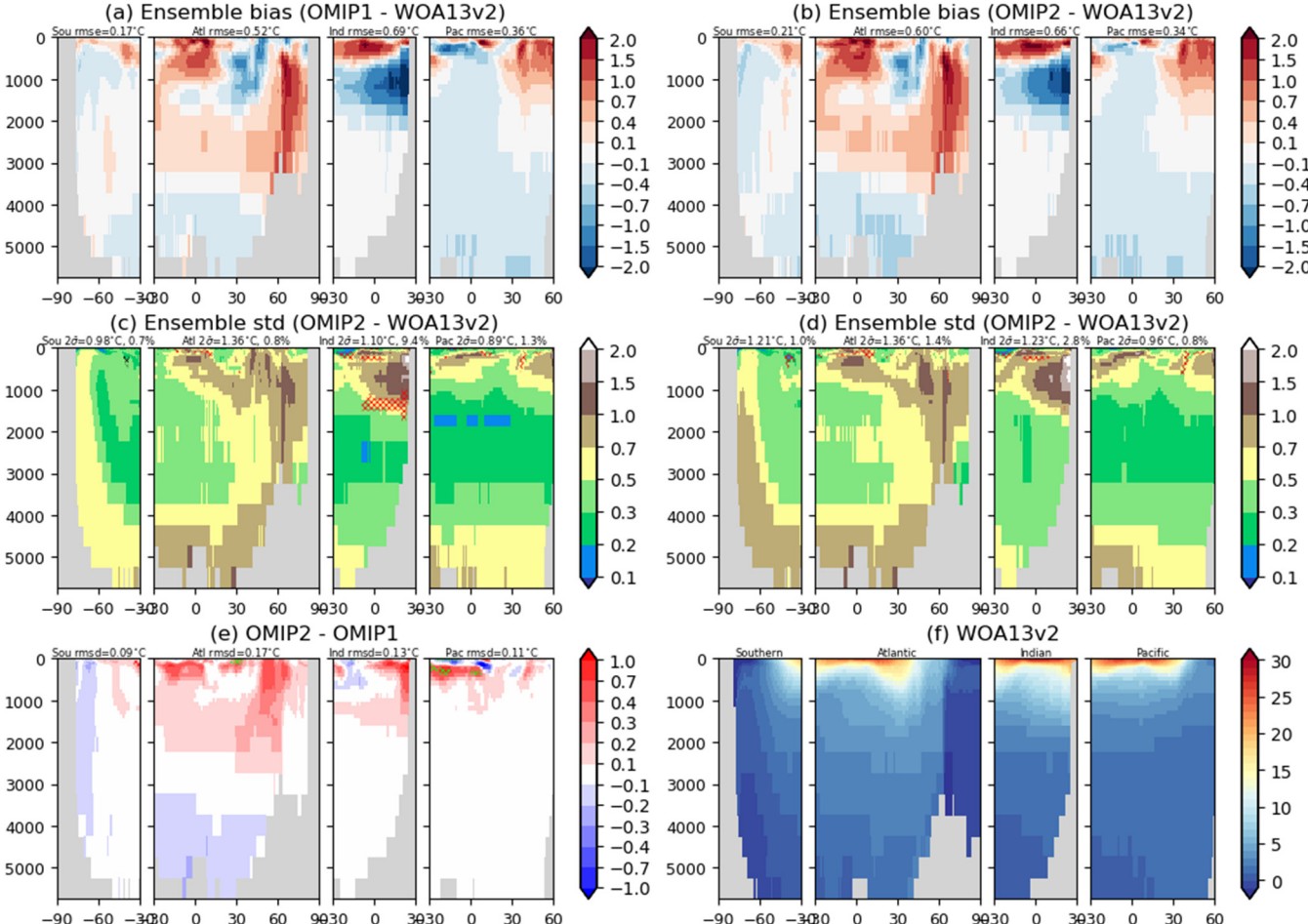

**Figure 13: Upper two panels show biases of multi-model mean, 30-year (1980–2009) mean basin-wide zonally averaged temperature of the last cycle relative to WOA13v2 (Locarnini et al. 2013). (a) OMIP-1 and (b) OMIP-2, with the basin mean root-mean-square biases depicted on the top. Middle two panels show the standard deviations of the ensemble, with the regions where the observation is outside the 95% confidence range of the model spread (±2σ) hatched with red. (c) OMIP-1 and (d) OMIP-2, with the basin mean confidence range (twice the standard deviation) and the fraction of the region where observation is uncaptured by the model confidence range depicted on the top. (e) Difference of 30-year (1980–2009) mean basin-wide zonal mean temperature between OMIP-2 and OMIP-1 (OMIP-2 minus OMIP-1), with the basin mean root-mean-square difference depicted on the top. The regions where the difference is significant at 95% confidence level are hatched with green as in Fig. 6. (f) Basin-wide zonal mean temperature of WOA13v2. Units are °C. See Figs. S33 through S35 for results of individual models.**

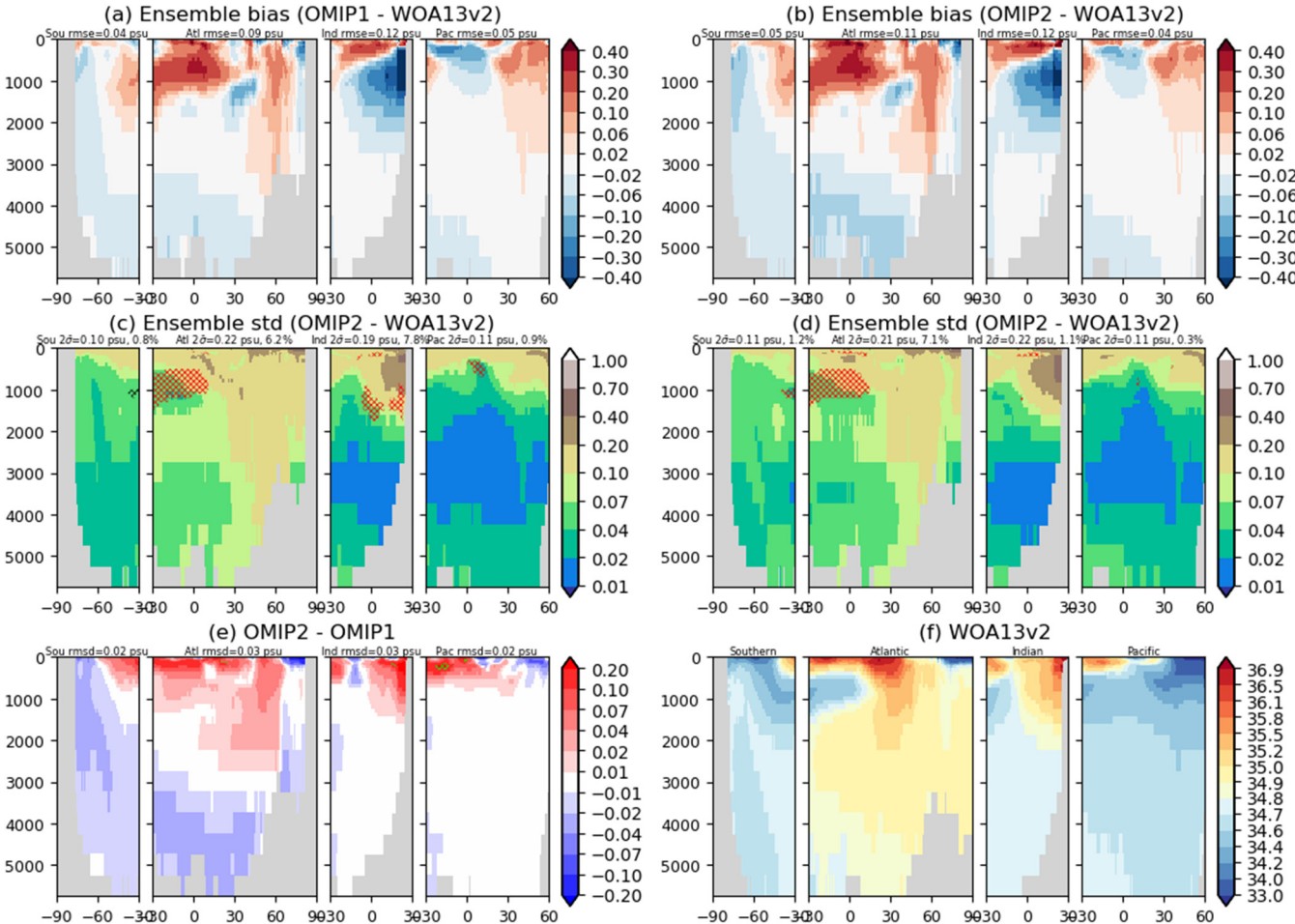

**Figure 14:** Upper two panels show biases of multi-model mean, 30-year (1980–2009) mean basin-wide zonally averaged salinity of the last cycle relative to WOA13v2 (Zweng et al. 2013) for (a) OMIP-1 and (b) OMIP-2. Middle two panels show the standard deviation of the ensemble, with the regions where the observation is outside the 95% confidence range of the model spread (±2σ) hatched with red. (c) OMIP-1 and (d) OMIP-2. (e) Difference of 30-year (1980–2009) mean basin-wide zonal mean salinity between OMIP-2 and OMIP-1 (OMIP-2 minus OMIP-1), with the regions where the difference is significant at 95% confidence level hatched with green as in Fig. 6. (f) Basin-wide zonal mean salinity of WOA13v2. Units are psu. On the top of each panel, basin mean values are depicted as in Fig. 13. See Figs. S36 through S38 for results of individual models.

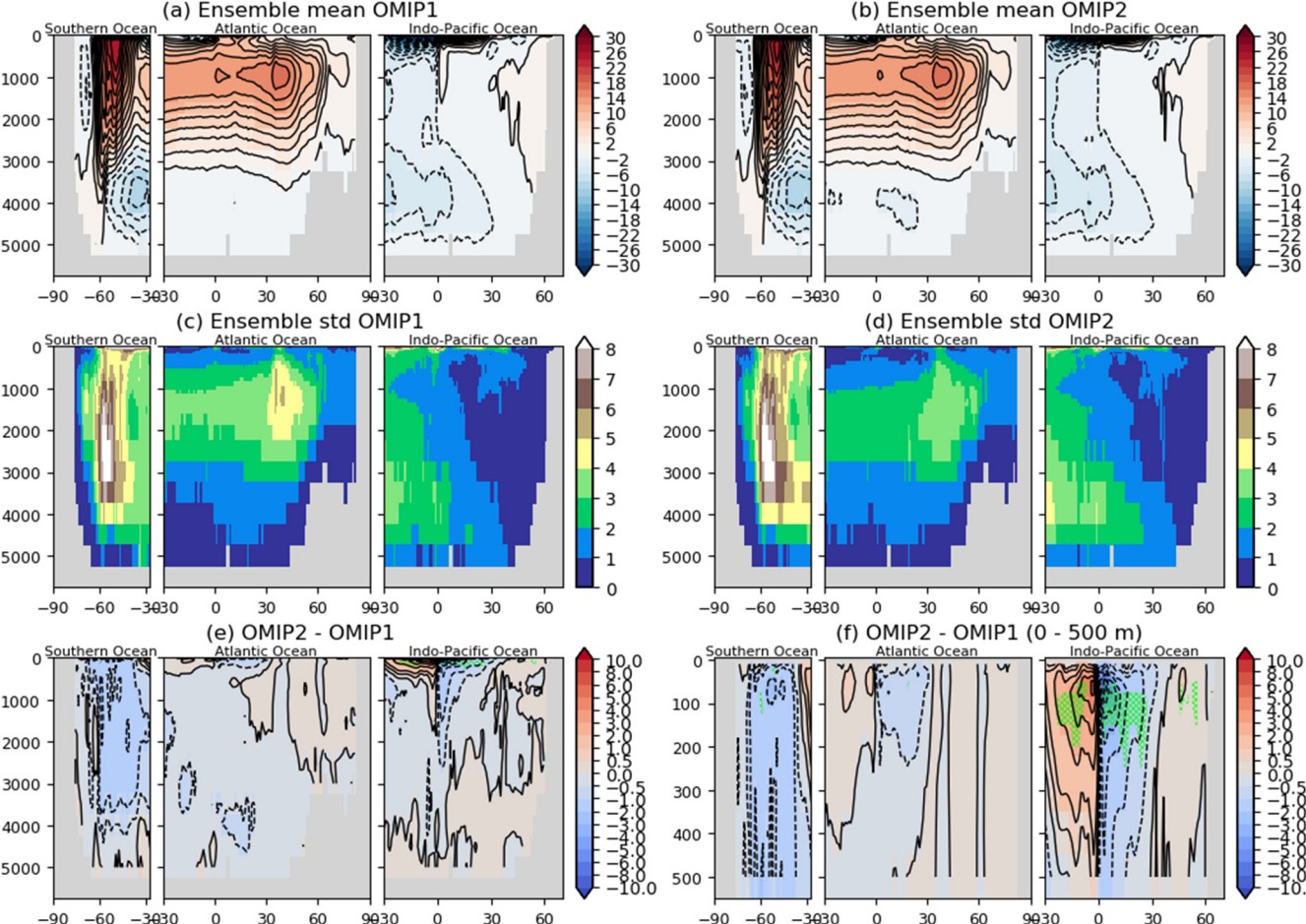

**Figure 15: Upper two panels show multi-model mean, 30-year (1980–2009) mean meridional overturning stream function in three oceanic basins. Clockwise circulations are implied around the positive extremes and vice versa. (a) OMIP-1 and (b) OMIP-2. Middle two panels show the standard deviation of the ensemble. (c) OMIP-1 and (d) OMIP-2. (e) Difference between OMIP-2 and OMIP-1 (OMIP-2 minus OMIP-1). (f) Same as (e) but for the upper 500 m depth. Units are $10^9$ kg s$^{-1}$. In (e) and (f), the regions where the difference is significant at 95% confidence level are hatched with green as in Fig. 6. See Figs. S39 through S41 for results of individual models.**


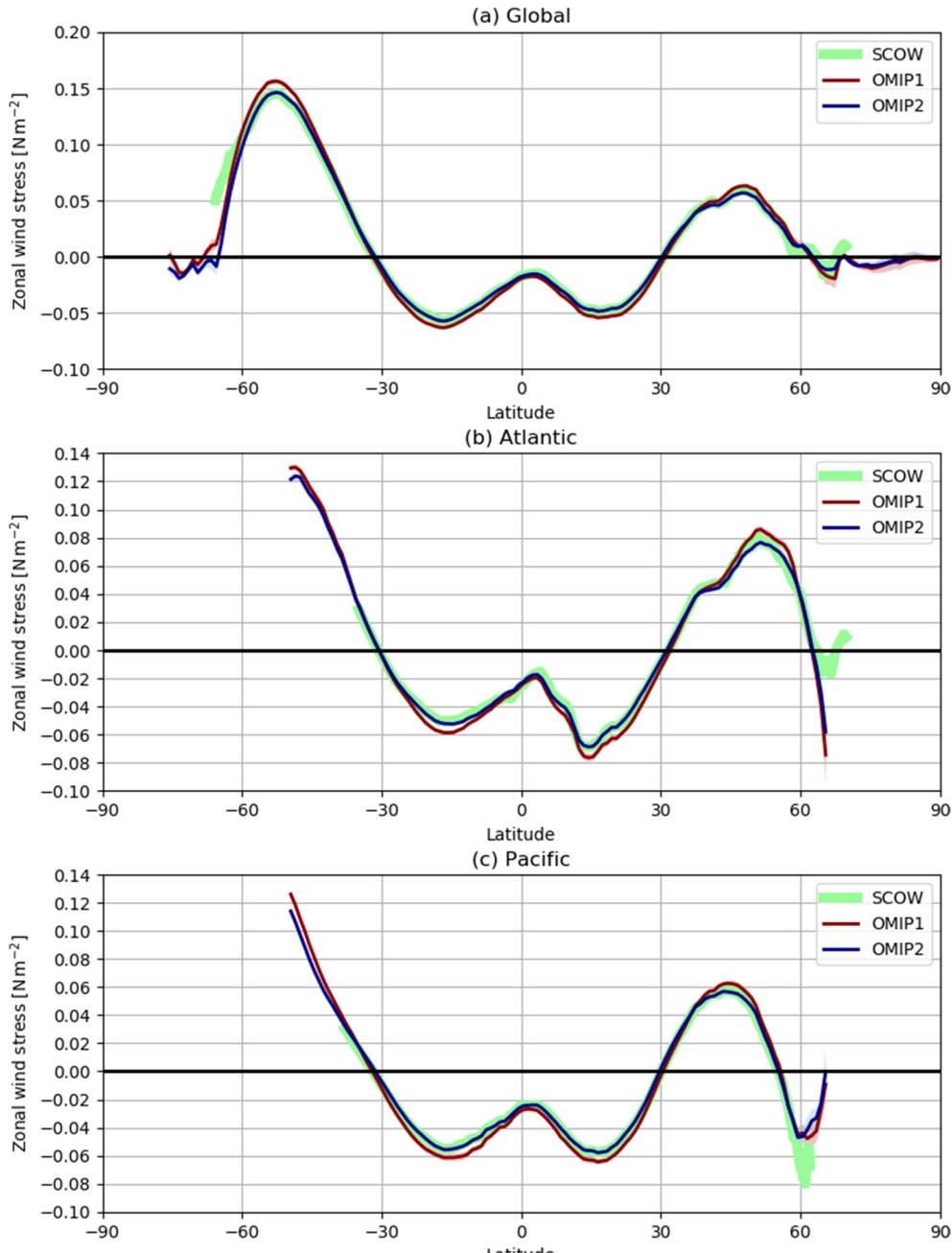


**Figure 16: Multi-model mean, 10-year (Nov1999–Oct2009) mean basin-wide averaged zonal wind stress (N m$^{-2}$). (a) Global ocean, (b) Atlantic Ocean, and (c) Pacific Ocean. Multi model mean (lines) and spread defined as one standard deviation of the ensemble (shades) of OMIP-1 (red) and OMIP-2 (blue). Note that model spread is very small. Green bold lines are Scatterometer Climatology of Ocean Winds (SCOW) provided by Risien and Chelton (2008).**


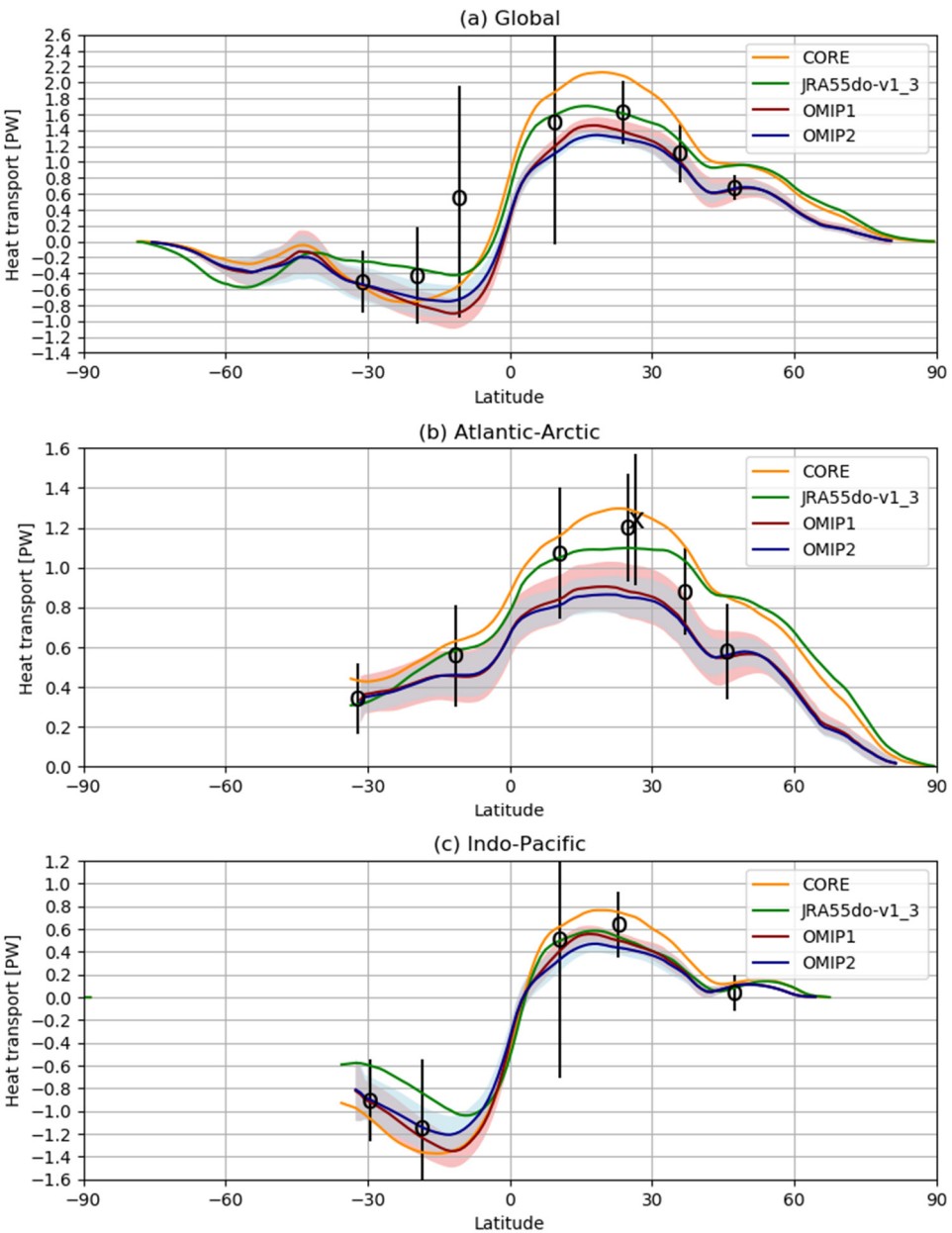

**Figure 17: Multi-model mean, 20-year (1988–2007) mean northward heat transport (PW = $10^{15}$ W m$^{-2}$) in three oceanic basins. (a) Global, (b) Atlantic-Arctic, and (c) Indo-Pacific Ocean basins. Multi-model mean (lines) and spread defined as one standard deviation of the ensemble (shades) of OMIP-1 (red) and OMIP-2 (blue). For reference, implied northward heat transports derived from CORE (orange) and JRA55-do (green) dataset using sea surface temperature from COBE-SST (Ishii et al. 2005) as the lower boundary condition are depicted as in Tsujino et al. (2018). The open circles are estimated from observations and assimilations complied by Macdonald and Baringer (2013). The cross at 26.5°N in the Atlantic (b) is an estimation from RAPID transport array reported by McDonagh et al. (2015). See Figs. S42 and S43 for results of individual models.**


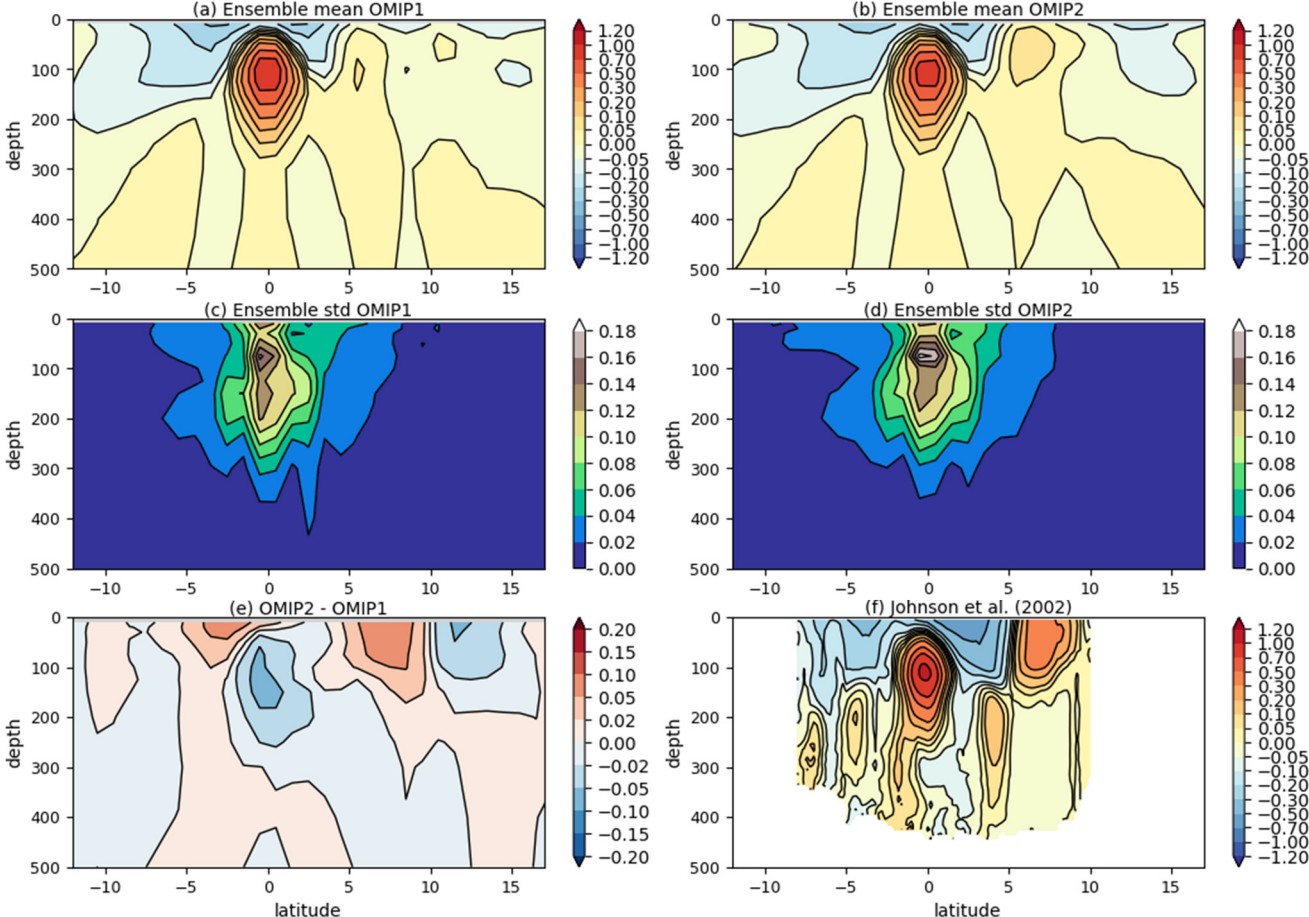

Multi Model Mean Zonal velocity at eastern Tropical Pacific (ave. from 1980 to 2009)

**Figure 18: Upper two panels show multi-model mean, 30-year (1980–2009) mean zonal velocity across 140°W in the eastern tropical Pacific. (a) OMIP-1 and (b) OMIP-2. Middle two panels show the standard deviation of the ensemble. (c) OMIP-1 and (d) OMIP-2. (e) OMIP-2 minus OMIP-1. (f) Observational estimates based on Johnson et al. (2002). Units are m s$^{-1}$. See Figs. S44 through S46 for results of individual models.**

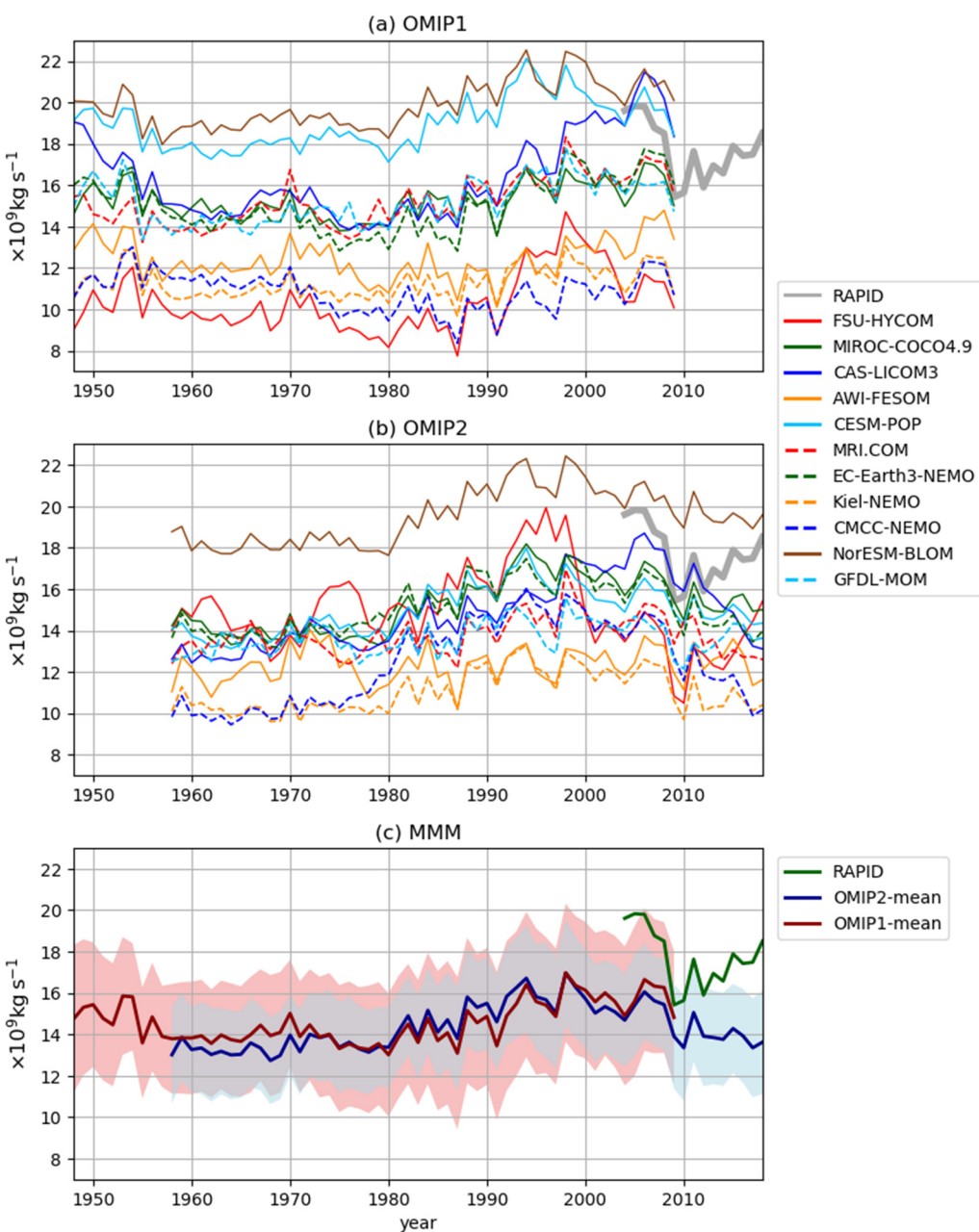

**Figure 19: Time series of annual mean Atlantic meridional overturning circulation (AMOC) maximum at 26.5°N, which represents the strength of AMOC associated with North Atlantic Deep Water formation. (a) OMIP-1, (b) OMIP-2, (c) Multi-model mean (lines) and spread defined as one standard deviation (shades) of OMIP-1 (red) and OMIP-2 (blue). The estimate based on the RAPID observation (e.g., Smeed et al. 2019) is depicted with the grey line in (a) and (b) and the green line in (c). From Fig. 19 to 26, all participating models have been included in the multi-model ensemble mean. Units are $10^9$ kg s$^{-1}$.**

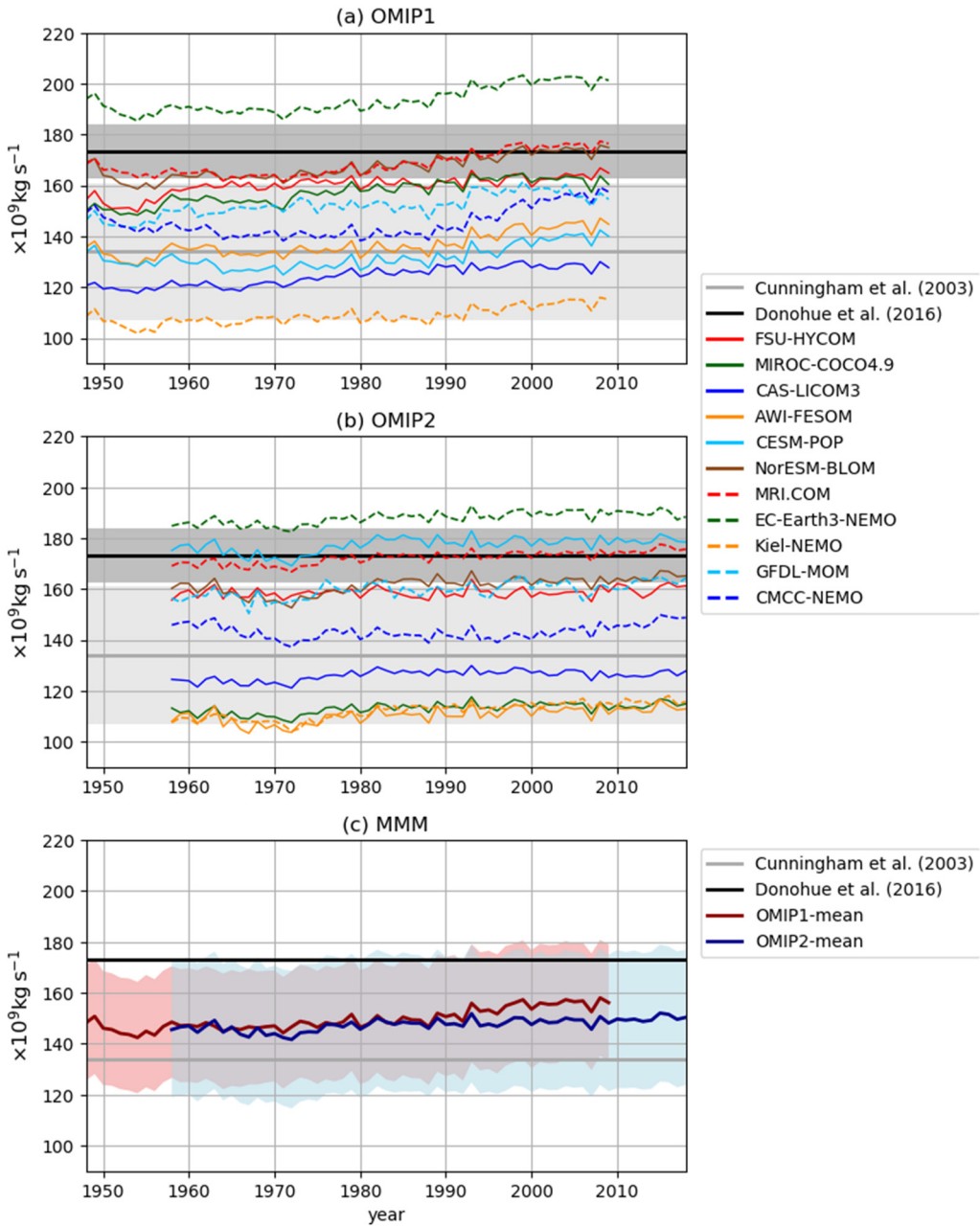

**Figure 20: Same as Fig. 19, but for the Drake passage transport (positive eastward), which represents the strength of Antarctic Circumpolar Current. Units are $10^9$ kg s$^{-1}$. Observational estimates are due to Cunningham et al. (2003) 134 ± 27 Sv (1 Sv = $10^9$ kg s$^{-1}$) and Donohue et al. (2016) 173.3 ± 10.7 Sv.**


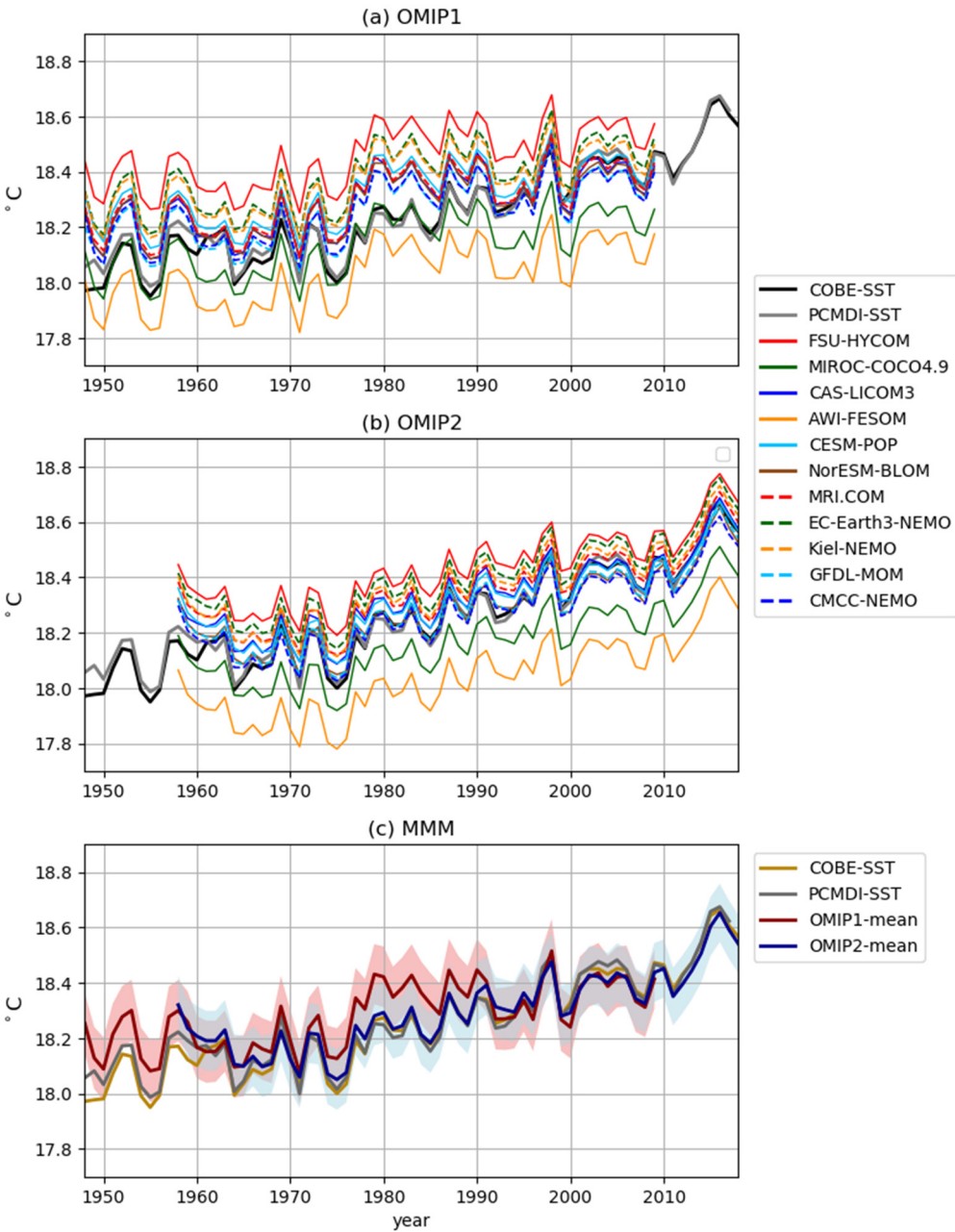

**Figure 21: Same as Fig. 19, but for the globally averaged sea surface temperature (°C). Observational estimates by COBE-SST (Ishii et al. 2005) and PCMDI-SST are depicted as references. The model spreads (±2σ) of both OMIP-1 and OMIP-2 capture the observation for the entire period. The z-value of the difference between OMIP-1 and OMIP-2 for the period from 1980 to 2009 is −0.46 (See also Table 2).**


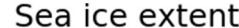

**Figure 22: Time series of sea ice extent in both hemispheres of the last cycle of the simulations ($10^6$ km$^2$). (a – c) March (winter) sea ice extent in the northern hemisphere. (d – f) September (summer) sea ice extent in the northern hemisphere. (g – i) March (summer) sea ice extent in the southern hemisphere. (j – l) September (winter) sea ice extent in the southern hemisphere. (a,d,g,j) OMIP-1, (b,e,h,k) OMIP-2, (c,f,i,l) Multi model mean (lines) and spread defined as one standard deviation (shades) of OMIP-1 (red) and OMIP-2 (blue). In each panel, National snow and ice data center Sea Ice Index (NSIDC-SII; Fetterer et al. 2017) has been depicted as a reference with bold black lines for the left and middle panels and bold green lines for the right panels. The model spreads (±2σ) of both OMIP-1 and OMIP-2 capture the observation except for summer in the southern hemisphere of the OMIP-1 simulations (55% of the period from 1979 to 2009). See Table 2 for the z-values of the difference between OMIP-1 and OMIP-2 for the period from 1980 to 2009.**



Sea ice extent

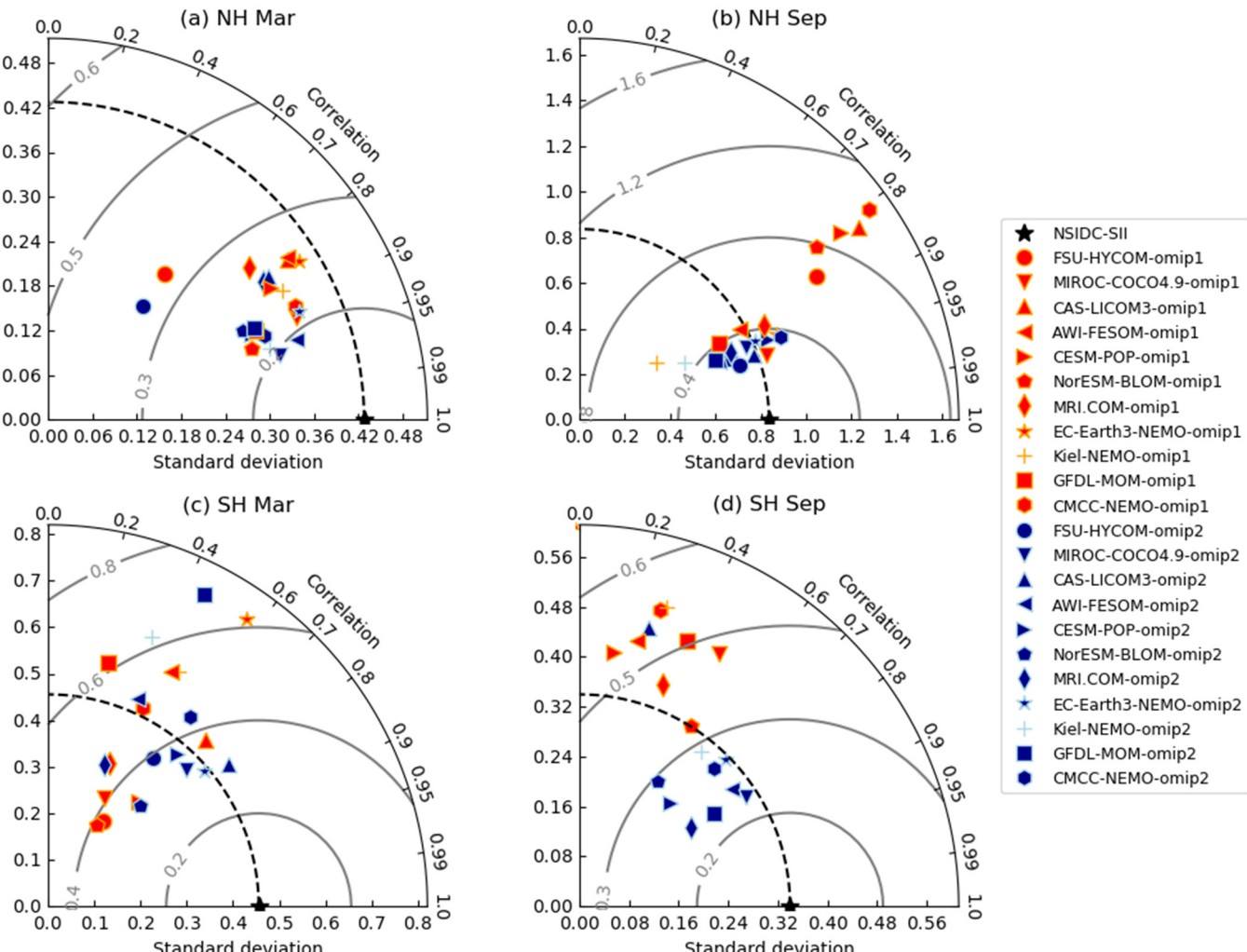

**Figure 23: Taylor diagram of the interannual variation of sea ice extent in both hemispheres relative to NSDIC_SII. (a) March (winter) and (b) September (summer) sea ice extent in the northern hemisphere. (c) March (summer) and (d) September (winter) sea ice extent in the southern hemisphere. Standard deviations are expressed in units of $10^6$ km$^2$.**

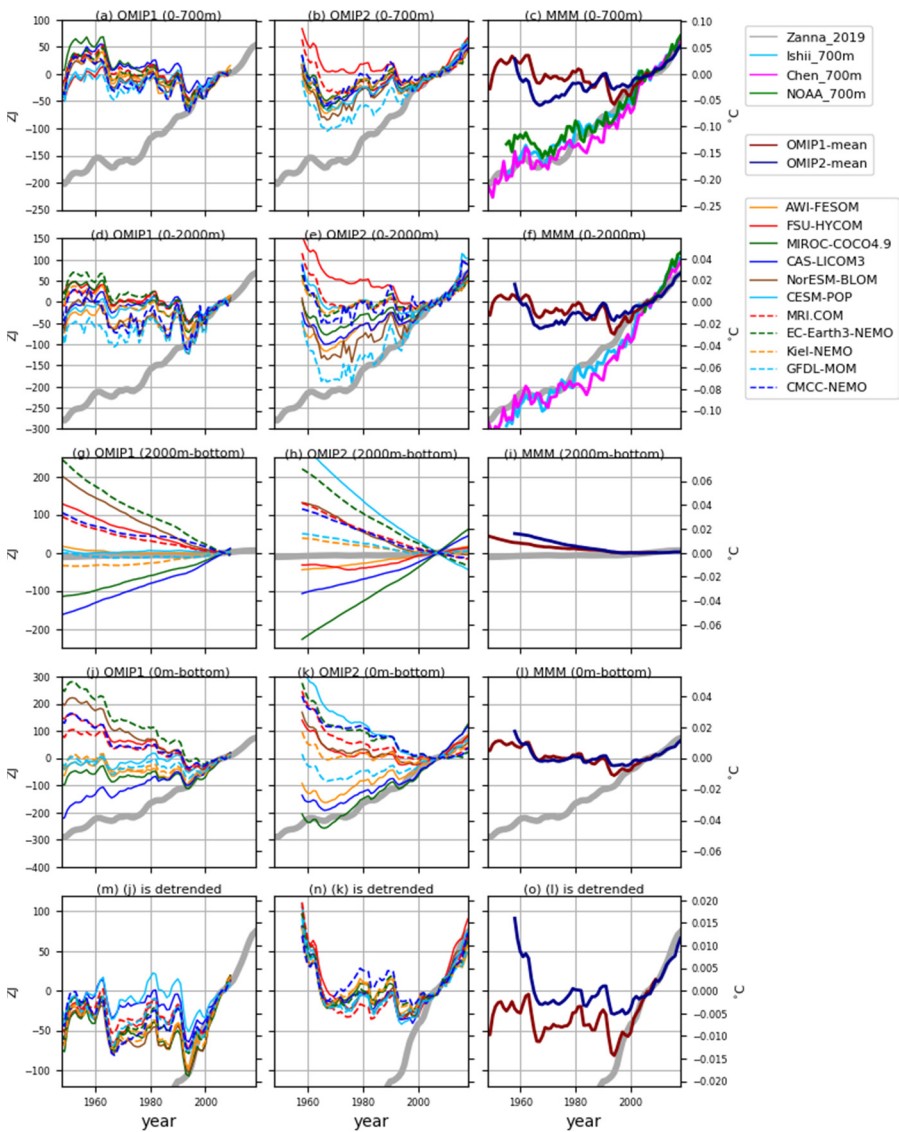

**Figure 24: Time series of annual mean globally integrated ocean heat content anomaly (ZJ = 10²¹ J) in several depth ranges relative to 2005 – 2009 mean. (a – c) 0 m – 700 m. (d – f) 0 m – 2000 m. (g – i) 2000 m – bottom. (j – l) 0 m – bottom. (m – o) 0 m – bottom detrended. (a,d,g,j,m) OMIP-1, (b,e,h,k,n) OMIP-2, (c,f,i,l,o) Multi model ensemble mean of OMIP-1 (red) and OMIP-2 (blue). Note that heat content anomalies from models are calculated by multiplying volume (based on valid points of the WOA13v2 dataset), specific heat (3990 J kg⁻¹ °C⁻¹), and density of sea water (1036 kg m⁻³) to vertically averaged temperatures (°C). Temperature scales are written on the righthand side of vertical axes. Observational estimates are due to Zanna et al. (2019) (grey lines) for all panels, and Ishii et al. (2017) (light blue lines), Chen et al. (2017) (magenta lines), and Levitus et al. (2012) (green lines) for the multi-model mean panels (the right column) if they are available.**

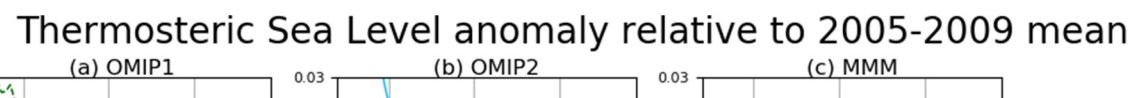

**Figure 25: Time series of annual mean thermosteric sea level anomaly relative to 2005 – 2009 mean (m). (a) OMIP-1, (b) OMIP-2, and (c) Multi-model mean (lines) and spread defined as one standard deviation (shades) of OMIP-1 (red) and OMIP-2 (blue). (d-f) Same as (a-c) except that linear trend is subtracted from each model. Grey lines in (a,b,d,e) and green lines in (c,f) are adopted from WCRP Global Sea Level Budget Group (2018).**

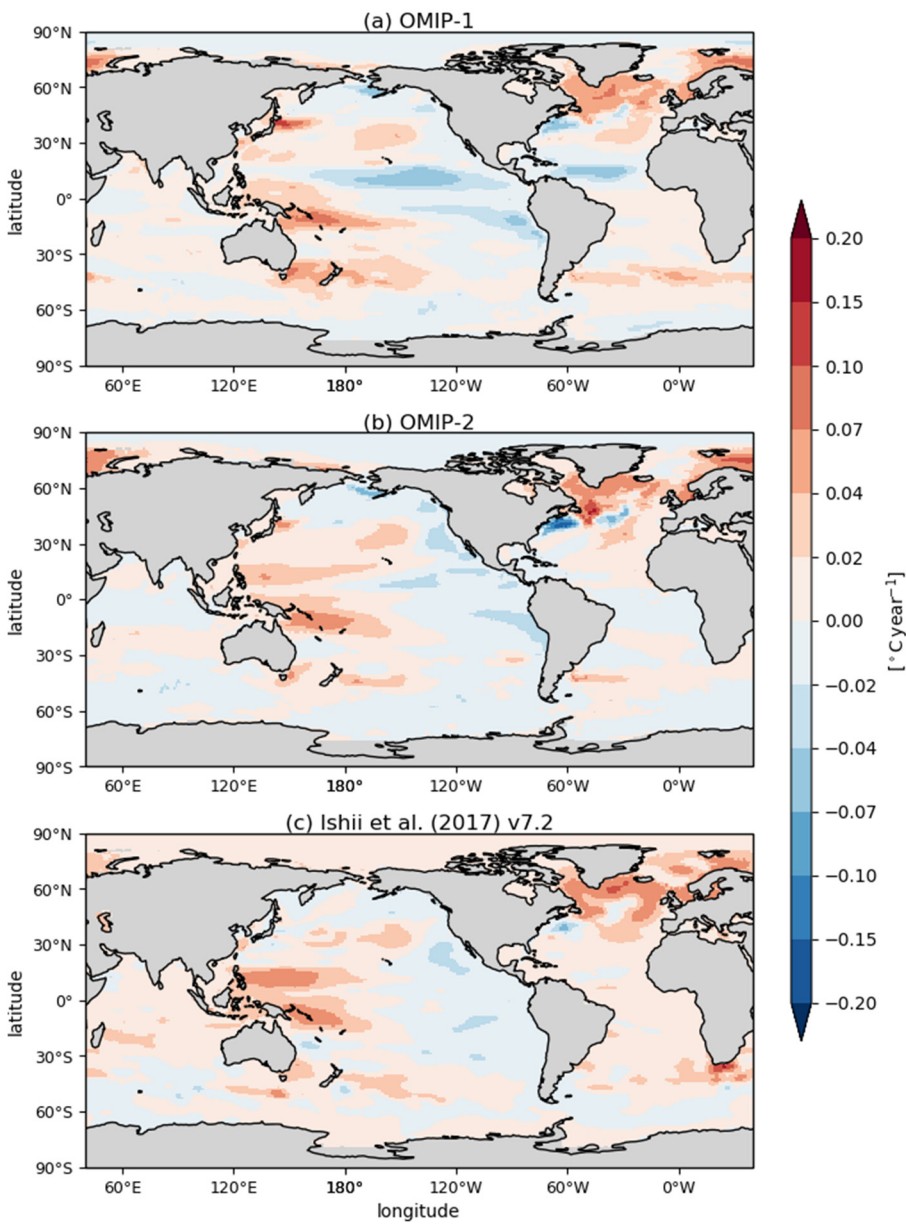

**Figure 26: Multi-model mean 17-year (1993–2009) trend of upper 700 m temperature (°C year⁻¹). (a) OMIP-1, (b) OMIP-2, (c) Ishii et al. (2017) v7.2. See Figs. S47 and S48 for the behavior of individual models.**


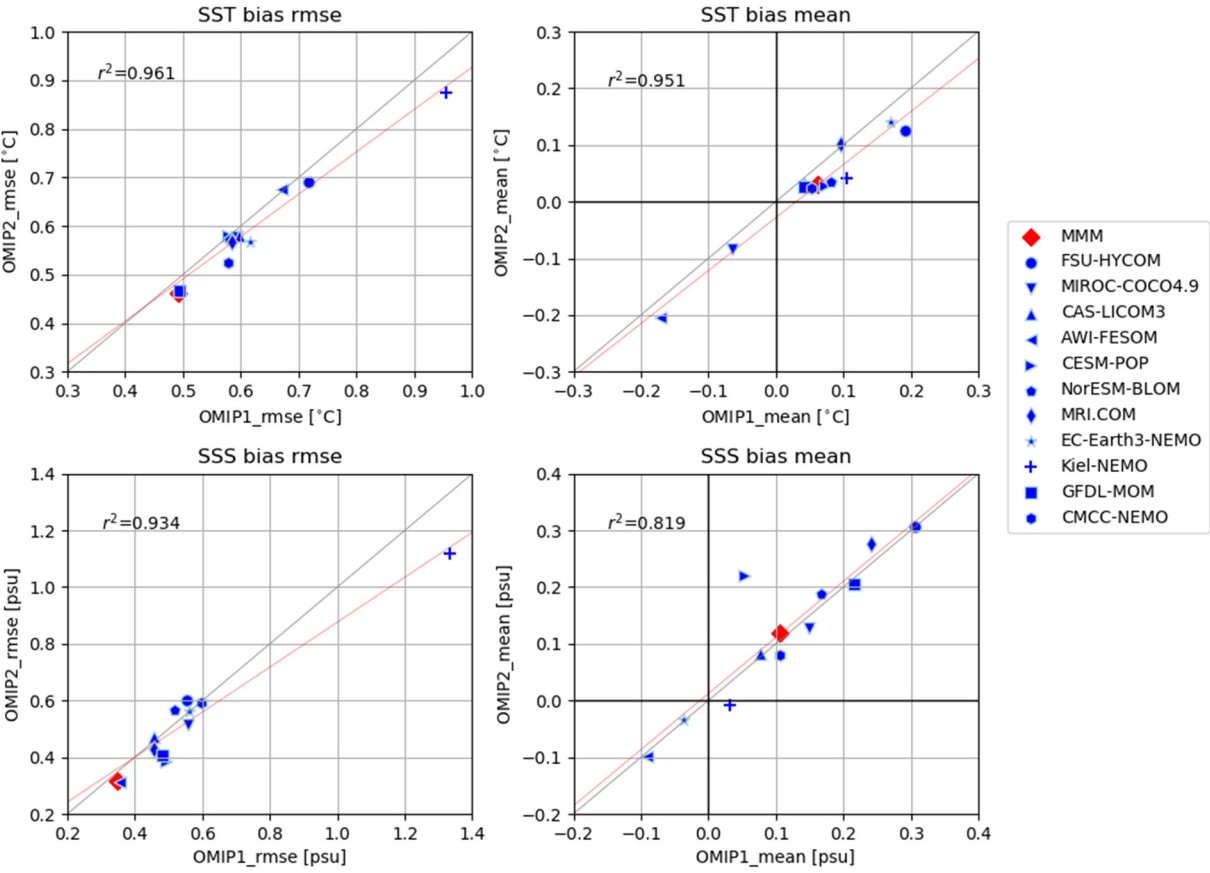

**Figure 27: Scatter diagram with linear fitting (red line) and its score ($r^2$) comparing the 30-year mean (1980–2009) SST bias rmse (upper left), SST bias mean (upper right), SSS bias rmse (lower left), and SSS bias mean (lower right) from OMIP-1 (abscissa) and OMIP-2 (ordinate). See Table D3 for specific values.**

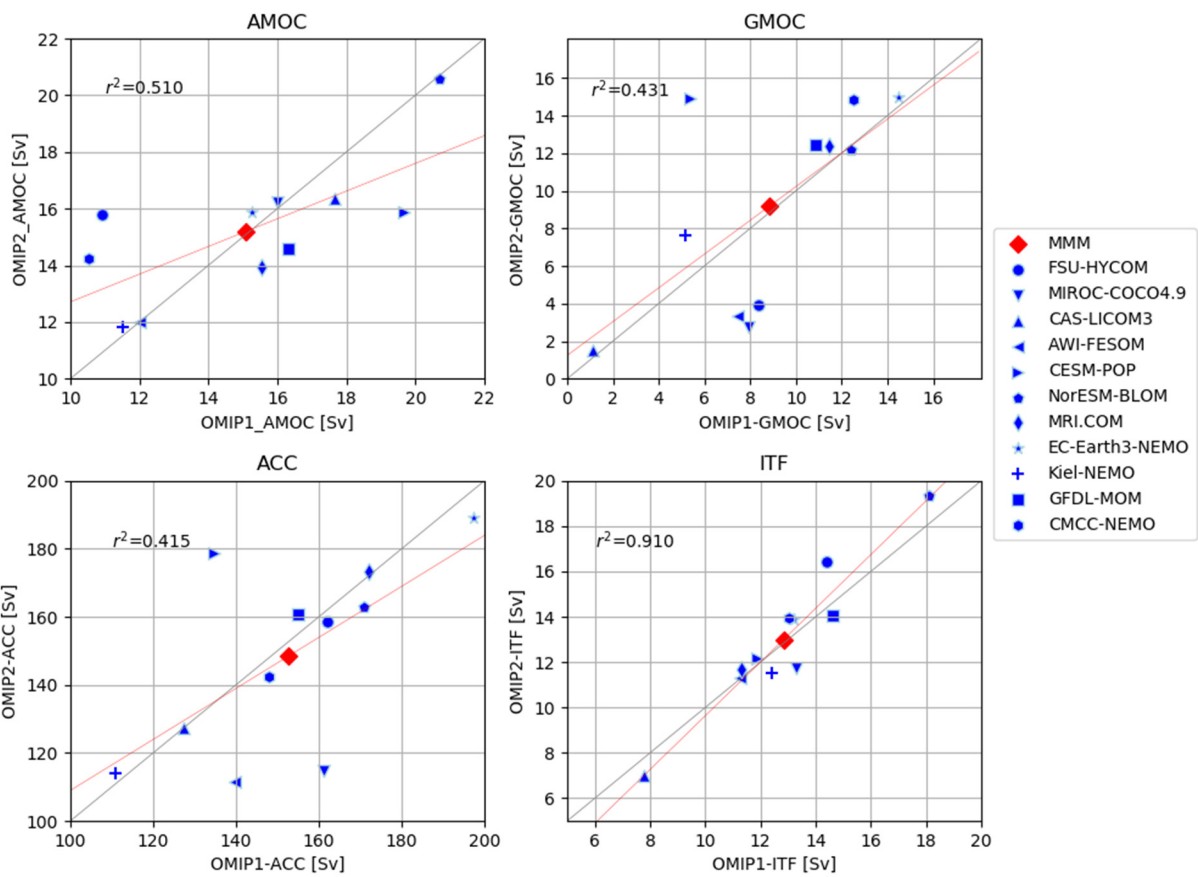

**Figure 28: Same as Fig. 27 but for the 30-year mean (1980–2009) AMOC (upper left), Global meridional overturning circulation (GMOC) minimum in 2000 m – bottom depths at 30°S (upper right), Antarctic Circumpolar Current (ACC) (lower left), and Indonesian Throughflow (ITF) (lower right) from OMIP-1 (abscissa) and OMIP-2 (ordinate). See Table D2 for specific values.**


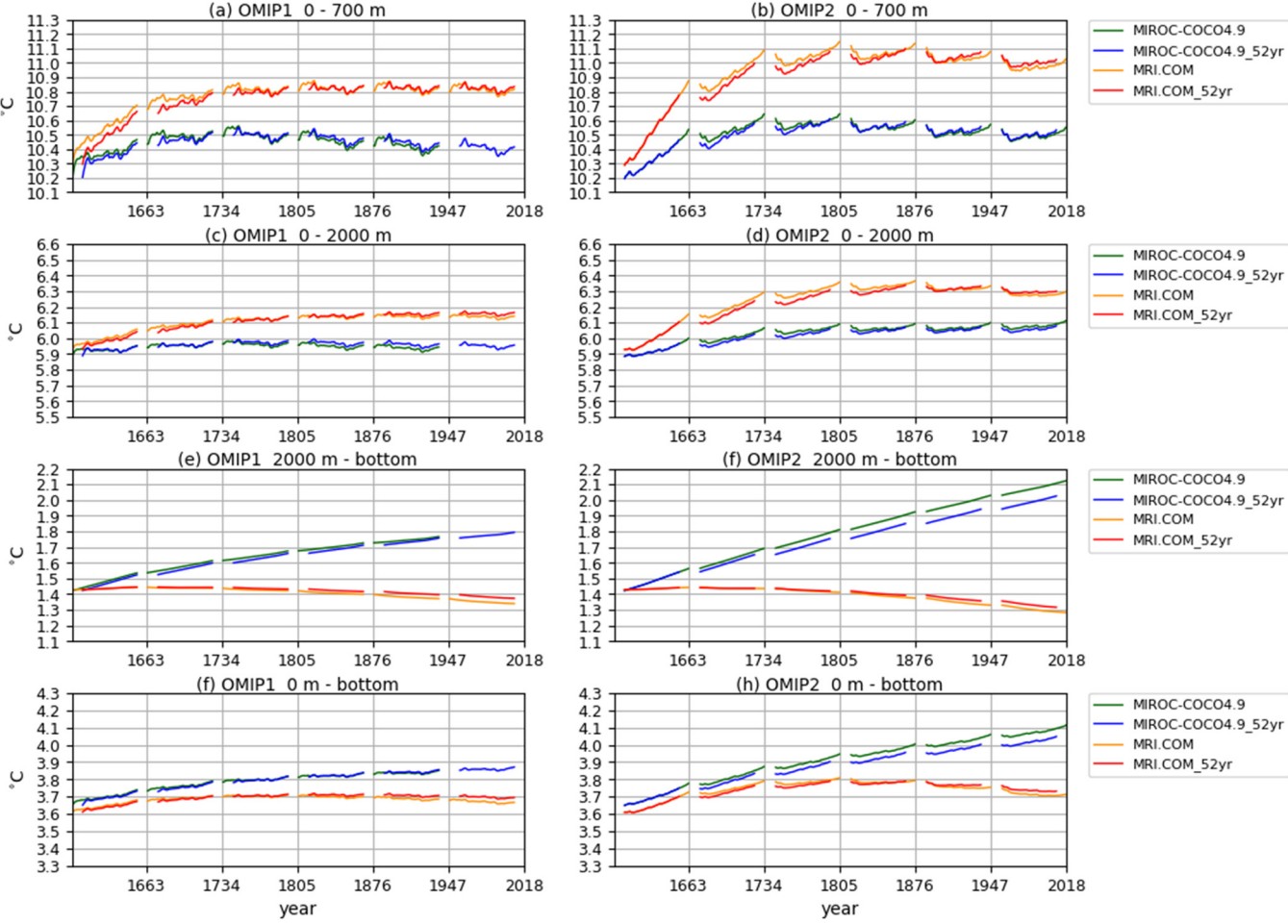

Figure B1: Drift of annual mean, global mean vertically averaged temperatures (°C) for four depth ranges (a, b) 0 – 700m, (c, d) 0 – 2000m, (e, f) 2000m – bottom, (g, h) 0 m – bottom of two sets of OMIP-1 and OMIP-2 simulations differing in the period used for repeating conducted by two models (MIROC-COCO4.9 and MRI.COM). (green) MIROC-COCO4.9 simulations using full period (1948–2009 for OMIP-1 and 1958–2018 for OMIP-2) for repeating. (blue) MIROC-COCO4.9 simulations using common period of OMIP-1 and OMIP-2 forcing (1958–2009). (orange) MRI.COM simulations using full period and (red) MRI.COM simulations using 1958–2009.

# Ocean Circulation Index

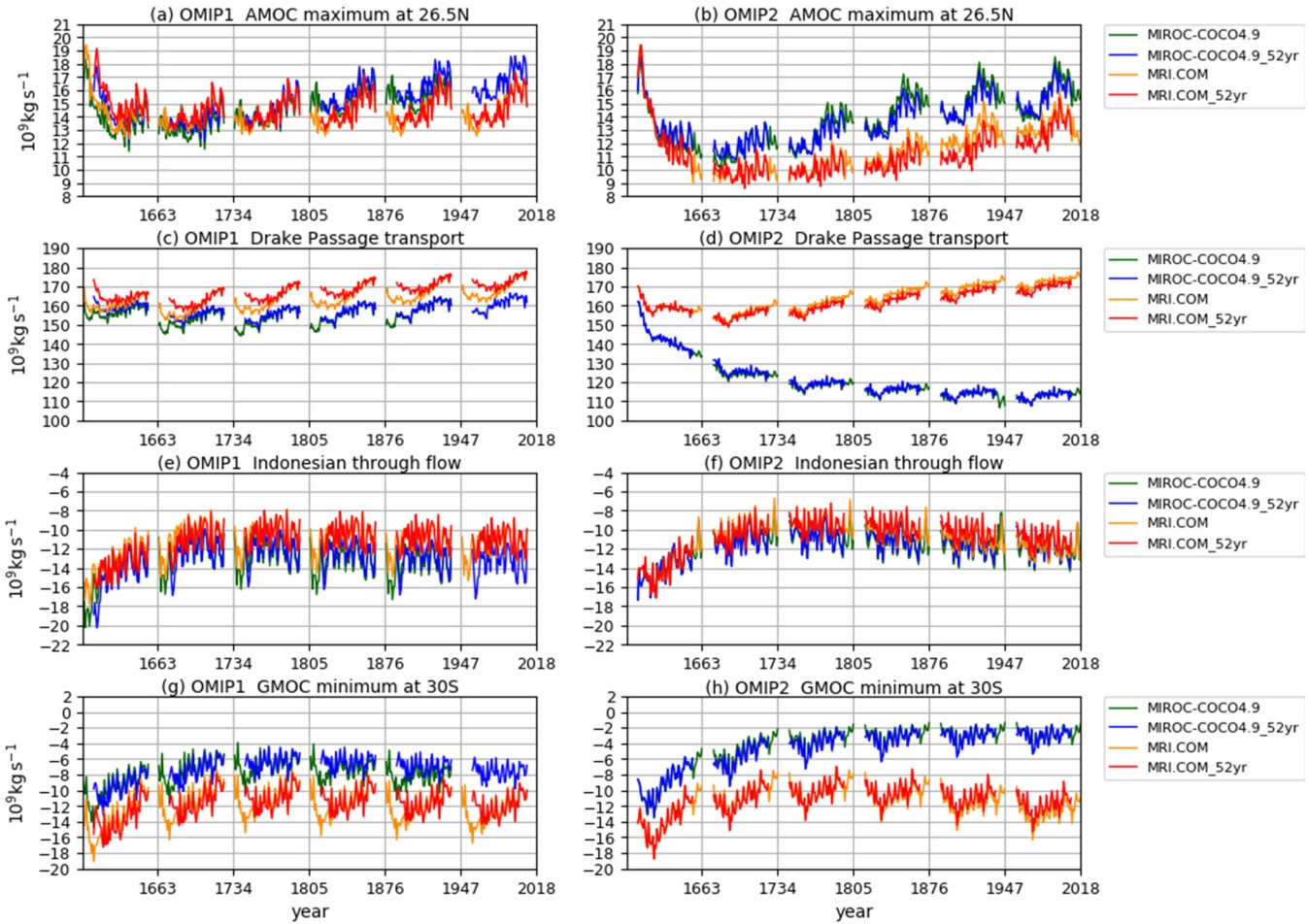

**Figure B2: Time series of annual mean ocean circulation metrics of the two sets of OMIP-1 and OMIP-2 simulations differing in the period used for repeating conducted by two models (MIROC-COCO4.9 and MRI.COM). (green) MIROC-COCO4.9 simulations using full period (1948–2009 for OMIP-1 and 1958–2018 for OMIP-2) for repeating. (blue) MIROC-COCO4.9 simulations using common period of OMIP-1 and OMIP-2 forcing (1958–2009) for repeating. (orange) MRI.COM simulations using full period and (red) MRI.COM simulations using 1958–2009. (a, b) Atlantic meridional overturning circulation (AMOC) maximum at 26.5°N. (c, d) Drake passage transport (positive transport eastward). (e, f) Indonesian Throughflow (negative into the Indian Ocean). (g, h) Global meridional overturning circulation (GMOC) minimum between 2000 m – bottom at 30°S. Units are $10^9$ kg s$^{-1}$.**

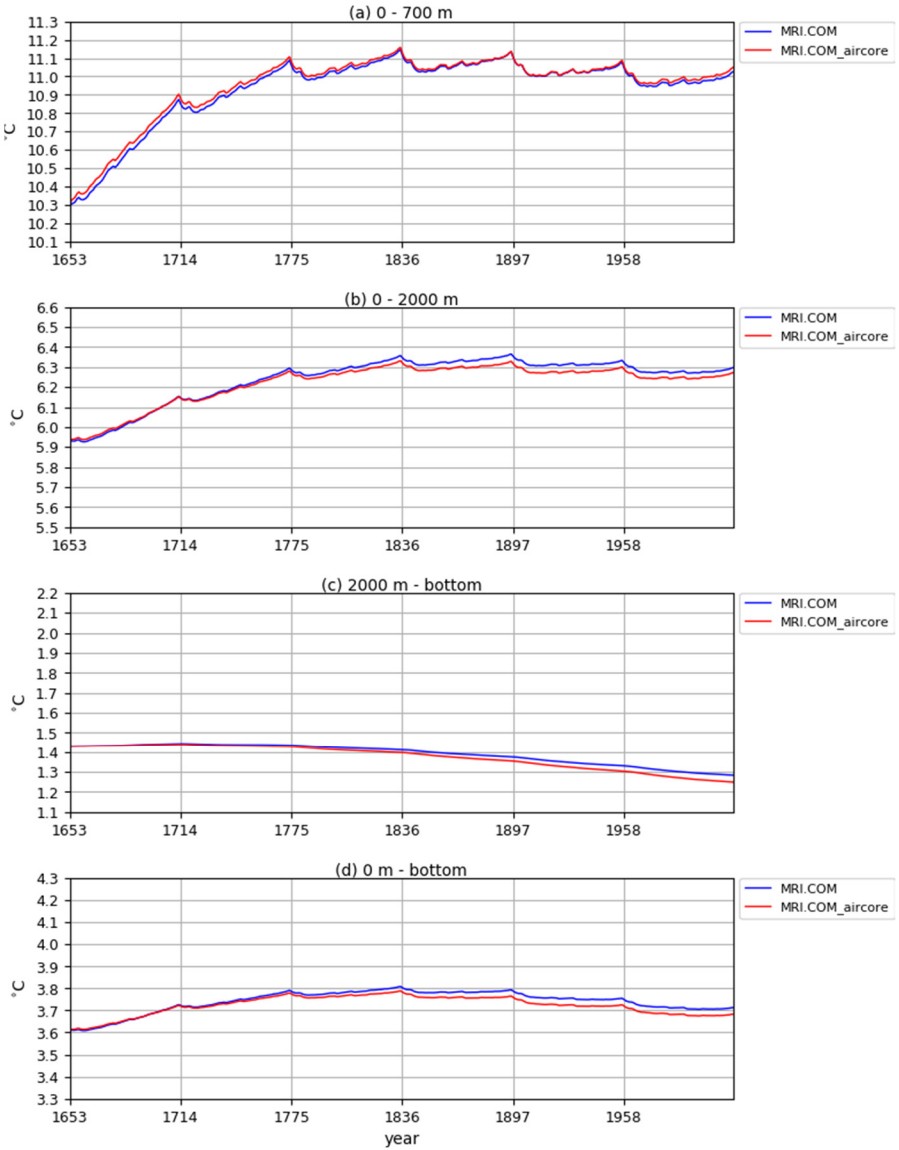

**Figure B3: Drift of annual mean, global mean vertically averaged temperatures (°C) for four depth ranges (a) 0 – 700m, (b) 0 – 2000m, (c) 2000m – bottom, (d) 0 m – bottom of two OMIP-2 simulations by MRI.COM differing in the set of formulae for properties of moist air used to compute surface turbulent fluxes. (blue) Gill (1982) and (red) Large and Yeager (2004; 2009).**


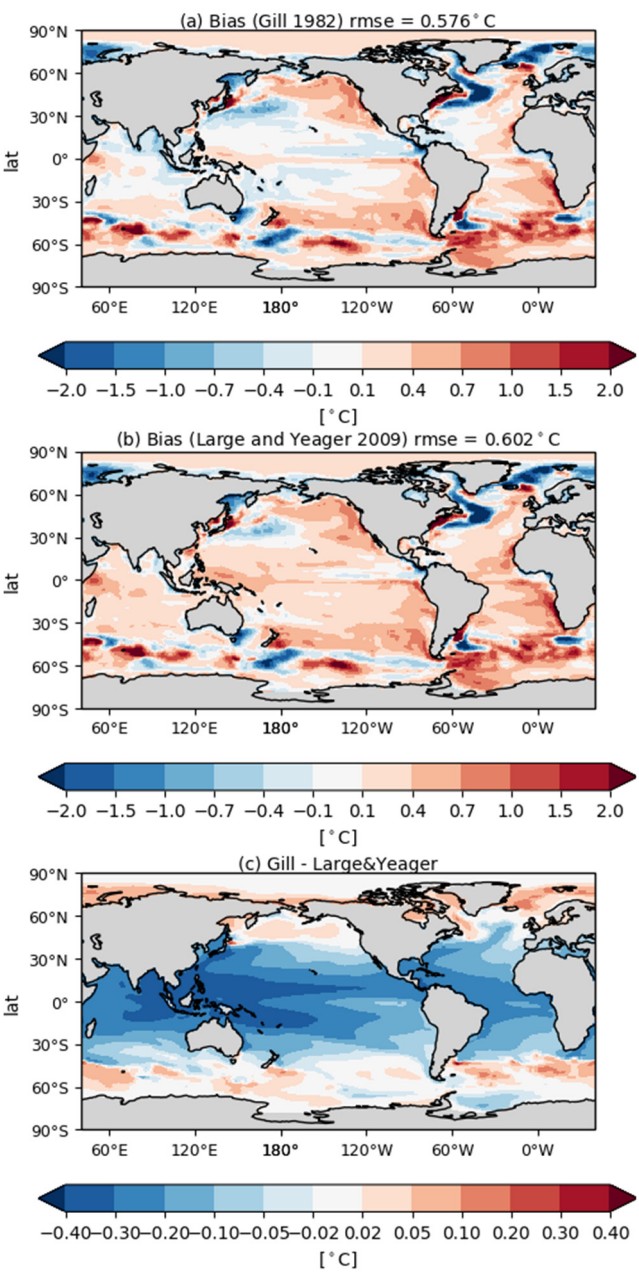

**Figure B4:** Upper two panels show the bias of 30-year (1980–2009) mean SST relative to PCMDI-SST of two OMIP-2 simulations by MRI.COM differing in the set of formulae for properties of moist air used to compute surface turbulent fluxes. (a) Gill (1982) and (b) Large and Yeager (2004; 2009). (c) (a) minus (b). Units are degrees Celsius (°C).


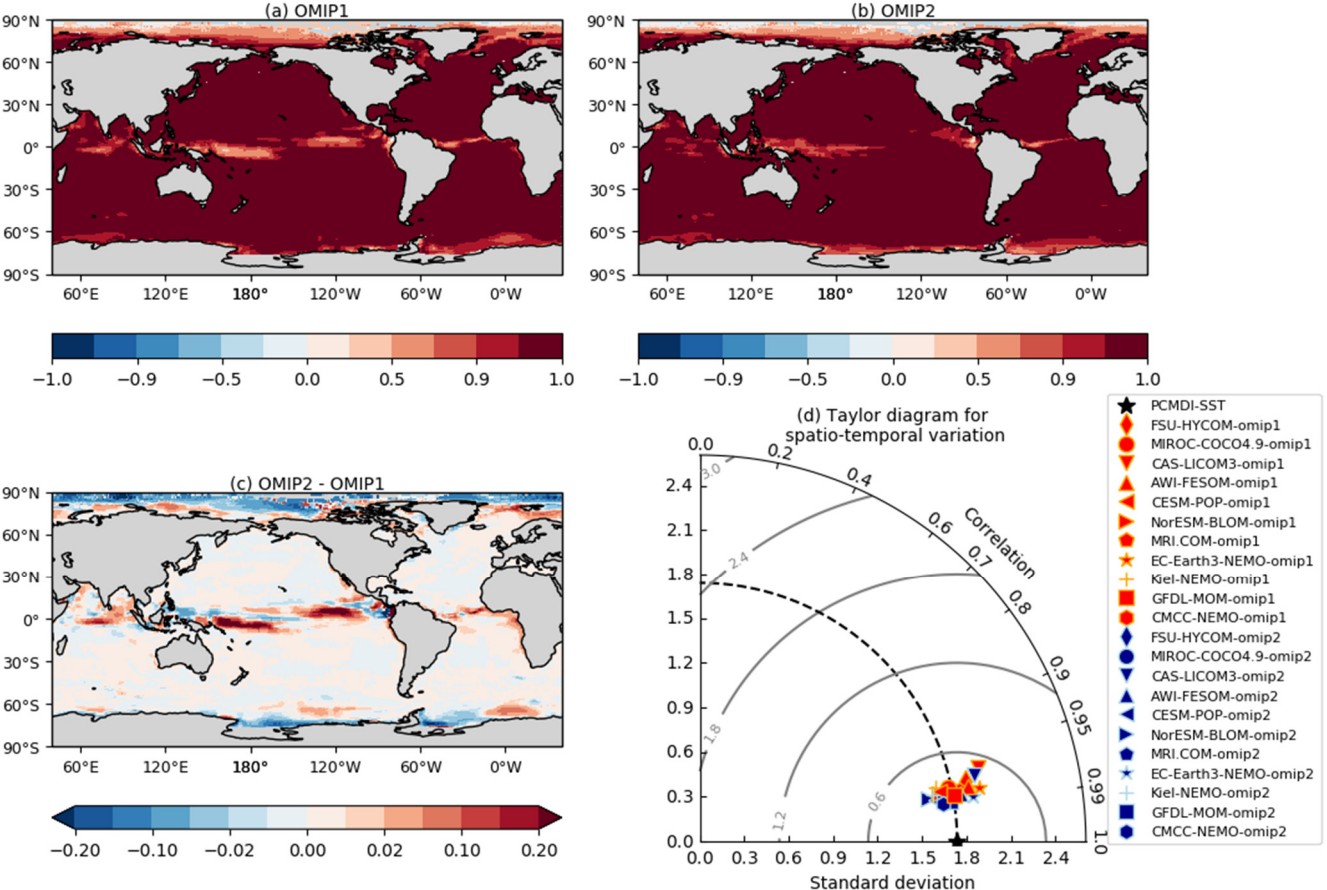

**Figure E1: Multi-model mean correlation coefficients of monthly climatology of SST for the period 1980–2009 between simulation and PCMDI-SST. (a) OMIP-1, (b) OMIP-2, (c) OMIP-2 − OMIP-1. (d) Taylor diagram of the total space-time pattern variability of the monthly climatology of SST of OMIP-1 and OMIP-2 simulations relative to PCMDI-SST, with standard deviations expressed in units of °C. For Figs. E1 to E5, all models are used for multi-model mean. See Figs. S49 through S51 for the results of individual models.**


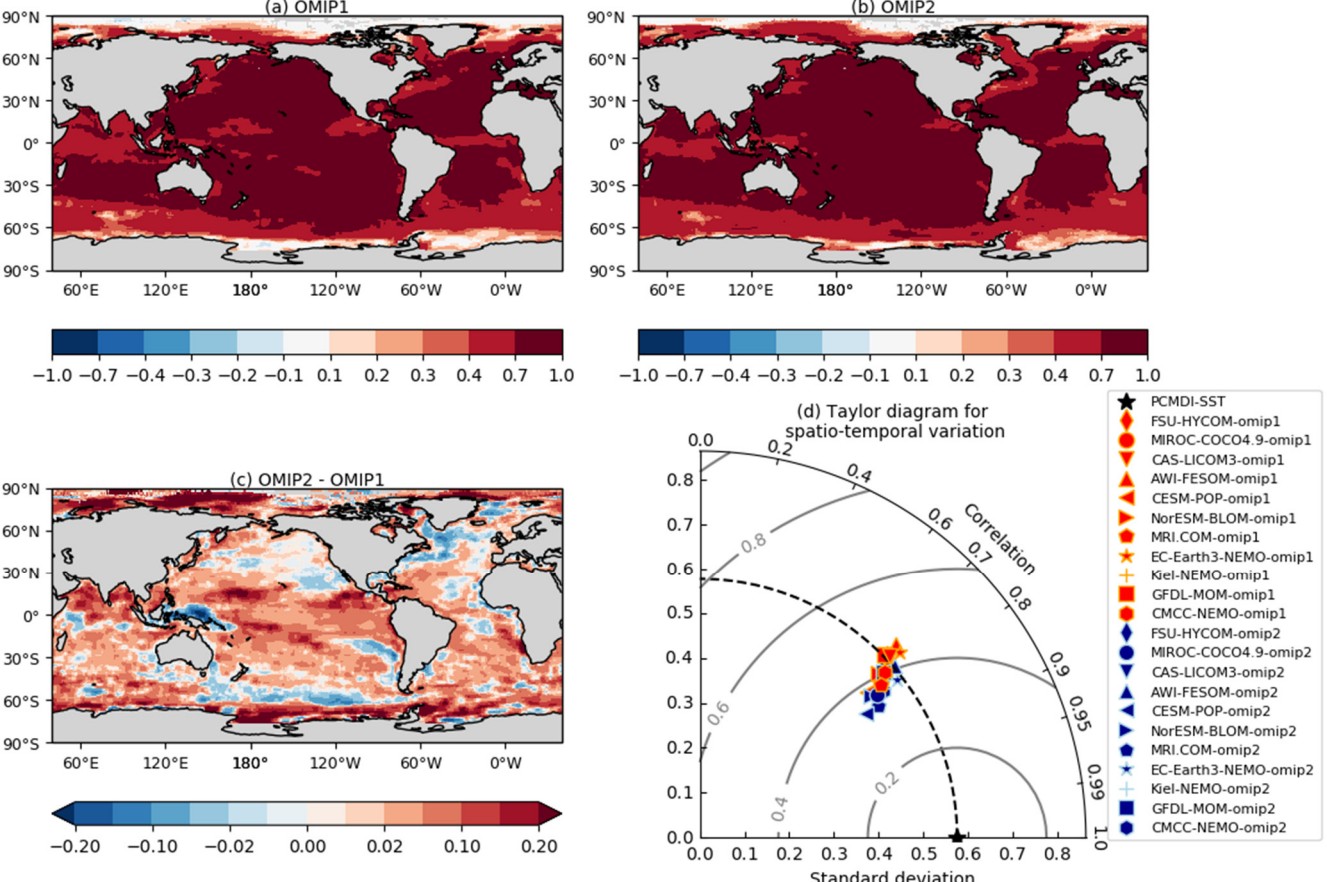

**Figure E2: Multi-model mean correlation coefficients of monthly SST anomaly relative to the monthly climatology for the period 1980–2009 between simulation and PCMDI-SST. (a) OMIP-1, (b) OMIP-2, (c) OMIP-2 − OMIP-1. (d) Taylor diagram of the total space-time pattern variability of monthly SST anomaly relative to the monthly climatology for OMIP-1 and OMIP-2 simulations relative to PCMDI-SST, with standard deviations expressed in units of °C. See Figs. S52 through S54 for results of individual models.**

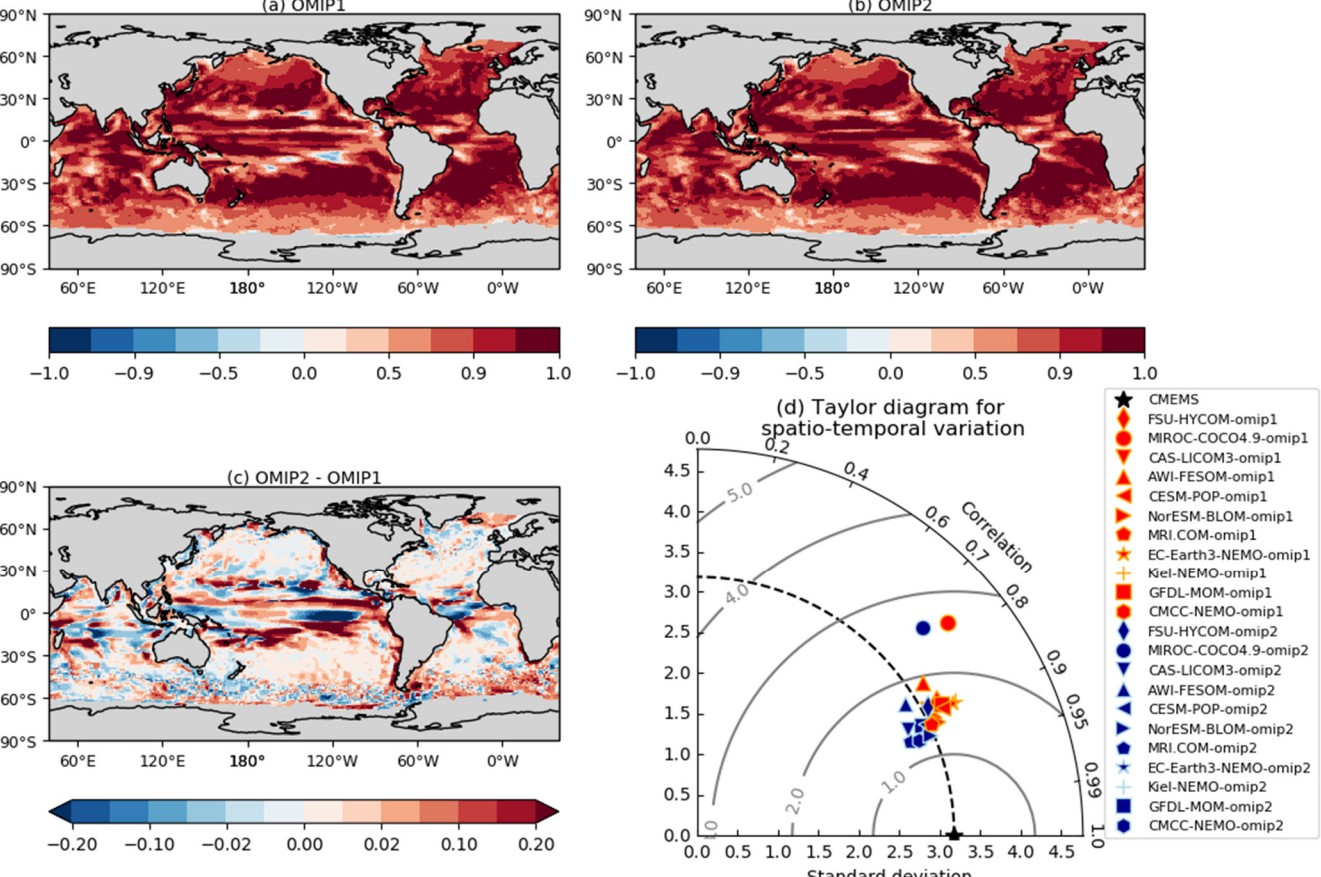

**Figure E3: Multi-model mean correlation coefficients of monthly climatology of SSH for the period 1993–2009 between simulation and CMEMS. (a) OMIP-1, (b) OMIP-2, (c) OMIP-2 − OMIP-1. (d) Taylor diagram of the total space-time pattern variability of the monthly climatology of SSH of OMIP-1 and OMIP-2 simulations relative to CMEMS, with standard deviations expressed in units of centimeters. See Figs. S55 through S57 for results of individual models.**


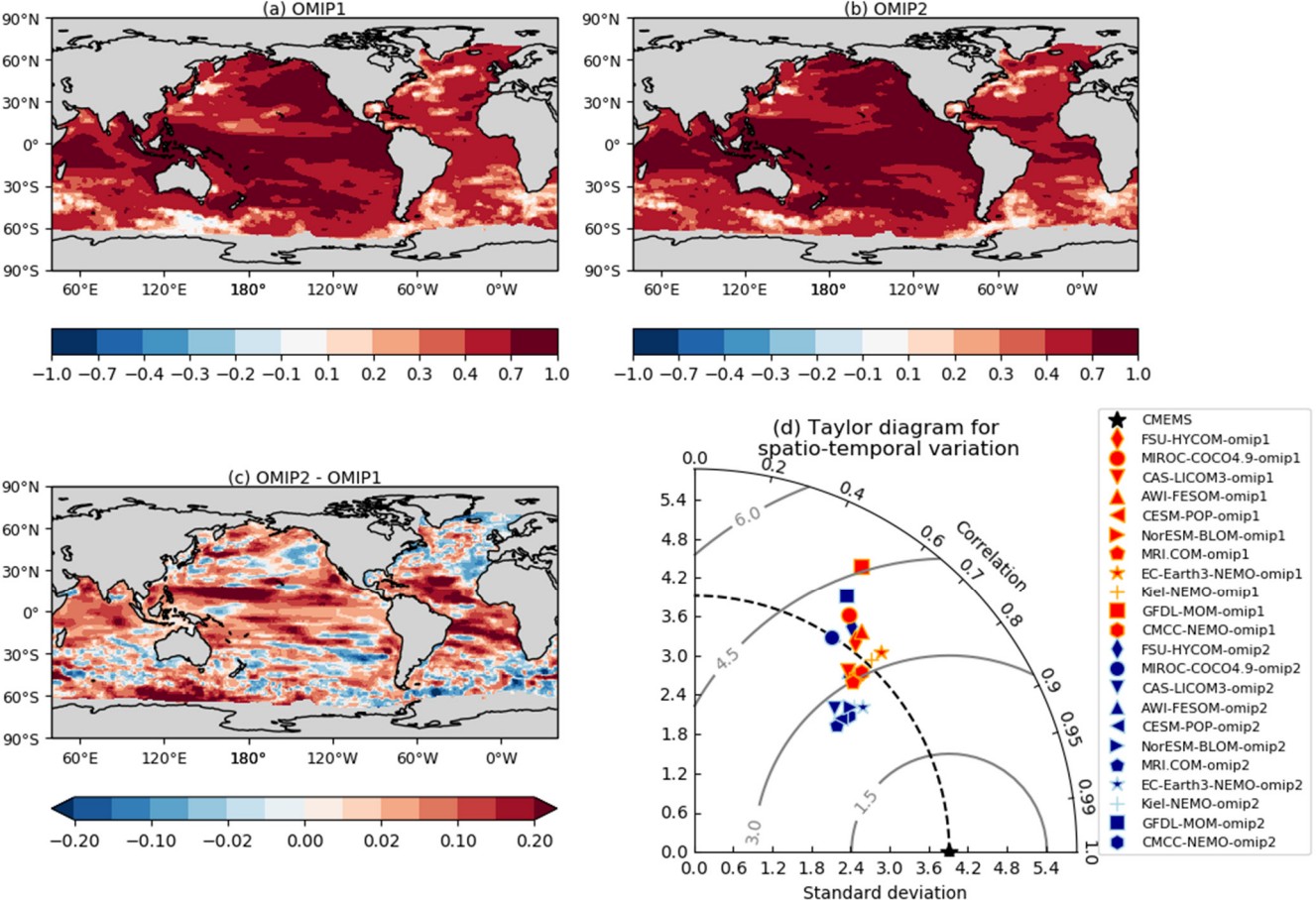

**Figure E4: Multi-model mean correlation coefficients of monthly SSH anomaly relative to the monthly climatology for the period 1993–2009 between simulation and CMEMS. (a) OMIP-1, (b) OMIP-2, (c) OMIP-2 − OMIP-1. (d) Taylor diagram of the total space-time pattern variability of monthly SSH anomaly relative to the monthly climatology for OMIP-1 and OMIP-2 simulations relative to CMEMS, with standard deviations expressed in units of centimeters. See Figs. S58 through S60 for results of individual models.**


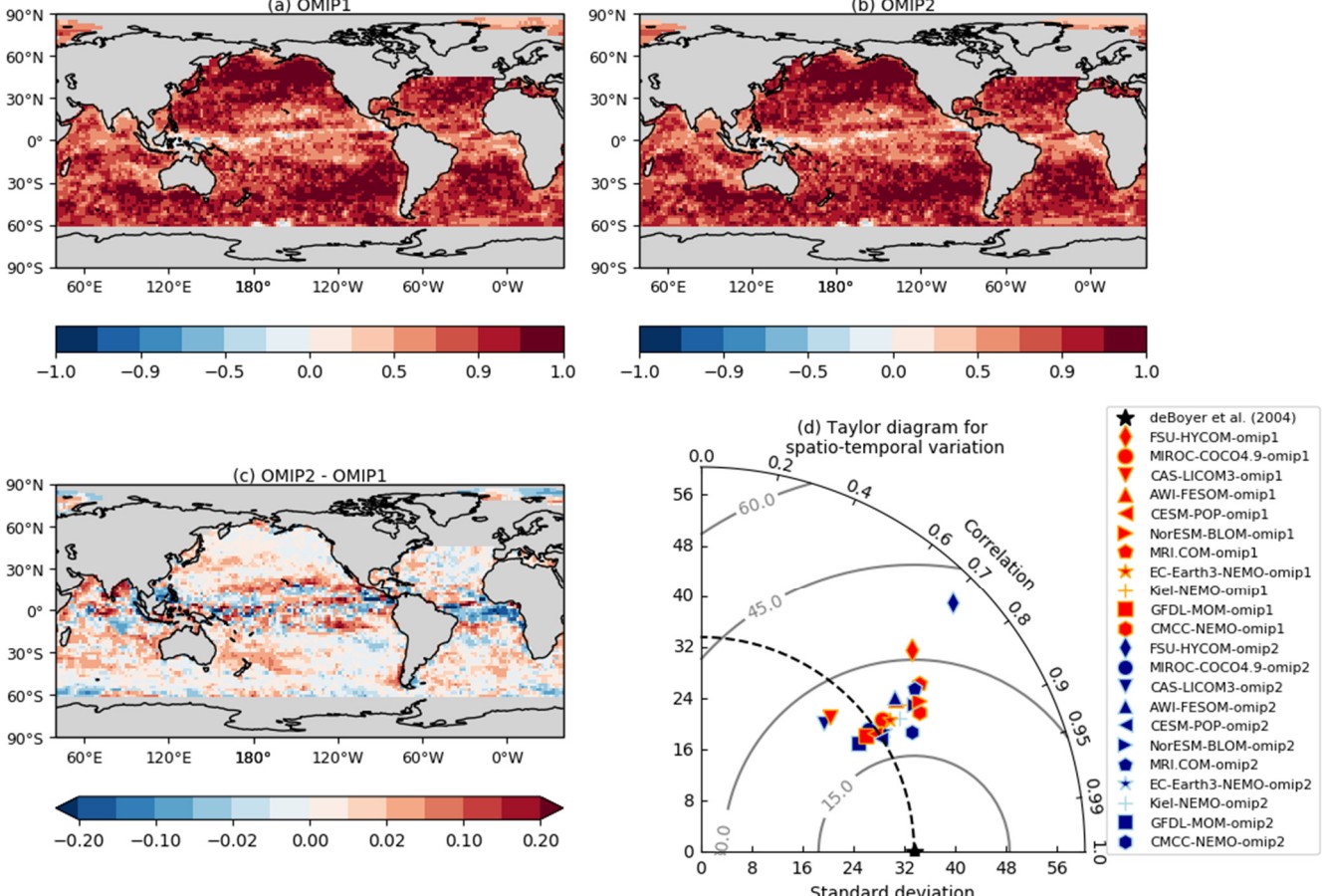


**Figure E5: Multi-model mean correlation coefficients of monthly climatology of mixed layer depth (MLD) for the period 1980–2009 between simulation and de Boyer Montégut et al. (2004). (a) OMIP-1, (b) OMIP-2, (c) OMIP-2 − OMIP-1. (d) Taylor diagram of the total space-time pattern variability of the monthly climatology of MLD of OMIP-1 and OMIP-2 simulations relative to de Boyer Montégut et al. (2004), with standard deviations expressed in units of meters. See Figs. S61 through S63 for**
**results of individual models.**

## Sea Surface Temperature (tos)

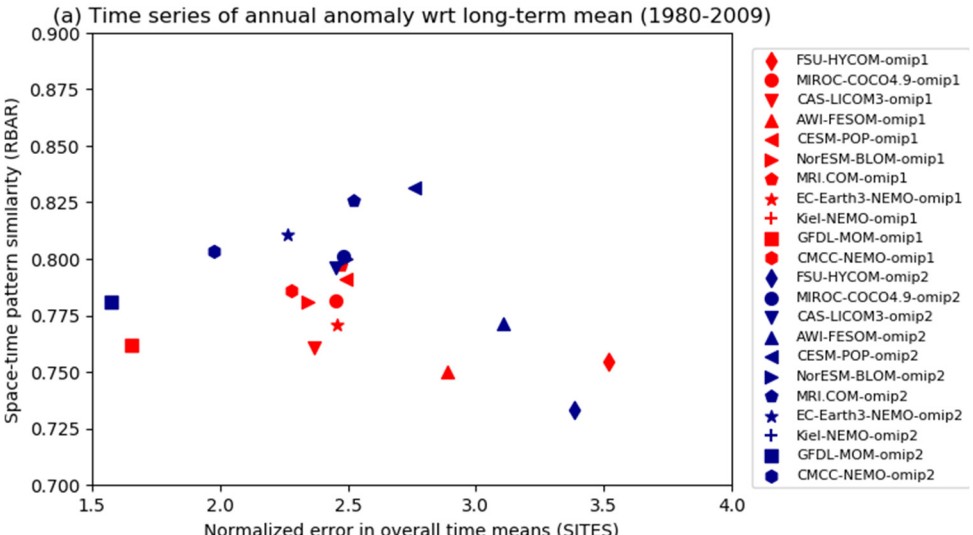

(a) Time series of annual anomaly wrt long-term mean (1980-2009)

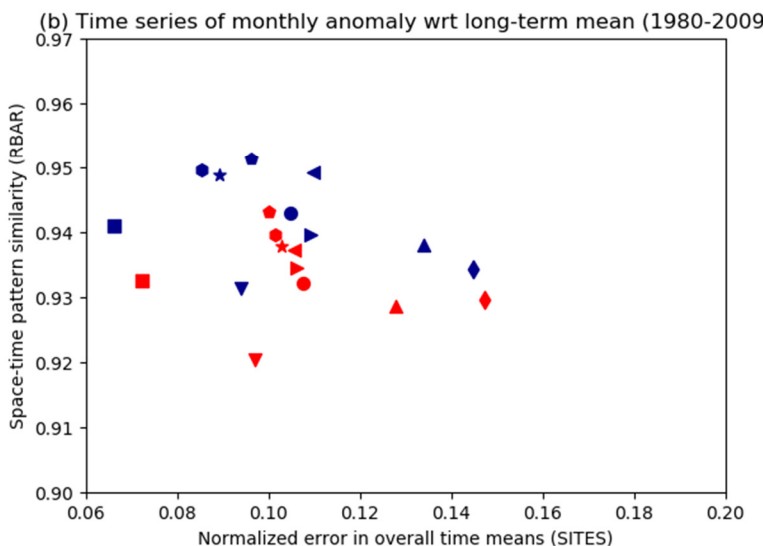

(b) Time series of monthly anomaly wrt long-term mean (1980-2009)

**Figure E6: A model performance diagram showing the OMIP-1 and OMIP-2 simulations of (a) annual mean and (b) monthly mean SST during 1980–2009 in terms of the normalized error of the long-term annual mean (SITES; abscissa) and the temporal mean of the spatial pattern correlation coefficients (RBAR; ordinate) relative to PCMDI-SST.**


## Sea Surface Height (zos)

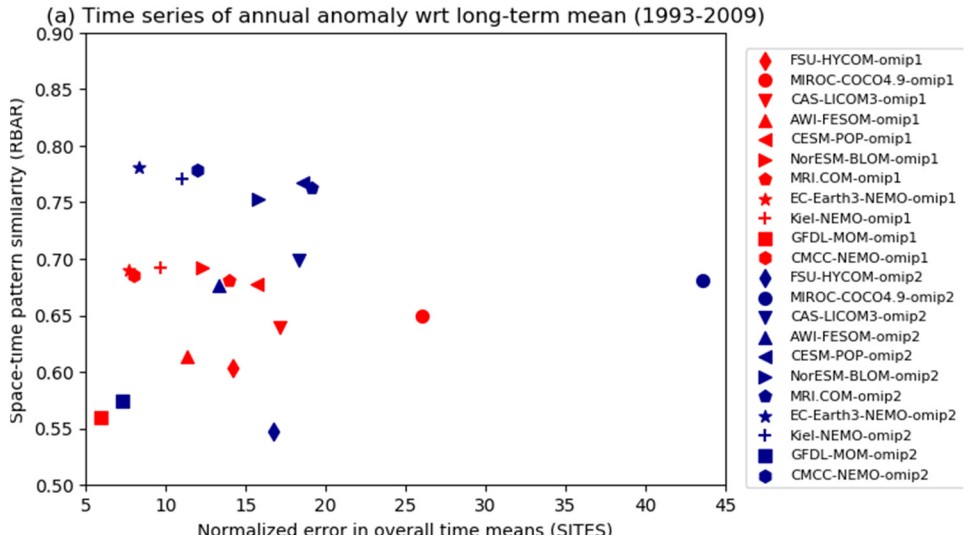

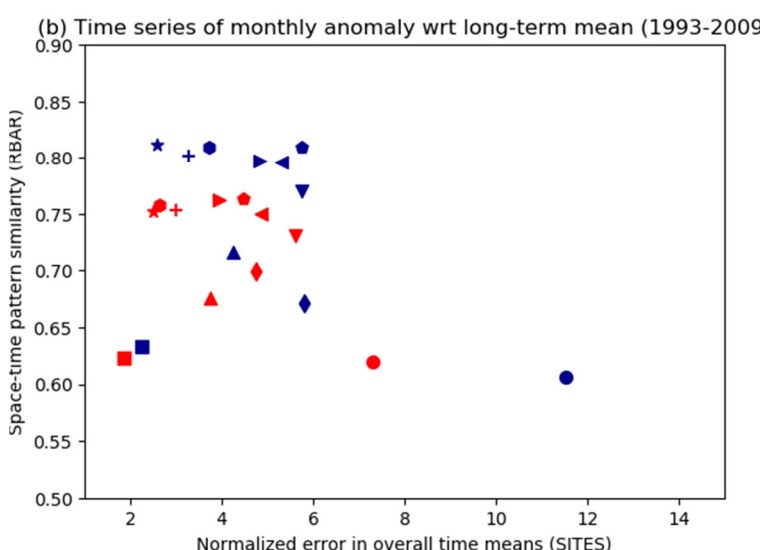

**Figure E7: Same as Fig. E6, but for SSH. Reference SSH dataset is from CMEMS.**