# Peer review of "Evaluation of global ocean-sea-ice model simulations based on the experimental protocols of the Ocean Model Intercomparison Project phase 2 (OMIP-2)"

_Geoscientific Model Development, 2019_

## Referee Comment (RC1) · F. O. Bryan (Referee) · 28 Feb 2020

This manuscript is the logical successor to Griffies et al (2016) defining the OMIP protocol and Tsujino et al (2018) describing the construction of the JRA55-do forcing dataset. The collective efforts of the world's leading ocean modeling groups both in preparing for this study (preparing the forcing data, developing and agreeing on the protocol, running the experiments) and in collecting and collating the results is monumental.

[Figure]

The manuscript documents that the massive effort took place, but to be perfectly honest, it is as dull as dirt. Paragraph after paragraph begins "Figure N presents ...", "Figure N+1 presents ...", going through the catalog of standard metrics. The authors set a low bar by declaring they they will offer only a "glimpse rather than an in depth view of the many elements of ocean model performance" (line 93). With 150 pages of material and several hundred figures, I believe there is more than can be described as a "glimpse", but agree that an in depth view is not offered. I guess this is the consequence of CMIP-ification of climate science. I am sure many groups will use the figures at some point to calibrate their efforts going forward, and more in depth studies will follow, but I would have hoped that we might have found a little more introspection on the successes and shortcomings of the protocol as well as more on the impacts of the structural changes in the forcing data. For example:

- Does each metric considered add value to the assessment, e.g., Do we need 0-700m heat content and SSH metrics or would one or the other be sufficient to discriminate among the included models?

- Will these metrics be relevant as resolution (and resolved variability) increase? There is already some indication that certain of these metrics become misleading.

- Does a change in ordering among models in various metrics in OMIP-1 vs OMIP-2 suggest the importance or not of different aspects of the forcing? What does the change in spread across the ensemble imply about the forcing?

- Are there any obvious groupings of models (e.g. the NEMO models or the hybrid coordinate models) in model skill metrics or not?

- Did variance in the solutions during in the pre-satellite era change more or less as compared to the later years between OMIP-1 and OMIP-2?

- The Tsujino et al (2018) manuscript calls out several "notable differences between CORE and JRA55-do" (pg 106, first pp) . Are these apparent in the solutions?

[Figure]

- How did the additional variability in runoff included in JRA55-do forcing impact the solutions?

In short, what are the high-level conclusions that we can draw about the value of this exercise? I doubt that this question will be addressed in subsequent studies, so this is the obvious place to address it.

SPECIFIC COMMENTS:

Figures: I find the color bar used for positive definite quantities (e.g. 2b,d,f,h) very difficult to interpret. More contrast would be helpful.

Figure 1 and similar: Some explanation of what accounts for the nearly instantaneous development of the ensemble spread in upper ocean heat content, SST etc would be helpful. Perhaps maps of the year 1 bias in each model and how it compares to the longer term mean bias. What structures are responding this rapidly? What can we learn form experiments integrated for a few years vs 360?

Line 303 and following: A comparison at a subsurface (maybe 50m) depth would be more enlightening to factor out the influence of salinity restoring.

Line 330: Would not a simple broadening of the front (irrespective of the occurrence of recirculation gyres) result in such a dipolar structure?

Figure 9a,b: A nonlinear color scale would be helpful to bring out more than the deep water formation sites.

Line 448 and following: It is notable that the SH mean bias improves more because the worst models get better.

Line 455 and following, Figure 22: I found this to be perhaps the most important figure when considering the limitations of the wash-rinse-repeat OMIP cycling. We really do not capture 60 years of variability with a 60 year cycle. Worth emphasizing more strongly.

Appendix B2: Figure B4 a bit of over kill to make the point (did we really think Drake passage transport might depend on small differences in the properties of moist air?), but oh well, only four more panels among 400!

MINOR TYPOS etc:

Line 100: The four . . . or All four . . .

Line 224: smaller drift

Line 227: "subsurface" not clear what depth range is being described

Line 609: piston velocity

Line 610 : 6 cycles (to be constant with rest of text)

Line 662 (CESM)

Line 730: with OM4 configured

---

## Referee Comment (RC2) · Anonymous Referee #2 · 2 Mar 2020

The manuscript describes overall results of ocean model intercomparision organized in the framework of OMIP-2. After the development of new surface boundary forcing dataset (JRA55-do; Tsujino et al. 2018), the performance of various ocean model simulations forced by this new dataset is now reported here.

Under the same protocol proposed by the authors, eleven state-of-the-art global ocean models are forced by not only newly developed JRA55-do atmospheric dataset but also previously referred CORE forcing. This design makes it possible for the authors to clearly evaluate what stems from the difference from the surface forcing and what is

from inter-model differences.

In previous OMIP-1 comparisons, the CORE forcing by Large and Yeager (2009) was developed for surface forcing dataset. This dataset has been widely used for ocean model community but not updated after 2009, therefore, its replacement by newly developed JRA55-do is awaited. The results reported here provide us with the solid evidence that new JRA55-do dataset is good enough to replace CORE forcing as a new forcing dataset for global ocean simulations. The manuscript also presents timely and valuable assessment about the overall performance of the state-of-the-art global ocean models.

Although the manuscript demonstrates the overall performance of global ocean simulations rather than detail analysis about specific topics, such documentation fits the scope of GMD and the ocean model and related communities will benefit from the results reported in this manuscript very much. Therefore, I can recommend the publication of this manuscript in GMD after minor revision. I have several comments which I hope will be useful for the authors to revise the manuscript before its publication.

Specific comments

Line144: "absolute wind vector"–> "wind vector"

Line157-163: It was difficult for me to understand the content of this paragraph. The authors appear to point out the possibility of weak bias of wind in JRA55, but its reasoning provided here is not clear. Is this related to the adjustment method of wind discussed in Sun et al. (2019)?

Line240-248: I think that the content of this paragraph appears to focus merely on a technical issue of the model and is not very useful.

Line295-296: In Figure 5, improvement from OMIP1 to OMIP2 can be found generally around the Eastern boundary regions of both Pacific and Atlantic basins. Therefore, rather specifically referring to Benguela region, the sentence here could be modified

such as "It is also the case for the Eastern boundary region in the Atlantic basin, but the warm bias is somewhat exacerbated offshore in OMIP-2".

Line323-325. This sentence is not clear. Do the authors just describe slight difference between OMIP-1 and OMIP-2 in (northern) equatorial Pacific area?

Line335-336: How about mentioning about the largest difference in the Arctic Ocean? (This seems related to salinity difference there)

Line444-445: It would be better to replace the word "hiatus" by "slowdown".

Section 6 (Line492-525): Many figures are prepared for this section (Figs. 25-31) with very short description provided. It is nice to see improvement from OMIP-1 to OMIP-2 in some statistics here but it appears better that the authors focus on the key result in the main text and most of the figures will be moved to Appendix.

Line573-574: "will be therefore become"–>"will therefore become"

―――――――――――――――

---

## Author Comment (AC1) · 22 Apr 2020

**Author responses to reviewer comments**

**1 **Responses to Reviewer #1 (Frank Bryan)**

**General Comments**

• Reviewer comment:

This manuscript is the logical successor to Griffies et al (2016) defining the OMIP protocol and Tsujino et al (2018) describing the construction of the JRA55-do forcing dataset. The collective efforts of the world's leading ocean modeling groups both in preparing for this study (preparing the forcing data, developing and agreeing on the protocol, running the experiments) and in collecting and collating the results is monumental.

The manuscript documents that the massive effort took place, but to be perfectly honest, it is as dull as dirt. Paragraph after paragraph begins "Figure N presents ...", "Figure N+1 presents ...", going through the catalog of standard metrics. The authors set a low bar by declaring they will offer only a "glimpse rather than an in depth view of the many elements of ocean model performance" (line 93). With 150 pages of material and several hundred figures, I believe there is more than can be described as a "glimpse", but agree that an in depth view is not offered. I guess this is the consequence of CMIP-ification of climate science. I am sure many groups will use the figures at some point to calibrate their efforts going forward, and more in depth studies will follow, but I would have hoped that we might have found a little more introspection on the successes and shortcomings of the protocol as well as more on the impacts of the structural changes in the forcing data. For example:

- Does each metric considered add value to the assessment, e.g., Do we need 0-700m heat content and SSH metrics or would one or the other be sufficient to discriminate among the included models?
- Will these metrics be relevant as resolution (and resolved variability) increase? There is already some indication that certain of these metrics become misleading.
- Does a change in ordering among models in various metrics in OMIP-1 vs OMIP-2 suggest the importance or not of different aspects of the forcing? What does the change in spread across the ensemble imply about the forcing?
- Are there any obvious groupings of models (e.g. the NEMO models or the hybrid coordinate models) in model skill metrics or not?
- Did variance in the solutions during in the pre-satellite era change more or less as compared to the later years between OMIP-1 and OMIP-2?
- The Tsujino et al (2018) manuscript calls out several "notable differences between CORE and JRA55-do" (pg 106, first pp). Are these apparent in the solutions?
- How did the additional variability in runoff included in JRA55-do forcing impact the solutions?

In short, what are the high-level conclusions that we can draw about the value of this exercise? I doubt that this question will be addressed in subsequent studies, so this is the obvious place to address it.

**• Author's response:**

Firstly, we would like to thank reviewers for their time and effort to review this paper and to provide constructive comments. We acknowledge that the discussion paper is not clearly summarizing the outcome of the overall effort and is failing to convey some important messages to the reader. The following are the main conclusions based on the analysis originally conducted and the additional analysis conducted for this revision:

- Both OMIP-1 and OMIP-2 ensembles capture observations, while the multi-model spread greatly exceeds the difference caused by the change in forcing datasets.
- Many ocean climate indices are very similar between OMIP-1 and OMIP-2 simulations, and yet we could also identify key qualitative improvements in transitioning from OMIP-1 to OMIP-2, which represents a new capability of the OMIP2 framework for evaluating process-level responses.
- A clear distinction is found between the metrics that are directly forced and those that require complex model adjustments, causing well-ordered and potentially less-organized responses among models to a change in forcing, respectively.

In the revised version, the following modifications will be incorporated:

One of our key findings, that models tend to disagree with each other more than the forcing products do, or more specifically, the multi-model spread greatly exceeds the difference between the two datasets, will be more highlighted in the revised version. To reinforce this conclusion, we explicitly quantify as many metrics as possible. For example, the figure of SST bias assessment (Figure 5 of the discussion paper) will be revised and look like Figure 1. The mean ensemble standard deviation exceeds the root-mean-square-difference between OMIP-1 and OMIP-2 simulations. Also, as shown in the middle panels, the regions where the observation is outside the 90% confidence range of the model spread ( $\pm 2\sigma$ ) are generally less than 15% of the global ocean, implying that both OMIP-1 and OMIP-2 ensembles capture the observation. The discussion will become clearer with such quantification. Also, we will extend the quantitative assessment of model biases and spreads to MLD and zonal mean temperature and salinity (Figures 9 through 12 of the discussion paper).

surface temperature (°C). Upper two panels show the bias of the multi-model mean, 30-year (1980-2009) mean SST relative to an observational estimate provided and updated by Program

for Climate Model Diagnosis and Intercomprison (PCMDI) following a procedure described by Hurrell et al. (2008) (hereafter referred to as PCMDI-SST). (a) OMIP-1 and (b) OMIP-2. The middle two panels show the standard deviation of the ensemble, with the regions where the observation is outside the 90 % confidence range of the model spread ( $\pm 2\sigma$ ) are hatched with red. (c) OMIP-1 and (d) OMIP-2. (e) Difference between OMIP-1 and OMIP-2 (OMIP-2 minus OMIP-1), with the regions where the difference is significant at 95% confidence level hatched with green. The uncertainty of multi-model mean is computed based on the method proposed by Wakamatsu et al. (2017). (f) 30 year (1980-2009) mean SST of PCMDI-SST.

- In doing the above analysis, we also found some features of potential importance about the
  ordering among models in the metrics. These quantifications will be summarized in new
  Tables and the outcome will be highlighted throughout the paper. The specific features are
  as follows.
  - For SST (rmse and mean), SSS (rmse and mean), SSH (rmse), sea ice extent (mean), MLD (rmse and mean), zonal mean temperature (rmse) in the Indian and the Pacific Oceans, and Indonesian Through Flow (mean), the change in ordering among models is small between OMIP-1 and OMIP-2. This may indicate that the behaviors of these metrics are largely determined by settings used by each model.
  - On the other hand, for some circulation metrics such as AMOC and GMOC (bottom water circulation), ACC, zonal mean temperature (rmse) in the Southern Ocean and Atlantic Ocean, zonal mean salinity (rmse), and the drift of vertically averaged temperature, the ordering among models is less consistent between OMIP-1 and OMIP-2. This may indicate that those metrics that involve thermohaline adjustment in models are sensitive to the differences in the forcing dataset.

Here we list some examples. Figure 2 shows scatter diagrams of rmse and mean of SST and SSS bias of the OMIP-1 and OMIP-2 simulations. The linear fitting and its  $r^2$ -score are also depicted. It would be notable that these metrics correlate well between OMIP-1 and OMIP-2. In other words, the ordering among models does not change significantly between OMIP-1 and OMIP-2. The implication would be that the behaviors of these metrics are largely determined by the settings used by each model.

Figure 3 shows the similar diagrams for metrics related to large scale circulations. Correlation coefficients are generally low except for the Indonesian Through Flow, which is thought to be determined by the model topography by the first order approximation. This implies that the metrics that involves thermohaline adjustments could show significantly different behaviors to different

---

## Author Response (AR1)

**Author responses to reviewer comments**

**1 Responses to Reviewer #1 (Frank Bryan)**

**General Comments**

- **Reviewer comment:**

  This manuscript is the logical successor to Griffies et al (2016) defining the OMIP protocol and Tsujino et al (2018) describing the construction of the JRA55-do forcing dataset. The collective efforts of the world's leading ocean modeling groups both in preparing for this study (preparing the forcing data, developing and agreeing on the protocol, running the experiments) and in collecting and collating the results is monumental.

  The manuscript documents that the massive effort took place, but to be perfectly honest, it is as dull as dirt. Paragraph after paragraph begins "Figure N presents ...", "Figure N+1 presents ...", going through the catalog of standard metrics. The authors set a low bar by declaring they will offer only a "glimpse rather than an in depth view of the many elements of ocean model performance" (line 93). With 150 pages of material and several hundred figures, I believe there is more than can be described as a "glimpse", but agree that an in depth view is not offered. I guess this is the consequence of CMIP-ification of climate science. I am sure many groups will use the figures at some point to calibrate their efforts going forward, and more in depth studies will follow, but I would have hoped that we might have found a little more introspection on the successes and shortcomings of the protocol as well as more on the impacts of the structural changes in the forcing data. For example:
  - Does each metric considered add value to the assessment, e.g., Do we need 0-700m heat content and SSH metrics or would one or the other be sufficient to discriminate among the included models?
  - Will these metrics be relevant as resolution (and resolved variability) increase? There is already some indication that certain of these metrics become misleading.
  - Does a change in ordering among models in various metrics in OMIP-1 vs OMIP-2 suggest the importance or not of different aspects of the forcing? What does the change in spread across the ensemble imply about the forcing?
  - Are there any obvious groupings of models (e.g. the NEMO models or the hybrid coordinate models) in model skill metrics or not?
  - Did variance in the solutions during in the pre-satellite era change more or less as compared to the later years between OMIP-1 and OMIP-2?
  - The Tsujino et al (2018) manuscript calls out several "notable differences between CORE and JRA55-do" (pg 106, first pp). Are these apparent in the solutions?
  - How did the additional variability in runoff included in JRA55-do forcing impact the solutions?

  In short, what are the high-level conclusions that we can draw about the value of this exercise? I doubt that this question will be addressed in subsequent studies, so this is the obvious place to address it.

- **Author's response:**

  Firstly, we would like to thank reviewers for their time and effort to review this paper and to

provide constructive comments. We acknowledge that the discussion paper is not clearly summarizing the outcome of the overall effort and is failing to convey some important messages to the reader. The following are the main conclusions based on the analysis originally conducted and the additional analysis conducted for this revision.

- Both OMIP-1 and OMIP-2 ensembles capture observations, while the multi-model spread greatly exceeds the difference caused by the change in forcing datasets.
- Many ocean climate indices are very similar between OMIP-1 and OMIP-2 simulations, and yet we could also identify key qualitative improvements in transitioning from OMIP-1 to OMIP-2, which represents a new capability of the OMIP2 framework for evaluating process-level responses.
- A clear distinction is found between the metrics that are directly forced and those that require complex model adjustments, causing well-ordered and potentially less-organized responses among models to a change in forcing, respectively.
- Overall, our recommendation that future model development and analysis studies use the OMIP-2 framework is justified by the present assessment.

Regarding the impacts of the structural changes in the forcing data, our basic understanding about the simulation results is that OMIP-1 and OMIP-2 are similar. There are indeed some qualitative successes and shortcomings that arise by changing the forcing dataset, but relating these simulations results with the structural changes in the forcing data is not so simple as we have expected, except for the difference in the time series of the global mean sea surface temperature (specifically the erroneous warming of OMIP-1 sea surface temperature from the late 1970s to the early 1980s). Thus, we decided not to take a further step into this issue in the present assessment.

The third point above is new and would warrant some explanation. We found this feature in doing the quantitative analysis to confirm the second point above and as a response to the reviewer comment referring to ordering among the models in metrics.

The specific features are as follows.

- For SST (rmse and mean), SSS (rmse and mean), SSH (rmse), sea ice extent (mean), MLD (rmse and mean), zonal mean temperature and salinity (rmse) in the Indian Ocean, zonal mean salinity (rmse) in the Atlantic Ocean, and Indonesian Through Flow (mean), the change in ordering among models is small between OMIP-1 and OMIP-2. This may indicate that the behaviors of these metrics are largely determined by settings used by each model.
- On the other hand, for some circulation metrics such as AMOC, GMOC (bottom water circulation), and ACC, zonal mean temperature (rmse) in the Southern Ocean, and the drift of vertically averaged temperature, the ordering among models is less consistent between OMIP-1 and OMIP-2. This may indicate that those metrics that involve thermohaline adjustment in models are sensitive to the differences in the forcing dataset.

Here we list some examples. Figure A shows scatter diagrams of root-mean-square bias and mean bias of SST and SSS in OMIP-1 and OMIP-2 simulations. The linear fitting and its $r^2$-score are also depicted. It would be notable that these metrics correlate well between OMIP-1 and OMIP-2. In other words, the ordering among models does not change significantly between OMIP-1 and OMIP-2. The implication would be that the behaviors of these metrics are largely determined by the settings used by each model.

Figure B shows the similar diagrams for metrics related to large scale circulations. Correlation coefficients are generally low except for the Indonesian Through Flow, which is thought to be determined by the model topography by the first order approximation. This implies that the metrics that involves thermohaline adjustments could show significantly different behaviors to different forcing datasets.

[Figure]

**Figure A: Scatter diagram with linear fitting (red line) and its score ($r^2$) comparing SST bias rmse (upper left), SST bias mean (upper right), SSS bias rmse (lower left), and SSS bias mean (lower right) from OMIP-1 (abscissa) and OMIP-2 (ordinate). Note that this figure is not used in the revised version, only tables that list specific values are included (Table 3 and Tables D1−D8).**

[Figure]

**Figure B:** Scatter diagram with linear fitting (red line) and its score ($r^2$) comparing AMOC (upper left), bottom water circulation (upper right), ACC (lower left), and ITF (lower right) from OMIP-1 (abscissa) and OMIP-2 (ordinate). Note that this figure is not used in the revised version, only tables that list specific values are included (Table 3 and Tables D1−D8).

- **Author's changes in manuscript:**

  To highlight the main conclusions, the following modifications have been incorporated in the revised version:

  - Abstract and Section 7 (summary and conclusions) have been rewritten to more highlight the main conclusions.

  - One of our key findings, that models tend to disagree with each other more than the forcing products do, or more specifically, the multi-model spread greatly exceeds the difference between the two datasets, has been more highlighted in the revised version. To reinforce this conclusion, we have explicitly quantified as many metrics as possible. For example, the figure of SST bias assessment (Fig. 5 of the discussion paper) has been revised and looks like Fig. C (Fig. 6 of the revised version). The standard deviation of the model ensemble exceeds the root-mean-square difference between OMIP-1 and OMIP-2 simulations (Fig. C(e)). Also, as shown in the middle panels, the regions where the observation is outside the 95% confidence range of the model spread (±2σ) are generally less than 15% of the global ocean, implying that both OMIP-1 and OMIP-2 ensembles capture the observation. We think that the discussion has become clearer with such quantification. Also, we have extended the quantitative assessment of model biases and spreads to MLD and zonal mean temperature and salinity (Figs. 11 through 14 of the revised version). For the metrics consisting of time series of index values, *z*-scores of the differences are listed in Table 2, which further confirms that the difference between OMIP-1 and OMIP-2 simulations is not statistically significant in many metrics. These findings are summarized in Section 6 (Lines 638–642 of the marked-up text).

[Figure]

**Figure C (replacing Fig. 5 of the discussion paper as Fig. 6 of the revised version): Evaluation of the simulated mean sea surface temperature (°C). Upper two panels show the bias of the multi-model mean, 30-year (1980–2009) mean SST relative to an observational estimate provided and updated by Program for Climate Model Diagnosis and Intercomprison (PCMDI) following a procedure described by Hurrell et al. (2008) (hereafter referred to as PCMDI-SST). (a) OMIP-1 and (b) OMIP-2, with global mean bias and global root-mean-square bias depicted on the top. The middle two panels show the standard deviation of the ensemble, with the regions where the observation is outside the 95% confidence range of the model spread (±2σ) hatched with red. (c) OMIP-1 and (d) OMIP-2, with the global mean confidence range (twice the standard deviation) and the fraction of the region where observation is uncaptured by the model confidence range depicted on the top. (e) Difference between OMIP-1 and OMIP-2 (OMIP-2 minus OMIP-1), with the global root-mean-square difference depicted on the top. The regions where the difference is significant at 95% confidence level are hatched with green, with the uncertainty of multi-model mean difference computed based on the method proposed by Wakamatsu et al. (2017). (f) 30 year (1980–2009) mean SST of PCMDI-SST. All models are used for multi-model mean.**

- Specific values of metrics realized by individual models are listed in Tables D1−D8 of newly added Appendix D (Lines 1119−1137 of the marked-up text). Linear regression is applied to model scatters of the metrics and $r^2$-scores are listed in Table 3. Sensitivity of ordering among the models to the change in forcing is discussed in Section 6 (Lines 643–661 of the marked-up text).

- Statistical methods used in this study are explained in Section 2.3 (Lines 202–217 of the marked-up text).

- **Author's response to specific comments in the general comment**

  It may not be necessary to respond to all of the points suggested by reviewer #1 as examples for more careful consideration toward the improvement of the manuscript. Nonetheless, we list our responses to them, whether positive or negative, in the following.

- **Comment:** **Does each metric considered add value to the assessment, e.g., Do we need 0-700m heat content and SSH metrics or would one or the other be sufficient to discriminate among the included models?**

  **Response and change in manuscript:** In the revised version, with an intention to more streamline the description of the main text, we have added a sentence or two to discuss about the meaning and usefulness of the chosen metrics when each metric is assessed (e.g., Lines 595−598 of the marked-up text). Now most paragraphs in Sections 3 through 5 do not start with "Figure N presents …".

- **Comment:** **Will these metrics be relevant as resolution (and resolved variability) increase? There is already some indication that certain of these metrics become misleading.**

  **Response and change in manuscript:** It is noted in Appendix E (Line 1205−1210 of the marked-up text) that it will not be appropriate to apply some common metrics to both eddying and non-eddying models (e.g., interannual variability of sea surface height). This point is mentioned in Section 6 of the main text (Line 665−667 of the marked-up text).

- **Comment:** **Does a change in ordering among models in various metrics in OMIP-1 vs OMIP-2 suggest the importance or not of different aspects of the forcing? What does the change in spread across the ensemble imply about the forcing?**

  **Response and change in manuscript:** The revised version is now more quantitative and takes care of the ordering among the models as described above (The second paragraph in Section 6 (Lines 643–661 of the marked-up text) and Appendix D (Lines 1119–1137 of the marked-up text), Table 3 and Tables D1–D8). Regarding the change in spread across the ensemble, we did not observe particularly notable changes in spread due to the change in forcing datasets, except perhaps for the larger spread in OMIP-2 for the metrics involving thermohaline adjustments such as vertically averaged temperatures. We do not have a clear conclusion about this relatively larger spread in OMIP-2. It might be due to the lack of experiences with the OMIP-2 forcing dataset of modelling groups, which is mentioned in the text (Line 358–360 of the marked-up text).

- **Comment:** **Are there any obvious groupings of models (e.g. the NEMO models or the hybrid coordinate models) in model skill metrics or not?**

  **Response and change in manuscript:** In this assessment, we did not notice any obvious grouping of models in model skill metrics in terms of model formulation and model code. This is mentioned in Section 7 (Summary and Conclusion; Line 746−747 of the marked-up text). A minor exception is the interannual variability of sea-ice extent in summer of the northern hemisphere, where the models using CICE show large variability in their OMIP-1 simulations (Line 592−594 of the marked-up text).

- **Comment:** **Did variance in the solutions during in the pre-satellite era change more or less as compared to the later years between OMIP-1 and OMIP-2?**

  **Response:** In this assessment, we did not notice major change in the variance in the solutions between the pre-satellite and the satellite era (e.g., Figs. 19 through 22 of the revised version). This is not mentioned in the text.

- **Comment:** **The Tsujino et al (2018) manuscript calls out several "notable differences between CORE and JRA55-do" (pg 106, first pp). Are these apparent in the solutions?**
  **Response and change in manuscript:** The positive heat flux anomaly from the late 1970s to the early 1980s in the CORE forcing dataset (Fig. 22e of Tsujino et al. 2018) may explain the failure of OMIP-1 simulation to reproduce the gradual increase of SST during the 1980s. This is explicitly mentioned in the paragraph that discusses Fig. 21 of the revised version (Line 576–580 of the marked-up text).

- **Comment:** **How did the additional variability in runoff included in JRA55-do forcing impact the solutions?**
  **Response and change in manuscript:** More fresh water discharge from Greenland in the JRA55-do forcing may have at least partly impacted the initial decline of AMOC in the OMIP-2 simulations (Fig. 5 and Line 331–333 of the marked-up text). Our internal assessment implies that the recent increase in the runoff from Greenland does not have major impact on the AMOC variability and trend. But this would be worth investigating further in the future studies. This is stated in the text (Line 562–565 of the marked-up text).

**Specific comments and author responses**

- **Comment:** **Figures: I find the color bar used for positive definite quantities (e.g. 2b,d,f,h) very difficult to interpret. More contrast would be helpful.**
  **Response and change in manuscript:** A more contrasting color sequence has now been used in all relevant figures. For example, Figure 2 of the discussion paper (Fig. 3 of the revised version) looks like Fig. D of this document.

[Figure]

**Figure D (replacing Fig. 2 of the discussion paper as Fig. 3 of the revised version):** Globally averaged drift of multi-model mean horizontal mean (a, c) temperature (°C) and (e, g) salinity (practical salinity units (psu)) as a function of depth and time. The drift is defined as the deviation from the annual mean of the initial year of the simulation by each model. For each, time evolution of the standard deviation of the model ensemble is depicted to the right. (a, b) OMIP-1 temperature, (c, d) OMIP-2 temperature, (e, f) OMIP-1 salinity, and (g, h) OMIP-2 salinity.

- **Comment:** **Figure 1 and similar: Some explanation of what accounts for the nearly instantaneous development of the ensemble spread in upper ocean heat content, SST etc would be helpful. Perhaps maps of the year 1 bias in each model and how it compares to the longer term mean bias. What structures are responding this rapidly? What can we learn from experiments integrated for a few years vs 360?**

  **Response and change in manuscript:** Regarding the apparently instantaneous development of the ensemble spread in some metrics, in particular the upper ocean heat content, the reason is that the models have somewhat distinct initial conditions. There are many details about model initialization that can create differences across models, most notably the methods each group uses to interpolate/extrapolate WOA to their grid/topography and how they initialize sea ice. In particular, the choices in how the bottom topography is constructed for a given model can result in significant differences in such volume average fields. And these differences could affect the initial adjustment processes in models as well. This issue was encountered by the earlier CORE studies such as Griffies et al (2009) and Griffies et al (2014). We continue to perform the initialization using distinct methods across groups for CMIP6-OMIP. This relaxed protocol for initialization is partly because we are not here focused on prediction (an initial value problem) but instead are most concerned with variations and trends after the initial adjustment phase. However, we should think about this issue more carefully in the next phase of this comparison effort.

  To explain the problem in a simple way and to help explain a new figure (Fig. 8) showing the similarity of biases between initial and later years, we add a new figure showing spin-up behavior of SST and SSS in the simulations as Fig. 1 of the revised version. Relevant discussions are included (Line 248-270 of the marked-up text).

  Regarding the implications of the first years of integration for later model biases, the spatial pattern of biases in later years is indeed discernible in the initial years of SST and SSS as shown in Fig. E (Fig. 8 of the revised version). This may not necessarily apply to other metrics, but we think that this would be worth mentioning and include Fig. 8 in the revised version (Line 410–418 of the marked-up text).

  It might not be an ideal approach to add these two new figures given all those materials already put in the paper, but we think that these figures and relevant descriptions are necessary and useful. We hope that the reviewer will agree to this approach.

[Figure]

**Figure E (inserted as Fig. 8 of the revised version):** Comparison of SST (a,b) and SSS (c,d) biases relative to observations (PCMDI-SST and WOA13v2, respectively) for the initial 5-year mean (left panels) and the long-term mean (1980-2009) in the last cycle (right panels) of the OMIP-2 simulation using MRI.COM. Pattern correlation of biases between the initial 5 year mean and the long-term mean in the last cycle is 0.75 for SST and 0.85 for SSS.

- **Comment: Line 303 and following: A comparison at a subsurface (maybe 50m) depth would be more enlightening to factor out the influence of salinity restoring.**

  **Response and change in manuscript:** We compared salinity distributions at 0 m, 50 m, and 100 m depths but they look qualitatively similar (not shown). Instead, we show the difference between salinity to which sea surface salinity is restored in OMIP-1 and OMIP-2 (Fig. F(f), Fig. 7f of the revised version). Figure F(f) indicates that the difference in salinity used for restoring is having nontrivial effect on the simulated difference in salinity of the Arctic Ocean (Fig. F(e)), although a more dedicated analysis would be necessary to thoroughly understand the simulated difference considering the many other processes contributing to determining the salinity fields in the Arctic Ocean. A relevant change in the text is found in Line 401–403 of the marked-up text.

[Figure]

**Figure F (replacing Fig. 6 of the discussion paper as Fig. 7 of the revised version):** Evaluation of simulated sea surface salinity (psu). Upper two panels show the bias of the multi-model mean 30-year (1980–2009) mean SSS relative to WOA13v2 (Zweng et al. 2013). (a) OMIP-1 and (b) OMIP-2. The middle two panels show the standard deviation of the ensemble, with the regions where the observation is outside the 95% confidence range of the model spread (±2σ) hatched with red. (c) OMIP-1 and (d) OMIP-2. (e) Difference between OMIP-1 and OMIP-2 (OMIP-2 minus OMIP-1), with the regions where the difference is significant at 95% confidence level hatched with green as in Fig. 6. (f) Difference of salinity to which sea surface salinity is restored in OMIP-1 and OMIP-2 (OMIP-2 minus OMIP-1). On the top of each panel, global mean values are depicted as in Fig. C.

- **Comment:** Line 330: Would not a simple broadening of the front (irrespective of the occurrence of recirculation gyres) result in such a dipolar structure?

  **Response:** As shown in Fig. G(f), the observation (CMEMS: red) show a pair of positive and negative bumps relative to the multi-model mean (blue), which seems essential for the sharpening of the front along 35°N. It would also be notable that the observed sea surface height shows a peak just to south of the front (~33°N), implying the existence of a recirculation gyre. We would like to keep the text unchanged in the revised version.

[Figure]

**Figure G (replacing Fig. 8 of the discussion paper as Fig. 10 of the revised version, except for (f), which is kept unchanged (climatology of CMEMS)): Evaluation of simulated sea surface height (m). Upper two panels show the bias of the multi-model mean, 17-year (1993-2009) mean SSH relative to CMEMS. (a) OMIP-1 and (b) OMIP-2. The middle two panels show the standard deviation of the ensemble, with the regions where the observation is outside the 95% confidence range of the model spread (±2σ) hatched with red. (c) OMIP-1 and (d) OMIP-2. (e) Difference between OMIP-1 and OMIP-2 (OMIP-2 minus OMIP-1), with the regions where the difference is significant at 95% confidence level hatched with green. (f) Annual mean SSH of CMEMS (red) and OMIP-1 multi-model mean (blue) along 150.5°E in the northwest Pacific (cutting the Kuroshio Extension from south to north). Note that all SSH fields are offset by subtracting their respective quasi-global mean values before evaluation as described in Appendix C.**

- **Comment: Figure 9a,b: A nonlinear color scale would be helpful to bring out more than the deep water formation sites.**

  **Response and change in manuscript:** In the revised version, we show biases of the simulated mixed layer depths (Fig. H(a) and H(b), Fig. 11a and 11b of the revised version), but a nonlinear color scale has been used to show observational distribution of mixed layer depth (Fig. H(f), Fig. 11f of the revised version) and the simulated mixed layer depth of individual models in Figs. S27 and S28. The revised color scale certainly clarifies the detailed distribution in the relatively shallower mixed layer depth region.

[Figure]

**Figure H (replacing Fig. 9 of the discussion paper as Fig. 11 of the revised version):** Evaluation of simulated mixed layer depth (m). Upper two panels show the bias of the multi-model mean, 30-year (1980–2009) mean winter mixed layer depth in both hemispheres relative to observationally derived mixed layer depth data from de Boyer Montégut et al. (2004). January-February-March mean for the northern hemisphere and July-August-September mean for the southern hemisphere. (a) OMIP-1 and (b) OMIP-2. The middle two panels show the standard deviation of the ensemble, with the regions where the observation is outside the 95% confidence range of the model spread (±2σ) hatched with red. (c) OMIP-1 and (d) OMIP-2. (e) Difference between OMIP-1 and OMIP-2 (OMIP-2 minus OMIP-1), which is not statistically significant at 95% confidence level everywhere. (f) Observationally derived mixed layer depth data from de Boyer Montégut et al. (2004). On the top of each panel, global mean values are depicted as in Fig. C. Note that the regions where mixed layer depths could reach more than 1000 meters in winter, specifically the marginal seas around Antarctica (south of 60°S) and the high latitude North Atlantic (50°−80°N; 80°W−30°E) are excluded from the computation of global means.

- **Comment: Line 448 and following: It is notable that the SH mean bias improves more because the worst models get better.**
  **Response and change in manuscript:** The text has been revised according to this suggestion, which reads:

  "The overall reduction of the mean bias in the southern hemisphere in OMIP-2 in both seasons is due to the improvement of outliers." (Lines 586–587 of the marked-up text)

- **Comment: Line 455 and following, Figure 22: I found this to be perhaps the most important figure when considering the limitations of the wash-rinse-repeat OMIP cycling. We really do not capture 60 years of variability with a 60 year cycle. Worth emphasizing more strongly.**
  **Response and change in manuscript:** This limitation has been more emphasized throughout the paper in the revised version. For example, the following descriptions are added:

"In particular, further efforts are warranted to resolve remaining issues in OMIP-2 such as the warm bias in the upper layer, the mismatch between the observed and simulated variability of heat content and thermosteric sea level before 1990s, and the erroneous representation of deep and bottom water formations and circulations." (Line 52–54 (abstract) of the marked-up text)

"Overall, the OMIP simulations under the protocol of repeating many cycles of the entire period of the atmospheric forcing dataset do not capture variability of heat content and thermosteric sea level in the entire atmospheric dataset period. Only recent (after 1990s) upper layer heat content variability is reproduced. This limitation should be taken into account in analysing the results of the OMIP simulations." (Line 619–622 (section 5) of the marked-up text)

"Further common biases can point to limitations in the forcing datasets. One example includes the weak eastward North Equatorial Counter Current arising from the method used to adjust the wind field. Another is the mismatch between the observed and simulated variability of heat content and thermosteric sea level before the 1990s, presumably linked to the long ocean memory in comparison to the relatively short length of the OMIP forcing datasets." (Line 725–729 (section 7) of the marked-up text)

- **Comment: Appendix B2: Figure B4 a bit of over kill to make the point (did we really think Drake passage transport might depend on small differences in the properties of moist air?), but oh well, only four more panels among 400!**
  **Response and change in manuscript:** Figure B4 has been removed.

**Minor typos etc and author responses:**

- **Comment: Line 100: The four ... or All four ...**
  **Response and change in manuscript:** This has been corrected accordingly. (Line 110 of the marked-up text)

- **Comment: Line 224: smaller drift**
  **Response and change in manuscript:** This has been corrected accordingly. (Line 299 of the marked-up text)

- **Comment: Line 227: "subsurface" not clear what depth range is being described**
  **Response and change in manuscript:** The depth range of 100–500 m is intended, which has been reflected in the revised text. (Line 303 of the marked-up text)

- **Comment: Line 609: piston velocity**
  **Response and change in manuscript:** This has been corrected accordingly. (Line 794 of the marked-up text)

- **Comment: Line 610: 6 cycles (to be constant with rest of text)**
  **Response and change in manuscript:** This has been corrected accordingly. (Line 795 of the marked-up text)

- **Comment: Line 662: (CESM)**
  **Response and change in manuscript:** This has been corrected accordingly. (Line 851 of the

marked-up text)

- **Comment: Line 730: with OM4 configured**

  **Response and change in manuscript:** This has been corrected accordingly. (Line 920 of the marked-up text)

**2 Responses to Reviewer #2**

**General Comments**

- **Reviewer comment:**

  The manuscript describes overall results of ocean model intercomparision organized in the framework of OMIP-2. After the development of new surface boundary forcing dataset (JRA55-do; Tsujino et al. 2018), the performance of various ocean model simulations forced by this new dataset is now reported here.

  Under the same protocol proposed by the authors, eleven state-of-the-art global ocean models are forced by not only newly developed JRA55-do atmospheric dataset but also previously referred CORE forcing. This design makes it possible for the authors to clearly evaluate what stems from the difference from the surface forcing and what is from inter-model differences.

  In previous OMIP-1 comparisons, the CORE forcing by Large and Yeager (2009) was developed for surface forcing dataset. This dataset has been widely used for ocean model community but not updated after 2009, therefore, its replacement by newly developed JRA55-do is awaited. The results reported here provide us with the solid evidence that new JRA55-do dataset is good enough to replace CORE forcing as a new forcing dataset for global ocean simulations. The manuscript also presents timely and valuable assessment about the overall performance of the state-of-the-art global ocean models.

  Although the manuscript demonstrates the overall performance of global ocean simulations rather than detail analysis about specific topics, such documentation fits the scope of GMD and the ocean model and related communities will benefit from the results reported in this manuscript very much. Therefore, I can recommend the publication of this manuscript in GMD after minor revision. I have several comments which I hope will be useful for the authors to revise the manuscript before its publication.

- **Author's response**

  Firstly, we would like to thank reviewers for their time and effort to review this paper and to provide constructive comments. Please read the following for how we have responded to your specific comments/suggestions.

**Specific Comments and author responses**

- **Comment: Line144: "absolute wind vector"–> "wind vector"**
  **Response and change in manuscript:** This has been corrected accordingly (Line 166−167 of the marked-up text).

- **Comment: Line157-163: It was difficult for me to understand the content of this paragraph. The authors appear to point out the possibility of weak bias of wind in JRA55, but its reasoning provided here is not clear. Is this related to the adjustment method of wind discussed in Sun et al. (2019)?**
  **Response and change in manuscript:** There are two issues (relative versus absolute wind and with versus without surface ocean current imprints on winds) involved. This paragraph has been revised by adding a few sentences including referencing to relevant papers to complement the explanation. The paragraph reads as follows (Line 169−184 of the marked-up text):

"There also remains ambiguity as to what is represented by the prescribed winds ($\overrightarrow{U_a}$) depending on the way they are constructed from the satellite-based and reanalysis atmospheric wind products. This ambiguity becomes an issue with the OMIP-2 dataset. First, its wind field is based on the JRA-55 reanalysis, which assimilates scatterometer winds yet not necessarily reproduces winds identical to scatterometer winds depending on the level of assimilation constraints. Since scatterometer winds represent wind relative to the surface current (e.g., Plagge et al., 2012) and contain imprints of surface currents (Renault et al., 2017, 2019b), assimilating scatterometer winds directly, yet not identically, to the absolute surface winds of the atmospheric circulation model would make the feature of surface winds of the JRA-55 reanalysis somewhat ambiguous. Second, only the long-term mean JRA-55 winds are adjusted with respect to the satellite-based winds in constructing the OMIP-2 dataset (JRA55-do). As a result, the long-term mean winds of the OMIP-2 (JRA55-do) dataset could be regarded to be replicating their scatterometer wind counterparts, but ocean current imprints on them have not been clarified yet. On the other hand, in short time scales, ocean current imprints on winds are shown to be small, if not negligible, in the OMIP-2 (JRA55-do) forcing dataset (Abel, 2018), which would make them possible to be treated as absolute winds without imprints of surface currents at least in short time scales. A future version of the OMIP-2 dataset will aim to resolve this ambiguity. Readers are referred to Renault et al. (2020) for more discussion on the issues of using satellite derived winds to force uncoupled ocean models."

- **Comment: Line240-248: I think that the content of this paragraph appears to focus merely on a technical issue of the model and is not very useful.**
  **Response and change in manuscript:** The paragraph is intended to explain the reason why we do not adopt global mean salinity, which would be virtually constant, as metrics. In the revision, we have more explicitly stated this point (Line 316−325 of the marked-up text). Specifically,

  "In contrast to heat content, the total salt content in the ocean–sea-ice system is essentially constant in nature. In most participating models, the global salt content in the ocean–sea-ice system is explicitly conserved, which is achieved by removing the globally integrated salt flux arising from salinity restoring at each time step (salinity normalization) as noted earlier. The same adjustment is applied to surface freshwater flux in most participating models, resulting in conservation of total mass of water in the ocean–sea-ice system. Thus, in such models, variation of global mean salinity only occurs due to variation of sea-ice volume and the global mean salinity would not be normally employed as a metric for the purpose of model intercomparison. Figure 4 implies that global mean salinity increases for the first 10 to 15 years of each forcing cycle and then decreases for the rest of the cycle in both OMIP-1 and OMIP-2 simulations. It also implies that a long-term drift of global mean salinity does not occur in those models that have applied both salinity and freshwater normalization."

- **Comment: Line295-296: In Figure 5, improvement from OMIP1 to OMIP2 can be found generally around the Eastern boundary regions of both Pacific and Atlantic basins. Therefore, rather specifically referring to Benguela region, the sentence here could be modified such as "It is also the case for the Eastern boundary region in the Atlantic basin, but the warm bias is somewhat exacerbated offshore in OMIP-2".**
  **Response and change in manuscript:** Thank you for the suggestion. The text has been

corrected accordingly (Line 387−389 of the marked-up text).

- **Comment: Line323-325. This sentence is not clear. Do the authors just describe slight difference between OMIP-1 and OMIP-2 in (northern) equatorial Pacific area?**
  **Response and change in manuscript:** Yes, both OMIP-1 and OMIP-2 ensemble spreads fail to capture the observation there and we thought that this is worth mentioning. This part has been revised as follows (Fig. 10 and Line 431−433 of the marked-up document, see also Fig. G of this document):
  > "A zonally elongated pattern of positive bias occurs from the western to central basin in OMIP-1 and from the central to eastern basin in OMIP-2. Both OMIP-1 and OMIP-2 ensemble spreads fail to capture the observation there (Figs. 10c and 10d)."

- **Comment: Line335-336: How about mentioning about the largest difference in the Arctic Ocean? (This seems related to salinity difference there)**
  **Response and change in manuscript:** The largest SSH difference in the Arctic Ocean is now mentioned along with the salinity difference that could possibly explains this difference, which read as follows (Line 445−447 of the marked-up text):
  > "A large difference in sea surface height is found in the eastern Arctic Ocean, with OMIP-2 higher than OMIP-1. This difference is presumably related to the lower upper ocean salinity (and thus less dense water) found in OMIP-2 (Fig. 7e)."

- **Comment: Line444-445: It would be better to replace the word "hiatus" by "slowdown".**
  **Response and change in manuscript:** The word "hiatus" has been replaced by "slowdown" throughout the manuscript.

- **Comment: Section 6 (Line492-525): Many figures are prepared for this section (Figs. 25-31) with very short description provided. It is nice to see improvement from OMIP-1 to OMIP-2 in some statistics here but it appears better that the authors focus on the key result in the main text and most of the figures will be moved to Appendix.**
  **Response and change in manuscript:** Following the suggestion, section 6 has been moved to Appendix E.

- **Comment: Line573-574: "will be therefore become"–>"will therefore become"**
  **Response and Change in manuscript:** This has been corrected accordingly (Line 758−759 of the marked-up text).

[revised manuscript text omitted]
) omip1 | Winter MLD bias rmse (m) omip2 | Winter MLD bias mean (m) omip1 | Winter MLD bias mean (m) omip2 | Summer MLD bias rmse (m) omip1 | Summer MLD bias rmse (m) omip2 | Summer MLD bias mean (m) omip1 | Summer MLD bias mean (m) omip2 | North Atlantic (m) omip1 | North Atlantic (m) omip2 | Antarctica (m) omip1 | Antarctica (m) omip2 |
|---|---|---|---|---|---|---|---|---|---|---|---|---|
| AWI-FESOM | 44.88 | 45.55 | 10.33 | 10.22 | 13.79 | 14.48 | −5.62 | −5.66 | 2001.7 | 2089.0 | 1539.8 | 994.4 |
| CAS-LICOM3 | 55.94 | 55.16 | −10.59 | −15.17 | 19.28 | 18.62 | −8.08 | −8.46 | 1802.0 | 1674.8 | 523.0 | 392.4 |
| CESM-POP | 35.12 | 31.56 | 11.57 | 10.59 | 11.01 | 10.17 | 1.76 | 2.19 | 1654.2 | 1527.5 | 294.7 | 1200.5 |
| CMCC-NEMO | 37.02 | **30.90** | 12.60 | 1.69 | 10.99 | 12.94 | −5.04 | −9.13 | 1011.7 | 1713.6 | 1183.4 | 1209.2 |
| EC-Earth3-NEMO | 36.71 | 32.88 | 4.98 | 3.80 | 10.94 | **9.93** | −3.32 | −1.92 | 1216.9 | 1305.0 | 1918.0 | 1465.9 |
| FSU-HYCOM | 67.69 | 80.25 | 22.62 | 34.95 | 12.92 | 12.28 | 3.09 | 4.11 | 2269.8 | 2575.6 | 4136.7 | 3368.4 |
| GFDL-MOM | **33.62** | 32.59 | −7.70 | −9.47 | **10.46** | 10.02 | −4.07 | −3.86 | 2641.7 | 2501.3 | 1749.4 | 2094.8 |
| Kiel-NEMO | 39.43 | 35.78 | 8.59 | −0.73 | 11.96 | 14.12 | −7.25 | −10.77 | 1288.3 | 1656.0 | 357.9 | 524.5 |
| MIROC-COCO4.9 | 40.73 | 38.59 | 12.75 | 6.51 | 11.46 | 9.99 | 4.64 | 2.33 | 1678.0 | 1509.1 | 3680.0 | 876.9 |
| MRI.COM | 49.95 | 48.35 | 17.86 | 15.69 | 12.06 | 11.47 | −5.68 | −5.08 | 2270.0 | 1414.8 | 4764.8 | 4846.1 |
| NorESM-BLOM | 45.46 | 46.86 | 19.53 | 20.96 | 14.64 | 14.97 | −0.07 | 1.85 | 2150.9 | 2141.8 | 2500.3 | 1459.8 |
| MMM | 33.08 | 30.93 | 8.92 | 6.74 | 10.43 | 9.55 | −2.82 | −3.24 | - | - | - | - |
| ensemble mean | - | - | - | - | - | - | - | - | 1816.8 | 1828.0 | 2058.9 | 1675.7 |

[revised manuscript text omitted]

---

## Author Response (AR2)

**Author responses to topical editor comments:**

Thank you for handling our paper and also for the positive assessment and constructive comments. Here are our responses to the comments.

- **Comment:** *I find Figures A and B in the response, and the associated discussion, to be insightful. I understand that the numerical data is already included in the manuscript (Tables 3 and D1-D8), as is some of the discussion. However, I suggest that you consider adding the figures and the rest of the discussion to the manuscript as well.*

  **Response and change in manuscript:** Thank you for the suggestion. Following your suggestion, Figures A and B have been included in the revised manuscript as Figures 27 and 28, respectively, and the relevant discussion in section 6 has been revised (Lines 629–657 of the marked-up text).

- **Comment:** *The equations at lines 452-453 might be better formatted as stand-alone equations, rather than as inline text.*

  **Response and change in manuscript:** This part has been revised using stand-alone equations (Lines 434–442 of the marked-up text).

- **Comment:** *I agree with the statement at lines 522-523, but it should be supported by at least one reference.*

  **Response and change in manuscript:** Several references have been inserted to justify the importance of comparing mean zonal currents (Lines 503–508 of the marked-up text).

- **Comment:** *At line 646, the word "essentially" appears to be unnecessary.*

  **Response and change in manuscript:** The word "essentially" has been removed from the description (Line 634 of the marked-up text).

- **Comment:** *Your code availability statement says "Python scripts used to process data and generate figures (Tsujino et al., 2020b) are available at https://doi.org/10.26300/e178-4220. The most recent version is available at https://github.com/HiroyukiTsujino/OMIP1-OMIP2.". I have two comments on this: (a) It is undesirable to have two different versions of the same dataset. I do not see how the scripts can change after the manuscript has been published, so the github link appears to be unnecessary. (b) Will the Brown University Digital Repository allow you to edit the metadata for the dataset later, to reflect the final citation for the manuscript? If not, then please consider using a repository such as Zenodo instead.*

  **Response and change in manuscript:** The github link has been removed accordingly. Since the

Brown University Digital Repository will allow us to edit the metadata afterwards, we would like to keep using this repository for archiving relevant datasets (Lines 1165–1167 of the marked-up text).

- **Comment:** *You have archived the supplement separately on the Brown University Digital Repository. This is unnecessary, as the supplement will be permanently archived on the GMD website. Please remove the statement "Supplemental materials for this paper (Tsujino et al., 2020a) are available at https://doi.org/10.26300/1sgm-dz11.".*

  **Response and change in manuscript:** This statement has been removed (Lines 1173−1174 of the marked-up text).

It is noted that the following modifications are made in addition to the above modifications. All of them are recorded in track changes and printed in red.

- Citations of Seland et al. (2020) describing NorESM2 have been updated (Lines 971 and 1572−1577 of the marked-up text).
- Figures 19–22, and 24–25: Grid lines are depicted.
- Minor errors have been corrected (Lines 124−125, 270, 399, 464, 890, Table/Figure captions of the marked-up text).
- Table 2: Numerical round-off of z-scores is corrected for SST and SIV-NH.
- Table 3: The $r^2$-scores for sea ice extent are corrected. Observation had been included by mistake in the previous version.
- Table D1: Ensemble standard deviations are corrected. Variances are now divided by the number of models, not the number of models minus one.

[revised manuscript text omitted]